# FAM3A reshapes VSMC fate specification in abdominal aortic aneurysm by regulating KLF4 ubiquitination

Chuxiang Lei [1,4], Haoxuan Kan [1,4], Xiangyu Xian [1,4], Wenlin Chen [1], Wenxuan Xiang [1], Xiaohong Song [1,2], Jianqiang Wu [1,2], Dan Yang [3] & Yuehong Zheng [1]

Reprogramming of vascular smooth muscle cell (VSMC) differentiation plays an essential role in abdominal aortic aneurysm (AAA). However, the underlying mechanisms are still unclear. We explore the expression of FAM3A, a newly identified metabolic cytokine, and whether and how FAM3A regulates VSMC differentiation in AAA. We discover that FAM3A is decreased in the aortas and plasma in AAA patients and murine models. Overexpression or supplementation of FAM3A significantly attenuate the AAA formation, manifested by maintenance of the well-differentiated VSMC status and inhibition of VSMC transformation toward macrophage-, chondrocyte-, osteogenic-, mesenchymal-, and fibroblast-like cell subpopulations. Importantly, FAM3A induces KLF4 ubiquitination and reduces its phosphorylation and nuclear localization. Here, we report FAM3A as a VSMC fate-shaping regulator in AAA and reveal the underlying mechanism associated with KLF4 ubiquitination and stability, which may lead to the development of strategies based on FAM3A to restore VSMC homeostasis in AAA.

Abdominal aortic aneurysm (AAA) is characterized by permanent and irreversible dilatation of the abdominal aorta involving all three layers of the vascular wall and most commonly affecting the infrarenal part[1,2]. The prevalence of AAA varies worldwide between ethnic groups and sexes[3], and the overall mortality in AAA patients is high[4]. Generally, AAA is described chiefly as the destruction of aortic integrity driven by an acquired immune process such as inflammation[1]. However, the etiology and pathophysiological mechanisms underlying the formation and progression of AAA remain unclear. As the majority of the cells in the aortic wall, vascular smooth muscle cells (VSMCs) are not terminally differentiated and retain significant plasticity throughout the lifespan. VSMCs in the AAA focal tissue undergo a variety of heterogeneous differentiations in response to the microenvironment remodeling, namely, VSMC reprogramming, giving rise to a range of cell subpopulations[5]. VSMC reprogramming is crucial for vessel maturation or injury repair, yet it is indeed linked to vascular structural disturbance as well as vascular diseases.

Family with sequence similarity 3, member A (FAM3A) is a newly discovered gene that encodes a cytokine-like protein[6]. FAM3A is ubiquitously expressed in various tissues. Studies have demonstrated that FAM3A promoted mitochondrial ATP synthesis and release, insulin secretion, and glucose tolerance[7–11]. In addition, FAM3A had a significant antioxidant effect and alleviated cellular damage caused by oxidative stress[12]. Currently, David Sala et al. reported that FAM3A was

[1]Department of Vascular Surgery, Peking Union Medical College Hospital, Chinese Academy of Medical Sciences and Peking Union Medical College, Dongcheng District, Beijing 100730, China. [2]State Key Laboratory of Complex Severe and Rare Diseases, Peking Union Medical College Hospital, Chinese Academy of Medical Sciences and Peking Union Medical College, Dongcheng District, Beijing 100730, China. [3]Department of Computational Biology and Bioinformatics, Institute of Medicinal Plant Development, Chinese Academy of Medical Sciences and Peking Union Medical College, Haidian District, Beijing 100193, China. [4]These authors contributed equally: Chuxiang Lei, Haoxuan Kan, Xiangyu Xian. ✉e-mail: dyang0226@163.com; zhengyuehong2022@outlook.com

a Stat3 (signal transducer and activator of transcription factor 3)-regulated secretory factor that promoted oxidative metabolism and differentiation of muscle stem cells[13], suggesting that FAM3A would be a potential molecular switch that determined cell fate. Despite these striking findings, the role of FAM3A in aortic aneurysms, particularly in terms of VSMC reprogramming, remains unknown.

In this work, we report a downregulation of FAM3A in both AAA patients and murine models. Here, the strengthening of FAM3A is shown to reshape the cell fate and specification of VSMCs and dramatically attenuate AAA formation in mice. In addition, FAM3A/KLF4 axis plays a role under these circumstances.

## Results

### FAM3A expression is decreased in AAA patients and murine models

HE staining revealed a disturbance in the arrangement of smooth muscle cells, degradation of the extracellular matrix, and an increase in inflammatory cell infiltration (Fig. 1a), and Masson-trichrome staining indicated a marked increase in fibrosis in the patients' aneurysm wall (Fig. 1a). To characterize the change in FAM3A expression in AAA, we investigated the FAM3A protein and mRNA levels. FAM3A protein and mRNA levels were significantly decreased in AAA tissues compared to normal aortas (Fig. 1b, c and Supplementary Fig. 1a, b). In addition, using enzyme-linked immunosorbent assays (ELISAs), we observed that plasma FAM3A levels were decreased in AAA patients compared to control subjects (Supplementary Fig. 1b). The clinical characteristics of normal abdominal aorta donors and patients with AAA were provided in Supplementary Table 1. To further evaluate the roles of FAM3A in the development of aortic aneurysms, we constructed murine AAA models using angiotensin II (AngII)-infused ApoE$^{-/-}$ mice and elastase-treated C57BL/6 mice. Both aneurysm murine models exhibited significant increases in arterial diameter and corresponding pathophysiological changes (Fig. 1d, e suggested AngII-induced ApoE$^{-/-}$ AAA model; Fig. 1g suggested elastase-induced C57BL/6 AAA model). Furthermore, consistant with the findings in AAA patients, immunofluorescence staining, western blots, and qRT-PCR confirmed a lower expression level of FAM3A in murine aortic aneurysm tissues (Fig. 1f, h and Supplementary Fig. 1c). Interestingly, based on the colocalization results of immunofluorescence staining, we found that FAM3A was abundantly in the smooth muscle cells of aortic tunica media in mice (Fig. 1c, f).

### Cell-specific FAM3A expression changes in the AAA microenvironment

Using single-cell RNA (scRNA) transcript sequencing data in murine AAA models constructed by AngII-infused ApoE$^{-/-}$ mice[14] and CaCl2-treated C57BL/6 mice[15], we analyzed FAM3A expression levels in different cell types in the context of the AAA microenvironment. The gross FAM3A mRNA level was decreased in AAA tissues compared with normal aortas. However, different cell types exhibited different changes (Supplementary Fig. 1d, e). The results from scRNA transcript sequencing data in murine models and protein levels in the cultured primary cells, suggested that FAM3A was produced abundantly in VSMCs, endothelial cells, and fibroblasts, and few in macrophages as well as T cells (Supplementary Fig. 1d–g). Furthermore, we found that FAM3A mRNA and protein levels in the cells, especially VSMCs and fibroblasts were significantly decreased under AAA as well as pathological stimuli (such as PDGF and TNFα) conditions (Supplementary Fig. 1d–g). Taken together, these results suggested that FAM3A had a close association with AAA as well as the cells such as VSMCs, endothelial cells, and fibroblasts.

### FAM3A overexpression attenuates pathological outcomes in murine AAA models

Given the possible roles of FAM3A in the formation of aortic aneurysm, we overexpressed FAM3A in mice using FAM3A adenovirus, and

found that the diameter of aneurysms was significantly reduced and survival rate was significantly increased in the FAM3A-overexpressing mice compared to those in the empty vector virus group (Ad-sham) (Fig. 2a, b). HE staining of aneurysm slides showed that the degree of inflammatory cell infiltration in FAM3A overexpression group was significantly lower than that in control group (Fig. 2a, b). Moreover, the levels of interleukin (IL)1β, IL6, and TNFα in plasma and AAA tissues were significantly lower in FAM3A overexpression group than in Ad-sham control group (Fig. 2c and Supplementary Fig. 2e). In addition, FAM3A did not significantly affect the AngII-induced increase in blood pressure (Fig. 2b). Similarly, overexpression of FAM3A in C57BL/6 mice attenuated elastase-induced abdominal aortic aneurysm (Supplementary Fig. 2a, b). Furthermore, the levels of inflammatory cytokines in plasma and AAA tissues were significantly lower in FAM3A overexpression group than in Ad-sham control group (Supplementary Fig. 2c, e). In addition, we detected a significant reduction in the expression of matrix metalloproteinases (MMP) 2 and MMP9, which are closely related to the development of AAA, in the aortic tissues of the FAM3A-overexpressing mice with AAA (Fig. 2i and Supplementary Fig. 2d).

Using RNA sequencing data from AngII-infused ApoE$^{-/-}$ aneurysm tissue, we identified 386 differentially expressed genes (DEGs) in FAM3A overexpression tissues compared to those of Ad-sham control tissues ($|\log_2 FC| > 1$ and FDR < 0.05): 192 upregulated and 194 downregulated transcripts (Fig. 2d). Functional enrichment analysis of Kyoto Encyclopedia of Genes and Genomes (KEGG) revealed several pathways that may play crucial roles related to FAM3A and AAA, including metabolism, PPAR signaling pathway, and cytokine-cytokine receptor interaction (Fig. 2e). We next extracted genes associated with VSMC phenotypic switching and differentiation from the KEGG database and compared their expression levels in two groups of mice (Fig. 2f). The otherwise reduced expression level of the marker genes of contractile VSMCs (including Myocd, Srf, Myh11, Cnn1, Cnn2, and Tagln) in aneurysms was rescued in FAM3A-overexpressing aortas (Fig. 2f, g). In addition, the expression levels of Olfm2 and Mkl1/Mrtfa were decreased in the FAM3A overexpression group (Fig. 2g). Some significant gene expression was further proven by qRT-PCR (Fig. 2h). At the protein level, similar trends and results were detected (Fig. 2i and Supplementary Fig. 2d), suggesting a potential role of FAM3A in rescuing the VSMC loss-of-function phenotype (namely, dedifferentiation). However, KLF4, a crucial regulator of VSMC phenotypic switching, was not significantly altered at the transcriptional level after FAM3A overexpression at the 4-week timepoint in AngII-infused ApoE$^{-/-}$ mice (Fig. 2h).

Additioinally, the in vivo distribution of exogenous FAM3A by systemic administration with adenovirus was determined by the flag fusin protein. As shown in Supplementary Fig. 4a, FAM3A-flag was located in VSMCs, endothelial cells, fibroblasts, and macrophages in the AAA microenvironment, suggesting that the exogenous FAM3A may be expressed in VSMCs and other cells, and may regulate VSMC activity in autocrine and paracrine manners. Furthermore, we also detected exogenous FAM3A distribution in other organs including the heart, liver, and kidney, and the results showed that the FAM3A-flag was distributed in the heart, liver, and kidney (Supplementary Fig. 4b), suggesting that the endocrine mechanism may also be involved in the exogenous FAM3A function.

### Supplementation with recombinant FAM3A attenuates pathological outcomes in murine AAA models

Using FAM3A recombinant protein, we also validated the in vivo potential protective roles of FAM3A in an AAA murine model. FAM3A is highly genetically conserved among mammalian species (Supplementary Fig. 11a), so it is reasonable to use human recombinant FAM3A protein for administration in mice. The outline of the grouping is shown in Fig. 3a. First, FAM3A supplementation did not

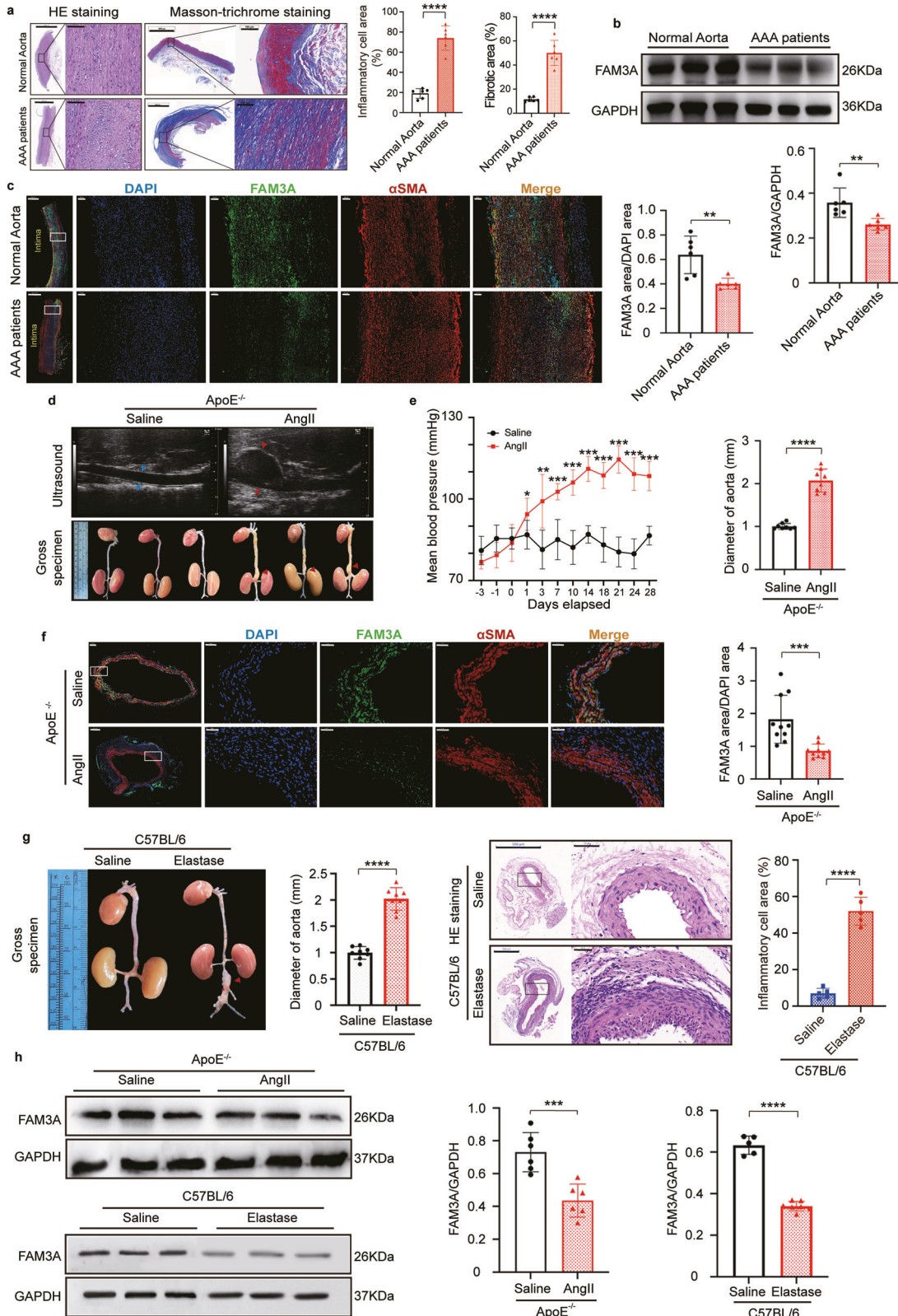

significantly affect the AngII-induced increase in blood pressure (Fig. 3b). Interestingly, by dynamic ultrasound monitoring, we found that the aortic diameter in the FAM3A-supplemented group was significantly smaller than that in the matched PBS-injected group as early as the first week after AngII exposure (Fig. 3c). In terms of survival, supplementation with recombinant FAM3A protein also resulted in a significant improvement (Fig. 3d). In addition,

FAM3A supplementation significantly lowered inflammatory markers such as IL1β, IL6, and TNFα both in plasma (Fig. 3e) and AAA tissues (Supplementary Fig. 2e) compared to the matched PBS group. Finally, we detected a significant reduction in the expression of MMP2 and MMP9 in the aortic tissues of the FAM3A-supplemented mice (Fig. 3f). Furthermore, FAM3A supplementation rescued contractile phenotype of VSMCs (Fig. 3f).

**Fig. 1 | The FAM3A protein level decreases in the arteries of AAA patients and murine models. a** Representative images of hematoxylin-eosin (HE) and Masson-trichrome staining of normal aortas and AAA of humans, and quantification of inflammatory cell infiltration and fibrosis in aorta tissues ($n = 6$ biologically independent samples). Scale bar: 2000 μm (HE staining) and 1000 μm (Masson-tri-chrome staining), insets: 200 μm (HE staining) and 100 μm (Masson-trichrome staining). **b** Western blot image and quantification to evaluate the expression of FAM3A in human aortic aneurysm compared to normal aorta ($n = 6$ biologically independent samples; quantitative comparisons between samples were run on the same gel). **c** Representative images and quantification of immunofluorescence staining identified FAM3A protein in human aortic aneurysm and normal aorta ($n = 6$ biologically independent samples). Scale bar: 1000 μm, insets: 100 μm. **d** Echocardiographic and gross specimen images of abdominal aortas from AngII-ApoE$^{-/-}$ murine AAA models and control mice (Representative images of $n = 8$ biologically independent animals). **e** Mean arterial pressure and arterial diameter were measured and quantified from AngII-ApoE$^{-/-}$ murine AAA models and control models ($n = 8$ biologically independent animals; *$P < 0.05$, **$P ≤ 0.001$, ***$P ≤ 0.0001$ versus Saline group in the term of Mean arterial pressure). **f** Representative images and quantification of immunofluorescence staining of FAM3A in aortas from AngII-ApoE$^{-/-}$ murine AAA models and control models ($n = 10$ biologically independent animals). Scale bar: 100 μm (Saline) and 200 μm (AngII), insets: 50 μm. **g** Gross specimen image and quantification of aortic diameter in Elastase-C57BL/6 murine AAA models and control models (left panel, $n = 8$ biologically independent animals); HE staining of abdominal aorta from Elastase-C57BL/6 murine AAA models and control models and quantification of inflammatory cell infiltration in aortic tissues (right panel, $n = 5$ biologically independent animals). Scale bar: 500 μm, insets: 50 μm. **h** Representative western blot images and quantification of the FAM3A expression levels in aortas from AngII-ApoE$^{-/-}$ and Elastase-C57BL/6 murine AAA models and their control models ($n = 6$ biologically independent animals in AngII-ApoE$^{-/-}$ modeling; $n = 5$ or 7 biologically independent animals in Saline or Elastase group, respectively in C57BL/6 modeling; quantitative comparisons between samples were run on the same gel). Data are presented as mean ± SEM (**a–c**, **e–h**). Statistical significance was calculated with two-tailed independent $t$ test (**a–c**, **e–h**) and $P$ values are indicated ($^{ns}P ≥ 0.05$, *$P < 0.05$, **$P ≤ 0.01$, ***$P ≤ 0.001$, ****$P ≤ 0.0001$). Source data are provided as a Source Data file.

We also detected the FAM3A function in female murine AAA models constructed by AngII-infused female ApoE$^{-/-}$ mice. Despite the higher mortality induced by AAA modeling in female mice than that in male mice, our results showed that the diameter of aneurysms was significantly reduced and the survival rate was increased in the mice supplemented with recombinant FAM3A compared to those in the matched control mice (Supplementary Fig. 3a, b). The plasma levels of IL1β, IL6, and TNFα were lower in the group treated with recombinant FAM3A than in the control group (Supplementary Fig. 3c). Moreover, the expression levels of MMPs in AAA tissues were suppressed in response to supplementation with recombinant FAM3A (Supplementary Fig. 3d).

Taken together, these results suggested that supplementation with recombinant FAM3A had a potential efficacy in alleviating AAA formation.

In addition, the in vivo distribution of exogenous recombinant FAM3A was determined by a c-myc tag protein. As shown in Supplementary Fig. 4c, recombinant FAM3A-c-myc colocalized with VSMCs, endothelial cells, and fibroblasts, and not so obviously with macrophages in the AAA microenvironment, suggesting that the exogenous FAM3A peptide maybe target VSMCs as well as other types of cells, leading to an overall inhibitory effects on AAA formation. Furthermore, we also detected exogenous FAM3A-c-myc distribution in other organs including the heart, liver, and kidney, and the results showed that the FAM3A-c-myc was present in these organs (Supplementary Fig. 4d), suggesting a potential function of recombinant FAM3A in other organs beyond AAA tissues. In addition, we explored the potential side effects of the systemic administration of recombinant FAM3A. The results showed that there was no significant toxicity of FAM3A in the heart, liver, kidney, and systemic circulation (Supplementary Fig. 4e).

To further investigate the effect of FAM3A on VSMC differentiation status, we stimulated human aortic smooth muscle cells with recombinant FAM3A protein. Our findings suggested that recombinant FAM3A peptide was able to increase the protein levels of VSMC contractile phenotype markers (Supplementary Table 2), including MYH11, SM-22α/TAGLN, CNN1, CNN2, MYOCD, and SRF, in the setting of PDGF-BB exposure in a concentration-dependent manner (Supplementary Fig. 5a). Through a VSMC contraction assay, improved contractility of VSMCs was also observed upon stimulation with recombinant FAM3A (Supplementary Fig. 5b). Furthermore, the elevated reactive oxygen species (ROS) levels in VSMCs caused by PDGF-BB or cholesterol induction could be inhibited by administration of recombinant FAM3A (Supplementary Fig. 5c). In addition, using RNA silencing of FAM3A in VSMCs, we detected a downregulation of contractile phenotype-related protein expression following stimulation with PDGF-BB (Supplementary Fig. 6a) or cholesterol (Supplementary

Fig. 6b), confirming the function of FAM3A in maintaining the contractile phenotype of VSMCs.

We also explored the cytotoxicity of recombinant FAM3A. The results showed that recombinant FAM3A had no significant toxicity in the normal cultured VSMCs, endothelial cells, fibroblasts, and macrophages (Supplementary Fig. 4f).

## FAM3A influences VSMC transdifferentiation toward other intermediate cells

During the course of VSMC differentiation reprogramming, the cells preferred to transdifferentiated toward several other intermediate cell types, including macrophage-, chondrocyte-, osteogenic-, adipocyte-, mesenchymal-, and fibroblast-like VSMCs. The markers selected to identify different cell types are provided in Supplementary Table 2. In the in vivo study, we detected a gross lower expression level of CD68 and ARG1 (macrophage), Aggrecan (chondrocyte), OPN (osteoblast), ADIPQ (adipocytes), CD34 (mesenchymal cells), and LUM (fibroblasts) in the FAM3A-overexpressing aneurysm tissues compared with the matched Ad-sham tissues (Fig. 4a). Furthermore, we sorted VSMCs (αSMA$^+$) in vivo by FACS with flow cytometry to quantify VSMC-related intermediate cell types (CD68$^+$: macrophage-like, RUNX2$^+$: osteoblast-like, CD34$^+$: mesenchymal-like, LUM$^+$: fibroblast-like, and ADIPQ$^+$: adipocyte-like). As shown in Fig. 4b, αSMA$^+$CD68$^+$, αSMA$^+$RUNX2$^+$, αSMA$^+$CD34$^+$, and αSMA$^+$LUM$^+$ cells were induced under AAA condition, and reversed by FAM3A overexpression mediated by adenovirus, suggesting the roles of FAM3A in suppressing VSMC transdifferentiation toward other intermediate cell types in vivo. In addition, in vivo immunofluorescence colocalization between specific markers of VSMCs (αSMA) and other intermediate cell types (CD68 for macrophages, Aggrecan for chondrocytes, RUNX2 for osteoblasts, CD34 for mesenchymal cells, and LUM for fibroblasts), showed suppressed VSMC-related intermediate cell transdifferentiation in response to FAM3A overexpression mediated by adenoviruses (Fig. 4c).

Consistent with the results for FAM3A overexpression in mice mediated by adenovirus, supplementation with recombinant FAM3A peptide also decreased the gross levels of CD68 and ARG1 (macrophages), Aggrecan (chondrocytes), OPN (osteoblasts), CD34 (mesenchymal cells), and LUM (fibroblasts) in female mice (Supplementary Fig. 3d) and male mice (Supplementary Fig. 7a) with AAA. Furthermore, an in vitro study showed that VSMCs stimulated by PDGF-BB (10 ng/mL for 48 h) failed to transdifferentiate toward other cell phenotypes in the presence of recombinant FAM3A. Markers of macrophages, including CD68, ARG1, and MAC2, were significantly downregulated when the FAM3A concentration was no less than 200 ng/mL (Supplementary Fig. 7b). The same trends were also found in the markers of chondrocytes (Aggrecan), osteoblasts (RUNX2 and

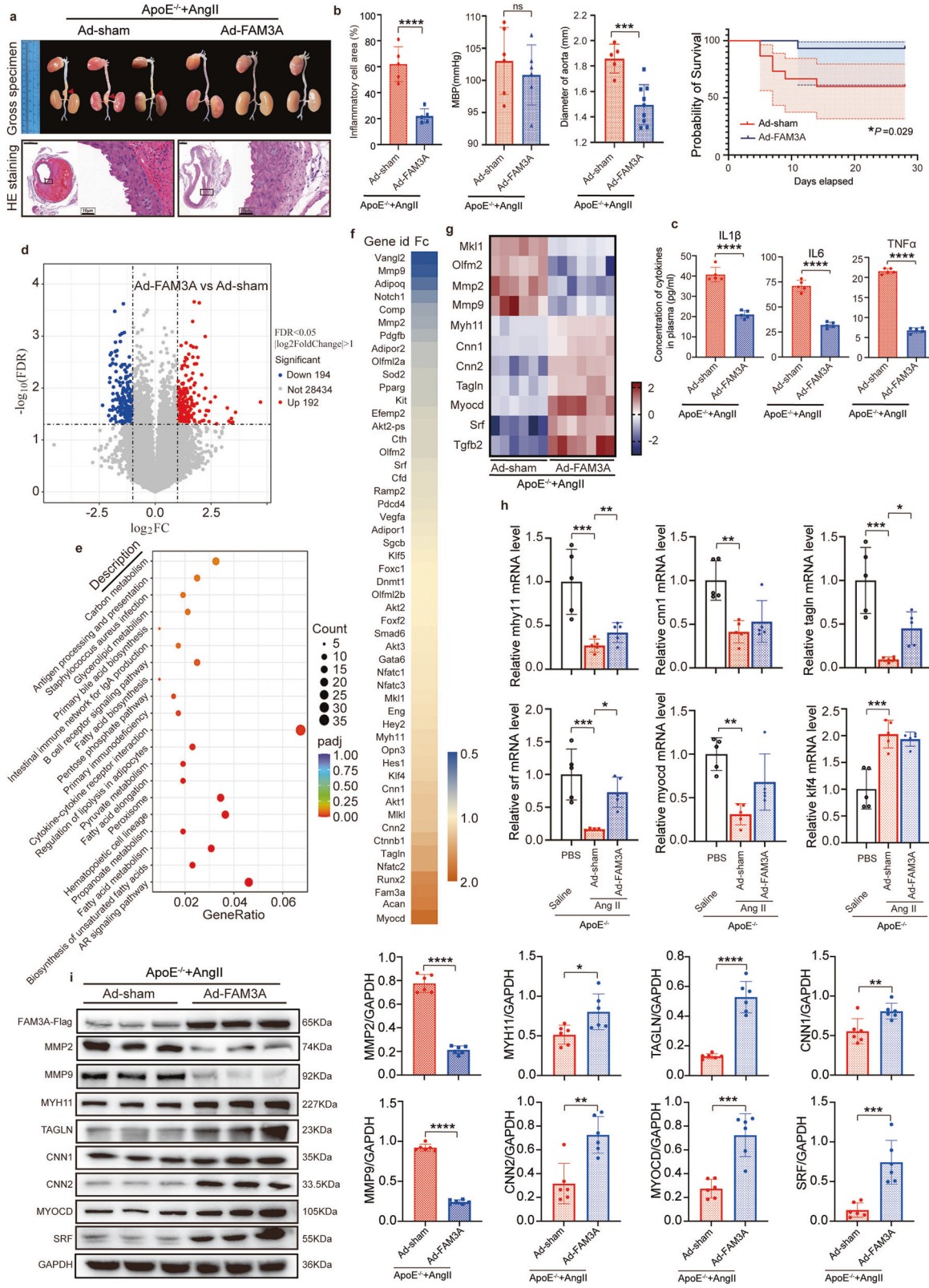

OPN), mesenchymal cells (CD34), and fibroblasts (LUM) (Supplementary Fig. 7b).

Interestingly, adiponectin (ADIPQ), a marker of adipocytes, was upregulated by FAM3A treatment in vivo and in vitro (Fig. 4a, b and Supplementary Fig. 7a, b), suggesting an impetus for VSMC transformation toward adipocyte-like intermediates in response to FAM3A stimulation.

## FAM3A influences KLF4 post-translational modification

KLF4 is a newly identified reprogramming factor of smooth muscle cells[16–18]. We therefore suspected that FAM3A would affect KLF4 signaling. First, we observed a significant increase in KLF4 production in aortic specimens from AAA patients and AAA murine models (Fig. 5a). Expectedly, the protein level of KLF4 in AAA tissues was significantly suppressed in both the FAM3A-overexpressing mice

**Fig. 2 | Overexpression of FAM3A in the AngII-ApoE$^{-/-}$ murine AAA model attenuates pathological outcomes. a** Gross specimen image and HE staining of aortas from AngII-ApoE$^{-/-}$ murine AAA models treated with or without FAM3A overexpression by adenovirus (Representative images of $n = 5$ or 8 biologically independent animals in Ad-sham or Ad-FAM3A, respectively). Scale bar: 500 μm (Ad-sham) and 200 μm (Ad-FAM3A), insets: 10 μm. **b** Inflammatory cell infiltration in the aorta ($n = 5$ biologically independent animals), mean arterial pressure ($n = 6$ biologically independent animals), arterial diameter ($n = 9$ or 14 biologically independent animals in Ad-sham or Ad-FAM3A, respectively), and survival probability ($n = 15$ biologically independent animals) were measured and quantified from mice treated as in (**a**). **c** ELISAs were used to evaluate the plasma inflammatory factors from mice treated as in (**a**) ($n = 5$ biologically independent animals). **d** Volcano plot showing differentiated expressed genes determined by RNA sequencing in aortas from mice treated as in (**a**). **e** Functional enrichment analysis of KEGG is shown.

**f, g** The differentially expressed VSMC-phenotype-switching-related genes (**f**) and the VSMC contractile marker genes (**g**) are shown in the heatmap. **h** The over-expressed crucial marker gene expression levels were evaluated by qRT-PCR ($n = 5$ biologically independent animals). **i** Representative western blot images and quantification of VSMC contractile marker proteins are shown in aortas from mice treated as in (**a**) ($n = 6$ biologically independent animals; quantitative comparisons between samples were run on the same gel). Data are presented as mean ± SEM (**b, c, h, i**) and mean ± 95% CI (Probability of survival in (**b**)). Statistical significance was calculated with two-tailed independent $t$ test (Inflammatory cell infiltration, MBP, Diamer of aorta in (**b**), (**c**), (**i**)), Kaplan–Meier analysis and log-rank test (Probability of survival in (**b**)) and one-way ANOVA followed by Tukey post hoc test (**h**) and $P$ values are indicated ($^{ns}P \geq 0.05$, $*P < 0.05$, $**P \leq 0.01$, $***P \leq 0.001$, $****P \leq 0.0001$). Source data are provided as a Source Data file.

and recombinant FAM3A-supplemented mice (Fig. 5b). Moreover, in vitro stimulation of VSMCs with recombinant FAM3A suppressed KLF4 protein level in a dose-dependent manner (Fig. 5d), and silencing of FAM3A increased KLF4 production (Fig. 5e). Recently, KLF2 and Zfp148 were shown to regulate VSMC phenotypes in AAA formation[19]. More interestingly, we found that FAM3A altered the protein expression of KLF2 and Zfp148 (Supplementary Fig. 8a), a clue for potential correlations between FAM3A and the genes involved in the regulation of the VSMC phenotypes.

Furthermore, we used nuclear extracts to explore the regulatory role of FAM3A in KLF4 nuclear localization. The in vivo results showed that total and phosphorylated KLF4 levels in the nuclei decreased in the FAM3A-treated group in contrast with the matched control group (Fig. 5c). Most importantly, with a confocal imaging system, our in vitro study showed that FAM3A stimulation significantly suppressed KLF4 nuclear localization (Fig. 5f). In addition, confocal imaging confirmed higher expression of SRF (namely, KLF4 DNA-binding competitor) in the VSMCs treated with FAM3A (Fig. 5f). Together, these findings suggest a suppressive effects of FAM3A on KLF4 nuclear localization in the AAA microenviroment as well as in the VSMCs.

Given that the mRNA expression of KLF4 was not downregulated significantly by overexpression of FAM3A most of the time in the AAA modeling process, the changes in KLF4 protein levels are probably due to post-transcriptional modifications (Fig. 2h and Supplementary Fig. 8b, c). KLF4 was reported to be regulated by post-translational modification, such as phosphorylation[16,20,21] and ubiquitin proteasome system (UPS)-mediated degradation[22]. Therefore, we explored whether FAM3A affected the KLF4 ubiquitination process resulting in reduced KLF4 levels. The VSMCs were incubated with MG-132, a UPS inhibitor. After proteasomes were blocked, the reduction in KLF4 levels caused by FAM3A was eliminated (Fig. 6a). Furthermore, the polyubiquitinated KLF4 proteins were accumulated in FAM3A-overexpressing or FAM3A-stimulated HEK293T cells (Fig. 6b) and FAM3A-stimulated VSMCs (Fig. 6d), suggesting an induction of the KLF4 ubiquitination process by FAM3A. Protein phosphorylation changes sometimes influence its ubiquitination status. We therefore investigated the level of phosphorylated KLF4 (Ser254) in the presence of FAM3A. As shown, the increased FAM3A concentration gradient was accompanied by a gradual decrease in the KLF4 phosphorylation level (Fig. 6c).

We also detected the involvement of KLF4 in the context of FAM3A biological function in vitro. As shown, FAM3A silencing induced an elevated ROS level in vitro, which could be significantly reversed by the KLF4 inhibitor kenpaullone (Supplementary Fig. 9a). Furthermore, downregulation of contractile marker expression levels in VSMCs due to FAM3A silencing was rescued after concomitant knockdown of KLF4 or administration of the kenpaullone (Supplementary Fig. 9b). Moreover, KLF4 silencing or inhibition abolished FAM3A silencing-induced increases in the expression levels of markers for macrophages (CD68, ARG1), chondrocytes (Aggrecan), osteoblasts

(RUNX2), mesenchymal cells (CD34), and fibroblasts (LUM) (Supplementary Fig. 9c). On the other hand, we further treated VSMCs with KLF4 inducer APTO-253 to detect whether KLF4 activation would abolish the FAM3A biological function. As shown in Supplementary Fig. 9d, APTO-253 reversed the effects of FAM3A on suppressing the expression of the markers for macrophages (CD68) and osteoblasts (OPN, RUNX2).

## The signaling pathways involved in VSMC differentiation reprogramming

We proceeded to define the mechanisms by which FAM3A induces such a regulation of the VSMC phenotype. Considering that FAM3A has been reported previously to activate the AKT and ERK1/2 pathways[7,8,23], and that the TGFβ pathway has been proven to be an essential regulator of VSMC fate commitment[24–26], we first explored the effects of FAM3A on the AKT, ERK1/2, and TGFβ pathways in cultured VSMCs. We found that FAM3A promoted the phosphorylation of AKT, ERK1/2, and SMAD3, and elevated TGFβ signaling (Fig. 7a). Furthermore, antagonists of AKT (MK-2206), ERK1/2 (U0126), and TGFβ/SMAD3 (SIS3) were administered to block the corresponding pathway. The results showed that the effects of FAM3A on maintaining the contractile phenotype of VSMCs were significantly abrogated in the presence of the TGFβ/SMAD3 antagonist SIS3 (Fig. 7b).

We further examined whether TGFβ was involved in the effects of FAM3A on hampering transdifferentiation of VSMCs toward other cell types. An in vitro study showed that TGFβ blockade induced a stronger tendency of VSMCs to transform toward macrophage-like, osteoblast-like, MSC-like, and fibroblast-like VSMCs (Fig. 7c).

We also questioned whether TGFβ contributed to the effects of FAM3A on regulating KLF4 ubiquitination and production. As shown, phosphorylated KLF4 levels were significantly reduced by the stimulation with FAM3A, but this suppression level was significantly reversed by blockade of the TGFβ pathway (Fig. 7d). More importantly, the FAM3A-induced promotion of KLF4 ubiquitination and suppression of KLF4 nuclear localization were also inhibited by TGFβ blockade (Fig. 7e, f).

We also explored the involvement of TGFβ signaling in FAM3A biological function in vivo. As shown, the protein level of TGFβ in AAA tissues was significantly increased in both the FAM3A overexpression group and recombinant FAM3A supplementation group compared to their matched control groups (Fig. 8a). The effects of FAM3A on maintaining the contractile phenotype of VSMCs were significantly abrogated in the presence of the TGFβ/SMAD3 antagonist SIS3 or KLF4 inducer APTO-253 (Fig. 8b and Supplementary Fig. 10a). Furthermore, the in vivo study showed a strong tendency of VSMCs transforming to intermediate cell types following additional TGFβ/SMAD3 blockade or KLF4 induction, as shown by western blots (Fig. 8c and Supplementary Fig. 10b) and flow cytometry (Fig. 8d). In addition, the suppressed ROS production by FAM3A treatment was abrogated by the administration of SIS3 in vivo (Fig. 8e).

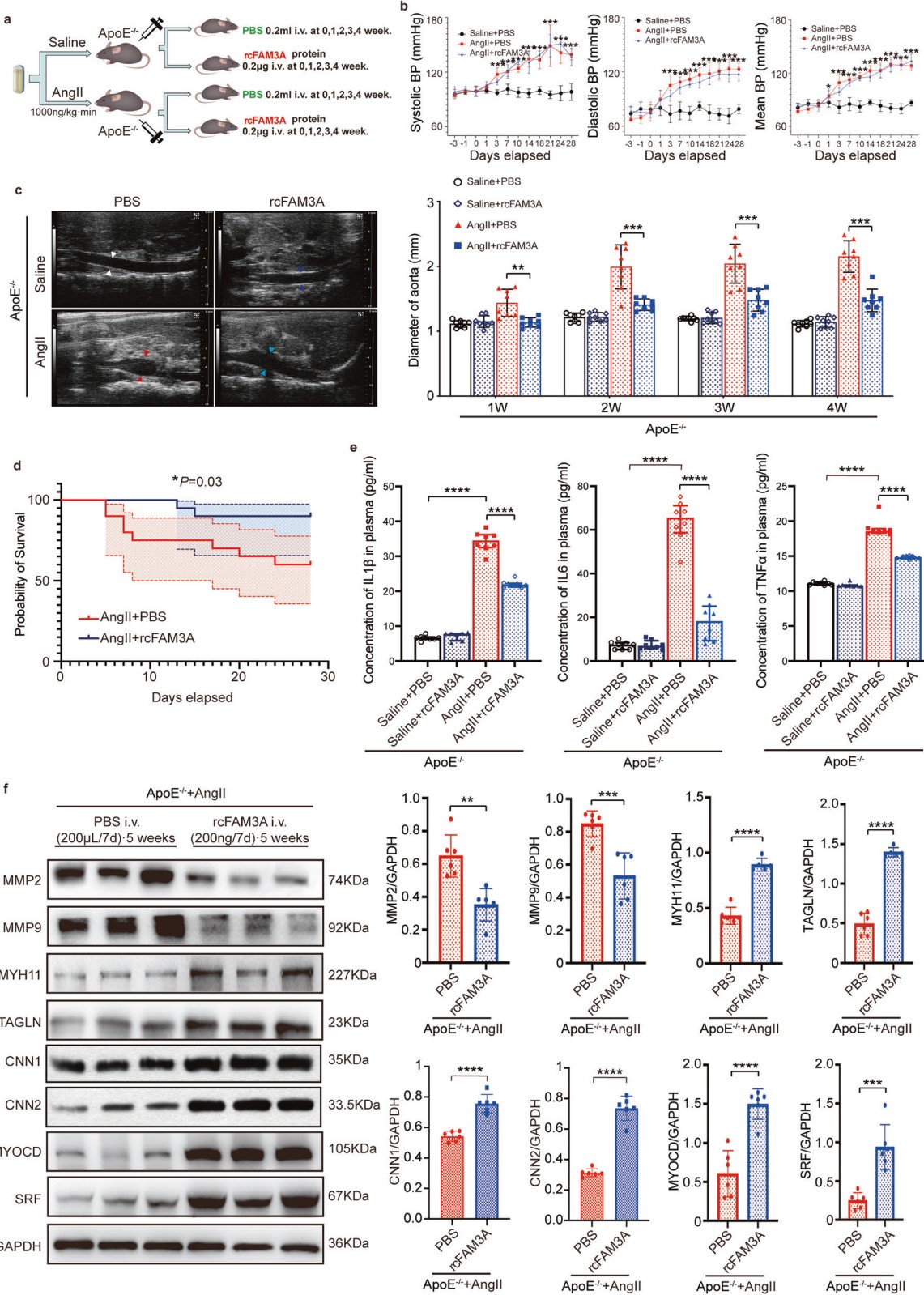

We also used nuclear extracts to explore the in vivo role of TGFβ signaling in the context of the regulatory function of FAM3A in KLF4 nuclear localization. The results showed that TGFβ/SMAD3 blockade abolished the suppressive effect of FAM3A on the total and phosphorylated KLF4 levels in the nuclei (Fig. 8f). Taken together, these findings suggest the involvement of the TGFβ/KLF4 axis in FAM3A-mediated reshaping of smooth muscle cell phenotypes.

## Discussion

Although previous studies have documented the essential roles of FAM3A in metabolic process and vascular pathology[7, 8, 23, 27], virtually little is known about the role of FAM3A in VSMC differentiation events, especially in the setting of AAA. In the current study, we unveiled important effects of this newly identified cytokine-like protein, FAM3A, on VSMC fate specification in aortic aneurysms. For decades,

**Fig. 3 | Supplementation with recombinant FAM3A protein in the AngII-ApoE$^{-/-}$ murine AAA model ameliorates pathological outcomes. a** Schematic diagram of recombinant FAM3A intravenous injection in AngII-ApoE$^{-/-}$ murine AAA models. **b** Quantification of blood pressure is shown from mice treated as in a ($n = 6$ biologically independent animals; *$P < 0.05$, ***$P < 0.0001$ AngII + PBS group versus Saline + PBS group). **c** Representative echocardiographic images and quantification of diameters of abdominal aorta from mice treated as in (**a**) ($n = 8$ biologically independent animals). **d** Kaplan−Meier curve and log-rank test showed the survival probability from mice treated as in (**a**) ($n = 20$ biologically independent animals). **e** ELISAs were used to evaluate the plasma inflammatory factors from mice treated

as in (**a**) ($n = 8$ biologically independent animals). **f** Representative western blot images and quantification of MMPs and VSMC contractile marker proteins are shown in aortas from mice treated as in (**a**) ($n = 6$ biologically independent animals; quantitative comparisons between samples were run on the same gel). Data are presented as mean ± SEM (**b, c, e, f**) and mean±95% CI (**d**). Statistical significance was calculated with one-way ANOVA followed by Tukey post hoc test (**b, c, e**), Kaplan−Meier analysis and log-rank test (**d**), and two-tailed independent $t$ test (**f**), and $P$ values are indicated (*$P < 0.05$, **$P \leq 0.01$, ***$P \leq 0.001$, ****$P \leq 0.0001$). Source data are provided as a Source Data file.

reprogramming of VSMCs, namely, VSMCs differentiating toward subpopulations with different specified fates, has been recognized as an important event not only in the progression of various cardiovascular diseases, but also in the homeostasis of healthy vessels[28–34]. In particular, VSMC reprogramming which leads VSMCs to switch to a dysfunctional status, has been proven to be an initial impetus for aortic aneurysm progression[5, 35, 36]. Except for loss of function or cell death, VSMCs preferred to transdifferentiate toward several other intermediate cell types, including macrophage-, chondrocyte-, osteogenic-, mesenchymal-, fibroblast-, and adipocyte-like VSMCs, each being distinguished by the specific marker genes[16]. However, the regulators as well as the underlying mechanisms concerning VSMC reprogramming, especially in the setting of aortic aneurysms, remain to be elucidated.

Family with sequence similarity 3 (FAM3) is a newly discovered cytokine-like gene family[37]. As the first member of FAM3, FAM3A is ubiquitously expressed in multiple cell types and tissues[6]. Currently, the biological function of FAM3A is poorly understood. Notably, the localization and function of FAM3A are reasonably controversial. Previous studies have shown that FAM3A was mainly localized at mitochondria within cells[7, 8, 23]. David Sala et al. recently demonstrated that FAM3A was downstream of Stat3, and was detected in the Golgi complex and endoplasmic reticulum mediated vesicle transport, and finally secreted by myogenic cells[13]. Consistant with this report, our results showed that FAM3A could be a secreted cytokine, because VSMCs were responsive to the direct stimulation with recombinant FAM3A (Figs. 5d−f, 6 and 7 and Supplementary Fig. 1e), and FAM3A was secreted and detectable in the culture medium (Supplementary Fig. 1g) and plasma (Supplementary Fig. 1b, c). However, FAM3A is a gene with a large number of splice variants. Thus, FAM3A biology is much complicated and varied. The numerous splice variants of FAM3A could explain the discrepancy among FAM3A investigations reported currently. Therefore, further studies are needed to address alternative splicing of FAM3A. Moreover, there are other unsolved problems in understanding FAM3A from the perspective of cell and molecular biology. Specifically, its cellular receptor is unknown, and the cell types targeted by FAM3A as well as the molecules downstream of ligand-receptor complex are unknown.

Our study provided evidence that FAM3A was decreased in aortic tissues and plasma from AAA patients and murine models (Fig. 1b, c, f, h and Supplementary Fig. 1a−c). For a more comprehensive understanding of the linkage between FAM3A and AAA, we further dissected FAM3A cell-specific changes in the AAA microenviroment. Our findings suggested that FAM3A was produced and secreted abundantly in VSMCs, endothelial cells, and fibroblasts, and few in macrophages and other cell types, such as T cells (Supplementary Fig. 1d−g). Even if there were positive findings in endothelial cells and fibroblasts, in this study, we addressed the association between FAM3A and VSMCs in the context of AAA pathology. Notably, our results also suggested the effects of autocrine, paracrine, and endocrine FAM3A on the VSMC phenotypes due to the FAM3A expression not being unique in VSMCs and its presence in plasma.

Interestingly, despite the discrepancy presented in previous reports in terms of the cellular vesicle transport and localization of FAM3A, almost all studies support the role of FAM3A as an impetus for

oxidative respiration and intracellular ATP synthesis[7–9, 13, 23, 38]. Correspondingly, the rates of $O_2$ consumption, oxidative respiration, and glycolysis were closely related to the smooth muscle contractile capacity and VSMC differentiation state[39–41]. Now that FAM3A is decreased in AAA, we questioned whether supplementation with FAM3A has the capacity to retain the well-differentiated VSMC status and reverse the VSMC reprogramming induced by the AAA microenvironment. Because of the lack of knowledge of the FAM3A receptor, we first systematically explored FAM3A function by treating murine AAA models via systemic administration. In addition, considering the autocrine, paracrine and endocrine manners, we delivered FAM3A in mice with adenovirus-mediated overexpression and recombinant FAM3A peptide. It has been proposed that the transcription of myocontractile genes, such as transgelin (TAGLN/SM-22α) and calponin 1 (CNN1) was downregulated in aortic aneurysms[5, 35], indicating the loss of VSMC structural integrity in the aortic wall[42]. Our study showed that FAM3A inhibited VSMC dedifferentiation and maintained VSMC contractile phenotype both in vivo and in vitro (Figs. 2 and 3 and Supplementary Figs. 2, 5 and 6). Notably, our in vivo and in vitro data proved that FAM3A inhibited the transformation of VSMCs toward macrophage-, chondrocyte-, osteoblast-, mesenchymal-, and fibroblast-like phenotypes (Fig. 4 and Supplementary Fig. 7). Thus, FAM3A could be a cell fate-shaping regulator of VSMCs.

We next addressed an important mediator in our study, namely, KLF4. Numerous studies have explored the crucial role of KLF4 in regulating VSMC phenotypes and reprogramming[17, 18, 43]. KLF4 was reported to abolished well-differentiation of VSMCs, which was manifested by hampered MYOCD-SRF complex formation and therefore decreased expression of myo-contractile proteins[17, 44–46]. On the other hand, KLF4 also plays a central role in the transdifferentiation of VSMCs. KLF4 at least contributes to the mesenchymal-[43, 47], macrophage-[43, 48], and osteogenic-like[49, 50] VSMC subpopulation occurrence by inducing transcription processes of pertinent markers. As presented in our study, the expression level of KLF4, one of the most important transcription factors regulating the VSMC phenotype switching, was strikingly repressed by FAM3A (Fig. 5b, d), which was accompanied by its decreased nuclear localization (Fig. 5c, f). Taking these previous evidence and our findings together, we propose KLF4 as a crucial mediator involved in the regulatory effect of FAM3A on VSMC fate specification. Previous studies have also reported that KLF4 could be phosphorylated by PLK1 (polo-like kinase-1), promoting KLF4 protein stability and therefore participating in the hyperplastic intima of injured vessels[20, 21]. In addition, the phosphorylation of a transcription factor is generally accompanied by its enhanced nuclear localization. Hence, we subsequently explored the phosphorylation level and protein stability, specifically ubiquitination modification, of KLF4 in the setting of FAM3A stimulation. Interestingly, our data indicate that FAM3A preferentially impaired the phosphorylation of KLF4 (Ser254) (Fig. 6c), and promoted the ubiquitination-mediated degradation of KLF4 (Fig. 6b, d). The decreased phosphorylation level and protein expression of KLF4 explained its decreased nuclear localization.

Another finding in our study is that FAM3A activated the AKT, ERK1/2, and TGFβ/SMAD3 signaling pathways. The TGFβ pathway was

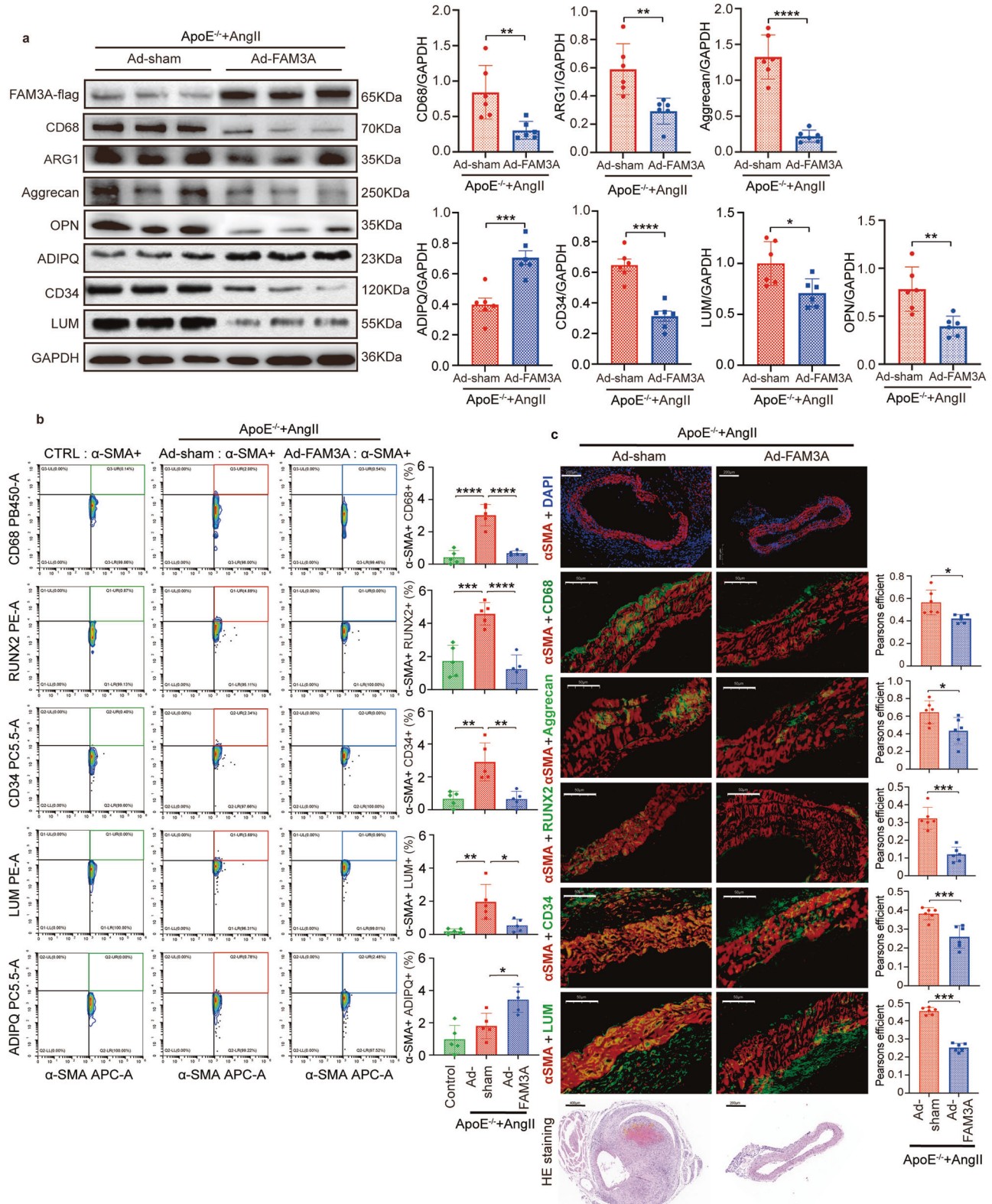

involved in FAM3A-mediated inhibition of the KLF4 phosphorylation level (Figs. 7d and 8b), protein amount (Figs. 7d and 8b), and nuclear localization (Figs. 7f and 8f), promoting KLF4 ubiquitination (Fig. 7e), and protecting against reprogramming of VSMCs (Figs. 7b, c and 8b–d). It has been documented that canonical TGFβ/SMAD3 signaling was enhanced after vascular injury and subsequently stimulated smooth muscle cell proliferation[24–26, 51, 52]. On the other hand, evidence suggested that a direct interaction of SMAD3 and SRF in response to

TGFβ stimulation played an essential role in the regulation of VSMC contractile genes[53–55]. Furthermore, disruption of TGFβR1 or TGFβR2 impaired aortic wall homeostasis and promoted aortic aneurysm formation[26, 56–58]. Recently, Chen et al. reported that abrogation of TGFβ signaling in combination with hypercholesterolemia induced strong VSMC reprogramming and transformation toward mesenchymal-, adipocyte-, chondrocyte-, osteoblast-, and macrophage-like cells[24]. Most interestingly, KLF4 was profoundly degraded in

**Fig. 4 | FAM3A influences VSMC transdifferentiation toward other intermediate cell types. a** Representative western blot images and quantification of markers of macrophages (CD68 and ARG1), chondrocytes (Aggrecan), and osteogenic cells (OPN) are shown in aortas from AngII-ApoE$^{-/-}$ murine AAA models treated with or without FAM3A overexpression by adenovirus (*n* = 6 biologically independent animals; quantitative comparisons between samples were run on the same gel). **b** Flow cytometry plot analysis of transdifferentiated cells in VSMCs in aortas from ApoE$^{-/-}$ control mice and AngII-ApoE$^{-/-}$ murine AAA models treated with or without FAM3A overexpression by adenovirus. Graphs show the percentage of VSMCs transdifferentiated toward the intermediate cell types of macrophages (αSMA$^+$CD68$^+$), osteogenic cells (αSMA$^+$RUNX2$^+$), mesenchymal cells (αSMA$^+$CD34$^+$), fibroblasts (αSMA$^+$LUM$^+$), and adipocytes (αSMA$^+$ADPIQ$^+$) (*n* = 5 biologically independent samples). **c** Representative images and quantification of immunofluorescent costaining of markers between VSMCs (αSMA) and macrophages (CD68), chondrocytes (Aggrecan), osteogenic cells (RUNX2), mesenchymal cells (CD34), or fibroblasts (LUM) respectively in aortas from mice treated as in (**a**) (red indicates αSMA, green indicates CD68, Aggrecan, RUNX2, CD34, or LUM respectively, and orange indicates αSMA$^+$CD68$^+$, αSMA$^+$Aggrecan$^+$, αSMA$^+$RUNX2$^+$, αSMA$^+$CD34$^+$, or αSMA$^+$LUM$^+$, respectively; *n* = 6 biologically independent animals). Scale bar (images of immunofluorescence): 200 μm (αSMA + DAPI) and 50 μm (for others), scale bar (HE staining): 400 μm (Ad-sham) and 200 μm (Ad-FAM3A). Data are presented as mean ± SEM (**a–c**). Statistical significance was calculated with two-tailed independent *t* test (**a, c**) and one-way ANOVA followed by Tukey post hoc test (**b**) and *P* values are indicated (*$P$ < 0.05, **$P$ ≤ 0.01, ***$P$ ≤ 0.001, ****$P$ ≤ 0.0001). Source data are provided as a Source Data file.

response to TGFβ signaling by posttranslational modification of ubiquitylation[59], and pSMAD2/3, the downstream of the TGFβ signaling pathway, directly bound to the KLF4 promoter and suppressed its expression[24]. This evidence indicates a decisive role of TGFβ and even TGFβ/KLF4 in VSMC reprogramming as well as AAA progression. Consistant with these observations, our study further supported the participation of TGFβ/KLF4 in AAA. In addition, we propose that the phosphorylation (Ser254) and ubiquitination levels of KLF4 were influenced by TGFβ in the setting of FAM3A biology. Rather than TGFβ/KLF4 signaling, the most significant finding in our study is about FAM3A, a VSMC fate-shaping regulator, which determines KLF4 ubiquitination and phosphorylation as well as VSMC fate specification through the TGFβ pathways.

Despite the interesting findings of FAM3A in shaping VSMC fate specification, there are still limitations and issues unknown in the present study. Our study addressed the protective effects of administration of FAM3A during the modeling of murine AAA. It would be significant to explore the therapeutic effects of FAM3A on preformed AAA using preformed AAA animals. The biological function of FAM3A is largely unknown currently, since the variable alternative splicing of FAM3A mRNA and the receptor are unclear. Furthermore, our study provided evidence of the function of FAM3A in AAA by systemic administration. These functions of FAM3A in VSMCs in vivo may be a combination of direct and indirect (such as cell–cell crosstalk) functions mediated by FAM3A. It could be interesting in the future to identify the FAM3A receptors and its targeting cells, and study FAM3A function and the underlying mechanisms cell-specifically, rather than a gross investigation using FAM3A systemic overexpression or supplementation. In addition, unraveling FAM3A endocrine, paracrine, and autocrine mechanisms may propose a potential perspective aimed at VSMC fate-shaping in AAA.

In summary, our study has unveiled a downregulation of FAM3A in AAA patients and two kinds of murine AAA models. Overexpression or supplementation with FAM3A attenuates the formation of AAA. As a VSMC fate-shaping regulator, FAM3A maintains well-defferentiated status of VSMCs, at least in part, by a marked activation of TGFβ signaling and thereby a suppression of KLF4 action (Supplementary Fig. 12). These findings offer a distinct perspective to develop further application strategies for AAA aimed at restoring VSMC homeostasis.

## Methods

### Ethics statement
Our research complies with all relevant ethical regulations and work with mice has been approved by Ethics Review Board of Peking Union Medical College Hospital.

### Human specimen collection
Six patients undergoing open abdominal aortic aneurysm resection were recruited (the basal clinical characteristics are provided in Supplementary Table 1), and normal abdominal aortas were acquired from non-AAA patients' surgical donor arteries. All the surgical specimens were washed in sterile saline and then frozen and embedded. Frozen slices were prepared to detect the changes in FAM3A protein expression. The use of human specimens was approved by the Ethics Review Board of Peking Union Medical College Hospital, Beijing, China. Written informed consent was obtained from all patients included in the study.

### Murine AAA model
For animal experiments, mice were housed in a controlled environment at a constant temperature (22–24 °C) and humidity (50–70%), under a 12 h–12 h light-dark cycle and provided with standard chow diet and water ad libitum.

### AngII infusion in ApoE$^{-/-}$ mice
AngII (Sigma-Aldrich, A9525) was infused into 10-week old male or female ApoE$^{-/-}$ mice (Beijing Huafukang Bioscience Co., Ltd, China) via subcutaneous osmotic pumps (Alzet, MODEL 2004; DURECT, Cupertino, CA) releasing a constant concentration of 1000 ng/kg/min for a maximum of 28 days[60]. Before and during the AAA modeling, the systolic and diastolic blood pressures were measured using the CODA® noninvasive BP system (Kent Scientific Co., Torrington, CT).

### Pancreatic elastase-induced AAA model
Male C57BL/6 mice (10–12 weeks old, Beijing Vital River Laboratory Animal Technology Co., Ltd, China) were anesthetized using an intraperitoneal injection of sodium pentobarbital (45 mg/kg body-weight) in accordance with the NIH Guidelines for the euthanasia of animals. After anesthetization, mice were kept on a constant temperature table to maintain body temperature at 37 °C. After median laparotomy, and the viscera were carefully separated using sterile gauze, and the posterior peritoneum was opened to expose the aorta from the level of the renal artery to the branch of the iliac artery. As previously described[61,62], the proximal and distal infrarenal aortas were temporarily ligated to create an aortotomy above the iliac bifurcation. Then, saline containing 20 U/ml Type I porcine pancreatic elastase (E1250, Sigma) was perfused with a 27 G indwelling needle (BD Biosciences) and fixed with a silk ligature for 5 min. For the control group, the porcine pancreatic elastase was heat-inactivated (100 °C) before use. Mice were administrated with 1% xylocaine and penicillin at the topical suture site immediately after the surgical procedure, and systemic application of opioids and NSAIDs was used for several days to relieve pain and prevent infection. All the mice continued to be fed for 14 days.

At the end of the AAA modeling (28 days for AngII-ApoE$^{-/-}$, 14 days for Elastase-C57BL/6), mice were euthanized by terminal heart puncture under deep isoflurane anesthesia followed by cervical dislocation, and the whole aortas were harvested under a stereomicroscope. The Ethics Review Board of Peking Union Medical College Hospital approved the animal study protocols (JS-2629).

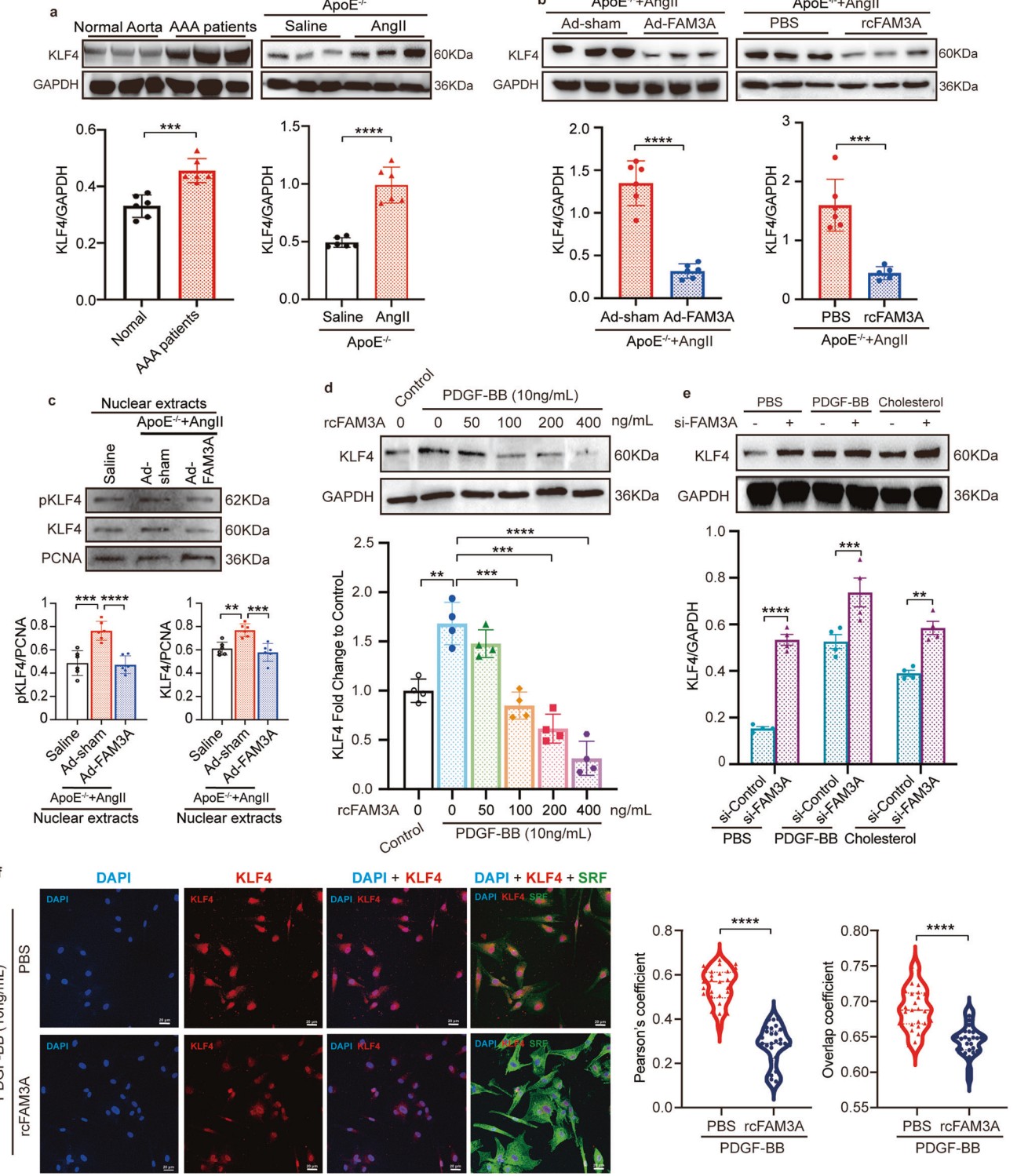

## Aortic diameter monitoring by abdominal ultrasound

Before and during modeling, the maximum abdominal aortic diameters were assessed by ultrasound using the Vevo 2100 platform (Visual Sonics, Toronto, CA) and MS-250 transducer (FUJIFILM VisualSonics). Mice were anaesthetized with 2% isoflurane and placed on a 37 °C heated plate. We performed a standardized imaging algorithm with longitudinal B-mode images during the systolic phase to obtain the inner aortic diameters. For all the experiments, the investigators were blinded to the group allocations during the measurements and data analysis, and the mice were tested in a randomized order.

## Overexpression of FAM3A in mice by adenovirus and aortic RNA sequencing analysis

In this study, we used an adenoviral vector (pAV[Exp]-EF1A>mFam3A [NM_001379181.1]/FLAG:IRES:mCherry) to overexpress the gene fragment of FAM3A with a FLAG tag in mice. Briefly, the pAd/CMV/V5-DEST Gateway Vector overexpressing mFAM3A was constructed and

**Fig. 5 | FAM3A suppresses KLF4 production and nuclear localization.**
**a** Representative western blot images and quantification of KLF4 in aortas from AAA patients and murine AAA models ($n = 6$ biologically independent samples or animals). **b** Representative western blot images and quantification of KLF4 in aortas from AngII-ApoE$^{-/-}$ murine AAA models treated with or without FAM3A overexpression by adenovirus (left, $n = 6$ biologically independent animals) and treated with or without recombinant FAM3A supplementation (right, $n = 6$ or 5 biologically independent animals in PBS or rcFAM3A group, respectively). **c** Representative western blot images and quantification of KLF4 signaling in aortic nuclear extracts from saline-ApoE$^{-/-}$ mice and AngII-ApoE$^{-/-}$ murine AAA models treated with or without FAM3A overexpression by adenovirus ($n = 6$ biologically independent animals). **d, e** VSMCs were cultured and treated with PDGF, cholesterol, recombinant FAM3A (**d**), or FAM3A siRNA (**e**) as indicated in the charts. Representative

western blot images and quantification of KLF4 in whole cell lysates are shown ($n = 4$ biologically independent experiments). Quantitative comparisons between samples were run on the same gel (**a–e**). **f** Confocal imaging showed the cellular localization of nuclei (blue), KLF4 (red), and SRF (green). The colocalization of nucleus and KLF4 was analyzed by ImageJ software with Pearson's correlation coefficient (PCC, left) and Manders' colocalization coefficients (MCC, right) quantifications ($n = 4$ biologically independent experiments with 24 scans for each group). Scale bar: 20 μm. Data are presented as mean ± SEM (**a–e**) and median ± IQR (**f**). Statistical significance was calculated with two-tailed independent $t$ test (**a, b, e**), one-way ANOVA followed by Tukey post hoc test (**c, d**), and ManneWhitney $U$ test (**f**) and $P$ values are indicated (*$P < 0.05$, **$P \le 0.01$, ***$P \le 0.001$, ****$P \le 0.0001$). Source data are provided as a Source Data file.

---

packaged and subsequently titered in the 293 A cell line using a commercial ViraPower Adenoviral Gateway Expression Kit (Invitrogen, Cat#K4930-00). The same vector without the mFam3A gene was used as a control. As adenovirus expression is generally maintained for no more than 2 weeks, we maintained continuous expression of the adenovirus vector in mice by multiple injections. Specifically, we diluted the adenovirus to a concentration of $1 \times 10^{11}$ virus particles (VPs)/mL using sterile saline at a volume of 200 μL (containing $2 \times 10^{10}$ VPs) per injection. For ApoE$^{-/-}$ mice, tail vein injections were performed 7 days prior to modeling, on the day of modeling, and 7, 14 and 21 days after modeling. A total of five injections were given, with each mouse being injected with a total of $1 \times 10^{11}$ VPs. For C57BL/6 mice, tail vein injections were performed 7 days prior to modeling, on the day of modeling, and 7 days after modeling. A total of three injections were given, with each mouse being injected with a total of $0.6 \times 10^{11}$ VPs due to its shorter modeling time.

For RNA sequencing, total RNA was extracted from aortic tissues of AngII-infused ApoE$^{-/-}$mice with or without FAM3A treatment using TRIzol (Ambion) following the manufacturer's instructions. RNA quality was determined by the RNANano 6000 Assay Kit of the Bioanalyzer 2100 system (Agilent Technologies, Co., Ltd., Santa Clara, CA). Libraries were prepared by an RNA Library Prep Kit for Illumina (New England Biolabs, Inc., Ipswich, MA) and filtered using the AMPure XP system (Beckman Coulter, Inc., Brea, CA). RNA sequencing was performed on an Illumina NovaSeq 6000 system. The read length was 150 bp, and 44 million to 56 million reads were generated per sample. Quality control of raw sequencing data was performed using FastQC and low-quality reads were removed. The filtered reads were aligned to the GRCm38 using Hisat2 (v2.0.5). FeatureCounts (v1.5.0-p3) was used to count the numbers of reads mapped to each gene. Differential gene expression analysis was performed using DESeq2 (v1.20.0) and edger (v3.22.5) packages in R, and the volcano plot was generated by ggplot2 package. Enrichment analysis of Gene Ontology (GO) and Kyoto Encyclopedia of Genes and Genomes (KEGG) was conducted by the cluster Profiler (v3.8.1) R package. All the RNA-sequencing data were deposited into the Gene Expression Omnibus (GEO) database with the accession number GSE230163.

### Administration of recombinant FAM3A protein in vivo
Human FAM3A (NM_021806) recombinant protein was purchased from OriGene (Cat#TP303495). Prior to use, we slowly dissolved the FAM3A protein at 4 °C and diluted the recombinant protein to a concentration of 1 μg/mL using sterilized PBS solution. The procedure was similar to the adenovirus overexpression injections in ApoE mice, i.e., 200 μl (containing 0.2 μg FAM3A protein) of recombinant protein solution was injected 7 days prior to modeling, on the day of modeling, and 7, 14 and 21 days after modeling, for a total of 5 injections of 1 μg per mouse.

### SIS3 and APTO-253 treatment in mice
Briefly, SIS3 (Cat#S0447, Sigma) at a dosage of 2.5 μg/g weight via intraperitoneal injection, daily[63] and APTO-253 (Cat#HY16291, MCE Chemicals & Equipment) at a dosage of 1 μg/g weight via intraperitoneal injection, daily[64–66] were administered to the mice for 4 weeks, whereas the control group were received solvent control (0.05% dimethyl sulfoxide) instead. Injections commenced one day before the first FAM3A administration.

### Cell culture and treatment
Primary human aortic smooth muscle cells (HASMCs) were purchased from the ScienCell Research Laboratories, Inc. (Cat#6110, San Diego, CA, USA) and cultured in smooth muscle cell medium (ScienCell, Cat#1101) containing 2% FBS, 100 U/ml of penicillin, and 100 μg/ml of streptomycin. HEK293T cells (Cat#CRL-3216, ATCC) were cultured in DMEM (Solarbio, China, Cat#12100-500) containing 10% FBS, 100 U/ml of penicillin, and 100 μg/ml of streptomycin. In the FAM3A stimulation experiments, we pretreated the cells with the appropriate concentration of FAM3A, followed by continuous stimulation of the cells with PDGF-BB (10 ng/mL) or cholesterol (10 μg/mL) for 48 h. On the other hand, three inhibitors were used in our study, MK2206, U0126 and SIS3, which were all stored at a concentration of 10 mM, dissolved in DMSO. For the experimental conduction, the storage solution was diluted at 1:1000 as a working solution, and then pretreated cells for 1 h. The control cells were added with the same volume of DMSO to remove the solvent effect. In addition, in some experiments, cells were pretreated for 1 h with KLF4 inducer, APTO-253, at a concentration of 10 μM

Primary human aortic endothelial cells (HAECs) were purchased from the ScienCell Research Laboratories, Inc. (Cat#6100, San Diego, CA, USA), and cultured in cell medium (ScienCell, Cat#1001) containing 2% FBS, 100 U/ml of penicillin, and 100 μg/ml of streptomycin. Primary human aortic adventitia fibroblasts (HAAFs) were purchased from the ScienCell Research Laboratories, Inc. (Cat#6120, San Diego, CA, USA), and cultured in cell medium (ScienCell, Cat#2301) containing 2% FBS, 100 U/ml of penicillin, and 100 μg/ml of streptomycin. Primary human macrophages (Cat#CP-H264, Procell Life Science & Technology, China) were purchased and cultured in DMEM (Solarbio, China, Cat#12100-500) containing 10% FBS, 100 U/ml of penicillin, and 100 μg/ml of streptomycin. For the detection of FAM3A expression or secretion, cells were exposed to TNFα (Cat#PRP1013, Abbkine, China, 25 ng/mL) or PDGF-BB (10 ng/mL) for 6 or 24 h, respectively.

### Transfection of small-interfering RNA
For knockdown of FAM3A or KLF4, the small-interfering RNA (siRNA) and control siRNA (Guangzhou RiboBio Co., Ltd.; siFAM3A Cat#-siG140909163939-1; siKLF4 Cat#siB170829015028-1) were transfected according to the manufacturer's protocol. Briefly, VSMCs at 70-80% confluence were transfected with 50 nM siRNA duplexes in 5 μL of Oligofectamine (Invitrogen, Carlsbad, CA, USA, Cat#12252011). After

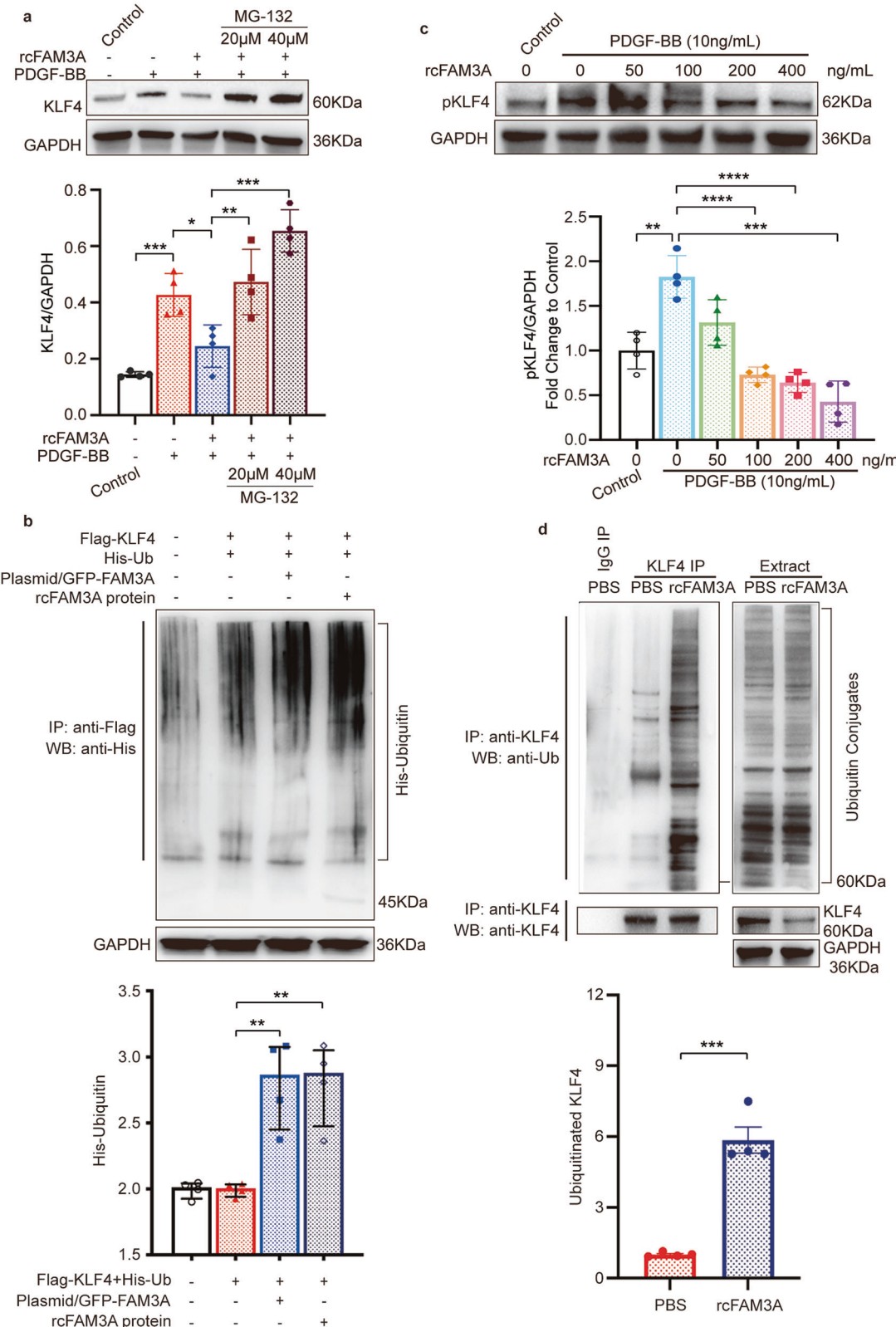

6 h, the medium was changed, and subsequent stimulation began 24 h after transfection.

## Cytological staining

At 24 h post-transfection and relative stimulation, walled HASMCs were digested using trypsin, centrifuged and resuspended, and then plated on glass-bottomed cell culture dishes (NEST, Cat#801001).

After serum deprivation overnight, cells were rinsed quickly in ice-cold PBS and fixed in 4% paraformaldehyde for 15 min at room temperature. The permeabilization was achieved by 1% Triton X-100 in PBS for 10 min and rinsed with PBS, and cells were then blocked using 1% BSA in PBS for 30 min. The primary antibodies were incubated with cells at 4 °C overnight (anti-KLF4: Proteintech, Cat#11880-1-AP, 1:50; anti-SRF: Santa Cruz, Cat#sc-25290, 1:50). Secondary antibodies (Alexa Fluor

**Fig. 6 | FAM3A influences KLF4 post-translational modification. a** VSMCs were treated with PDGF and recombinant FAM3A in the presence or absence of MG-132. Representative western blot images and quantification of KLF4 in whole cell lysates are shown ($n = 4$ biologically independent experiments). **b** The HEK293T cells were cotransfected with Flag-KLF4 and His-Ub, and then transfected with the FAM3A overexpression plasmid or treated with recombinant FAM3A protein. Whole cell lysates were prepared, and KLF4 protein was precipitated with anti-KLF4. The presence of ubiquitin-conjugated KLF4 in the immunocomplex was detected by western blot with anti-His (upper). GAPDH was detected in the same whole cell lysates (lower). Representative western blot images and quantification of KLF4 ubiquitination levels normalized to GAPHD are shown ($n = 4$ biologically independent experiments). **c** VSMCs were treated with recombinant FAM3A at the indicated concentration gradient. Representative western blot images and quantification of phosphorylated KLF4 (Ser254) in whole cell lysates are shown ($n = 4$ biologically

independent experiments). **d** VSMCs were treated with recombinant FAM3A. Whole cell lysates were prepared and KLF4 protein was precipitated with anti-KLF4. The presence of ubiquitin-conjugated KLF4 in the immunocomplex was detected by western blots with anti-Ub (left, upper). The amount of precipitated KLF4 in the immunocomplex was determined by anti-KLF4 (left, lower). The endogenous ubiquitin conjugates (right, upper), KLF4 (right, middle), and GAPDH (right, lower) were detected in the same whole cell lysates (Extract). Representative western blot images and quantification of KLF4 ubiquitination levels normalized to GAPHD are shown ($n = 4$ biologically independent experiments). Quantitative comparisons between samples were run on the same gel (**a**–**d**). Data are presented as mean ± SEM (**a**–**d**). Statistical significance was calculated with one-way ANOVA followed by Tukey post hoc test (**a**–**c**) and two-tailed independent $t$ test (**d**) and $P$ values are indicated (*$P < 0.05$, **$P \leq 0.01$, ***$P \leq 0.001$, ****$P \leq 0.0001$). Source data are provided as a Source Data file.

594-conjugated Goat anti-Rabbit IgG: Invitrogen, Cat#A-11012, 1:1000; Alexa Fluor 488-conjugated Goat anti-Mouse IgG: Invitrogen, Cat#A-11029, 1:200) were then incubated for 1 h at room temperature. Cells were mounted in a mounting medium with DAPI (Abcam, Cat#ab104139), and images were acquired with a Zeiss LSM 510 META Laser Scanning Confocal Microscope.

### Histological staining
The aortic tissue was fixed with 4% paraformaldehyde and then be dehydrated and embedded with paraffin. The paraffin-embedded tissue section was prestained and incubated at room temperature with 5% normal goat serum for 20 min. After the sealed serum incubation, the tissue section was further incubated with primary antibody (anti-αSMA: Abcam, Cat#ab7817, 1:50; anti-FAM3A (human): Sigma, Cat#HPA035268, 1:1000; anti-FAM3A (mouse): Novus, Cat#NBP3-17844, 1:100; anti-flag: Abcam, Cat#ab205606, 1:500; anti-c-myc: Abcam, Cat#ab32072, 1:100; anti-CD68: Santa Cruz, Cat#sc-20060, 1:100; anti-Aggrecan: Proteintech, Cat#13880-1-AP, 1:500; anti-RUNX2: Abcam, Cat#ab192256, 1:1000; anti-LUM: Abcam, Cat#ab252925, 1:4000; anti-CD34: Abcam, Cat#ab81289; 1:250) overnight at 4 °C in a wet box. Then, the secondary antibodies (Alexa Fluor 488-conjugated Goat anti-Rabbit IgG: Invitrogen, Cat#A-11008, 1:500; Alexa Fluor 594-conjugated Goat anti-Mouse IgG: Invitrogen, Cat#A-11032, 1:1000; Alexa Fluor 488-conjugated Goat anti-Mouse IgG: Invitrogen, Cat#A-11029, 1:200) were incubated at room temperature in the dark for 60 min and the fluorophore was combined. After the slices were slightly dried, the DAPI staining solution was added and incubated at room temperature in the dark for 10 min. The fluorescence-stained sections were then visualized using a panoramic scanning system (Pannoramic MIDI, 3D HISTECH).

Hematoxylin-eosin (HE) and Masson staining of tissue sections was routinely performed. Briefly, vessel tissues were cut into 5 μm sections on a microtome. The tissue sections were deparaffinated by immersion in xylene and rehydration, and then stained with HE or with a Trichrome Stain (Masson) Kit (#HT15-1KT, Sigma, USA) and examined with a light microscope. The inflammatory cell area and fibrotic area were quantified at a visual field (amplification × 400) with ImageJ (ver. 1.53). The values are expressed as a percentage of the total area in a visual field.

### Immunohistochemistry
The heart, liver, and kidney tissues were fixed in 10% formalin immediately after surgical resection and embedded in paraffin. The tissues were cut to 5 μm sections and then sections were incubated with 3% hydrogen peroxide to block endogenous peroxidase activity. Tissue sections were blocked with 10% BSA for 1 h and incubated with primary antibodies anti-flag (Abcam, Cat#ab205606, 1:500), anti-c-myc (Abcam, Cat#ab32072, 1:100), at 4 °C overnight. Then the HRP-conjugated secondary antibodies (goat anti-rabbit HRP: Cell

Signaling Technology, Cat#8114, being diluted by the manufacturer; goat anti-mouse HRP: Cell Signaling, Cat#8125, being diluted by the manufacturer) were added and incubated. The DAB substrate was added, and the sections were examined with a light microscopy.

### In vivo VSMC transdifferentiation analysis by flow cytometry
For analysis of VSMC transdifferentiation subsets in vivo, individual cells were isolated in aortas from mice with AAA with or without FAM3A treatment. Mice were anesthetized using an intraperitoneal injection of sodium pentobarbital (45 mg/kg). The suprarenal part of the aorta from normal mice or mice with AAA was dissected and cut on ice, and digested for 15 min at 37 °C in 1 mL of PBS containing 50 U/ml collagenase I (Sigma-Aldrich, Cat#C0130), 40 U/mL collagenase type XI (Sigma-Aldrich, Cat#C7657), 40 U/mL hyaluronidase type I-s (Sigma-Aldrich, Cat#H3506), 0.2 U/ml elastase (Sigma-Aldrich, Cat#E1250), and 0.3 U/ml deoxyribonuclease I (Sigma-Aldrich, Cat#D5025). The digestion was terminated by PBS containing 2% fetal bovine serum, and the process was repeated several times with the same amount of digestive enzymes until the tissue was completely digested. Cell suspensions were prepared by filtering through a cell strainer (No. 322350, 70 μm size; BD Bioscience, Billerica, MA). The harvested cells were first stained with PE/Cy5.5-anti-CD34 (BioLegend, Cat#119328, 1:100) and PB450-anti-CD68 (BioLegend, Cat#137017, 1:100) for 30 min and washed with PBS. Then, the cells were fixed, permeabilized, and stained with APC-anti-αSMA (Proteintech, Cat#CL647-67735, 1:100), PE-anti-RUNX2 (Cell Signaling, Cat#98059, 1:50) or LUM (Invitrogen, Cat#PA5-14570, 1:50; with PE Conjugation: Abcam, Cat#ab102918) and PE/Cy5.5-anti-ADIPQ (Abcam, Cat#ab181281, 1:50; with PerCP/Cy5.5 Conjugation: Abcam, Cat#ab102911) (different antibodies coupled by the same fluorescent group in a single flow assay were not used together, Supplementary Table 6). Flow cytometry was performed using a CytoFLEX system (Beckman Coulter, Inc., Bria, CA). The gating strategy is provided (Supplementary Fig. 11b, c). The results were analyzed using CytExpert2.4.0.28 (Beckman Coulter, Inc, Bria, CA). Various types of transdifferentiated VSMCs were defined by the simultaneous positivity of transdifferentiation tags and α-SMA.

### ROS assay
Cellular ROS levels were determined by MitoSOX™ Red mitochondrial superoxide indicator (Abmole, Cat#M19992), and the fluorescent values were quantified and normalized to the cell number. For in vivo ROS detection, the cell suspension was prepared from digested aorta tissues. A DHE (Dihydroethidium) Assay Kit/Reactive Oxygen Species (Abcam, Cat#ab236206) was used to measure the ROS level in live cells. Briefly, the resuspended cells were normalized and added to a V-bottom plate, and then, ROS staining buffer and Cell-Based Assay Buffer were added. The fluorescence intensity was measured using a

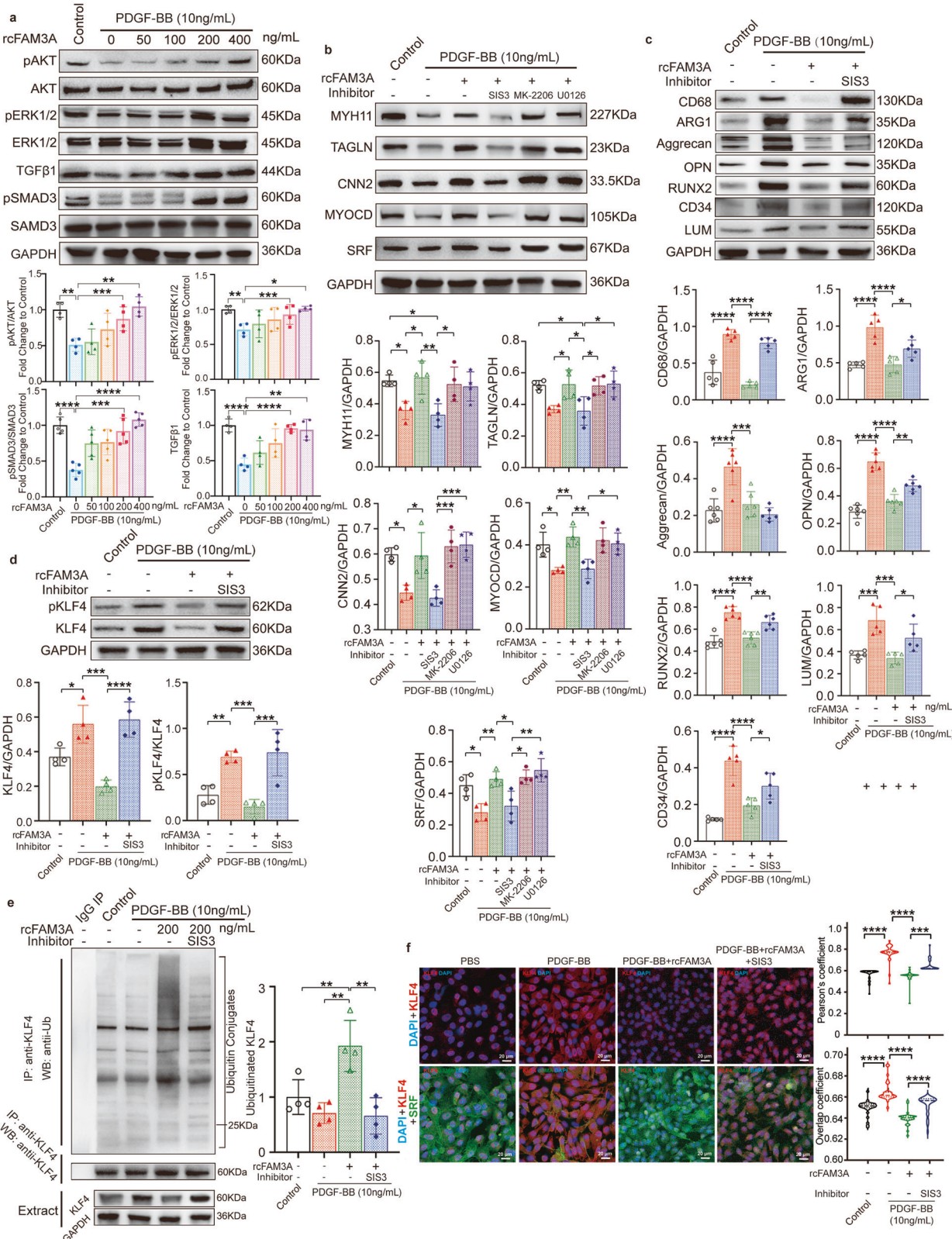

480 nm excitation wavelength and 570 nm emission wavelength with a microplate reader. The ROS production is expressed as total DHE fluorescence.

## Toxicity assays

Using colorimetric test, the heart, liver, and kidney toxicity was determined by Creatine kinase MB isoenzyme (CK-MB) Assay Kit

(E006-1-1, Nanjing Jian Cheng, China), Alanine aminotransferase (ALT) Assay Kit (C009-1-1, Nanjing Jian Cheng, China), and Creatinine (Cr) Assay Kit (C011-2-1, Nanjing Jian Cheng, China). Using ELISA test, the systemic inflammation were determined by CRP in the plasma (RAB1121, Sigma-Aldrich, USA).

Cytotoxicity was determined by monitoring cellular lactate dehydrogenase (LDH) release. Briefly, cell supernatants were collected

**Fig. 7 | Signaling pathways involved in the regulatory effect of FAM3A on VSMC differentiation reprogramming. a** VSMCs were stimulated with PDGF, and further treated with or without recombinant FAM3A at the indicated concentration gradient. The AKT, ERK1/2, and TGFβ/SMAD3 pathways in whole cell lysates were detected by western blots. Representative images and quantification are shown (*n* = 4 biologically independent experiments). **b–d** VSMCs were stimulated with PDGF and recombinant FAM3A (200 ng/ml) with or without the inhibitors of the TGFβ/SMAD3 (SIS3), AKT (MK-2206), or ERK1/2 (U0126) pathways as indicated in the charts. Representative western blot images and quantification of VSMC contractile markers (**b**, *n* = 4 biologically independent experiments), VSMC transdifferentiation markers (**c**, *n* = 5 biologically independent experiments), or phosphorylated KLF4 (Ser254) (**d**, *n* = 4 biologically independent experiments) in whole cell lysates are shown. **e** Cells were treated as in (**c**). Whole cell lysates were prepared and KLF4 protein was precipitated with anti-KLF4. The presence of ubiquitin-conjugated KLF4 in the immunocomplex was detected by western blot with anti-Ub (upper). The amount of precipitated KLF4 in the immunocomplex was determined by anti-KLF4 (middle). The KLF4 and GAPDH (lower) were detected in the same whole cell lysates (Extract). Representative western blot images and quantification of KLF4 ubiquitination levels normalized to GAPDH are shown (*n* = 4 biologically independent experiments). Quantitative comparisons between samples were run on the same gel (**a–e**). **f** VSMCs were treated as in (**c**). Confocal imaging showed the cellular localization of nuclei (blue), KLF4 (red), and SRF (green). The colocalization of nuclei and KLF4 was analyzed by ImageJ software with Pearson's correlation coefficient (PCC, upper) and Manders' colocalization coefficients (MCC, lower) quantifications (*n* = 4 biologically independent experiments with 24 scans for each group). Scale bar: 20 μm. Data are presented as mean ± SEM (**a–e**) and median ± IQR (**f**). Statistical significance was calculated with one-way ANOVA followed by Tukey post hoc test (**a–e**) and Kruskal–Wallis test followed by Nemenyi post hoc test (**f**) and *P* values are indicated (*$P < 0.05$, **$P \le 0.01$, ***$P \le 0.001$, ****$P \le 0.0001$). Source data are provided as a Source Data file.

---

and analyzed by colorimetric test with a LDH activity assay kit (C0016, Beyotime, China) according to the manufacture's recommendations.

### Colony formation assay
Briefly, the VSMCs were seeded with a low-density about 1000 cells/mL at six-well plates and normal cultured with smooth muscle cell medium containing 2% FBS, 100 U/ml of penicillin, and 100 μg/ml of streptomycin in the presence of PDGF-BB and FAM3A. Seven days later, the colony formation ratio was quantified by the percentage of the number of clones to the number of inoculated cells.

### Quantitative real-time PCR (qRT-PCR)
Total RNA was extracted from AAA tissues or cells using TRIzol reagent (Invitrogen, Cat#15596018), and cDNA was synthetized with the FastQuant RT Kit (TIANGEN Biotech, Cat#KR118) according to the manufacturer's guidelines. Quantitative real-time PCR was performed using TransStart Green qPCR SuperMix (TransGen Biotech, Cat#AQ101). Target DNA was analyzed with Bio-Rad CFX Manager software (Bio-Rad, Hercules, CA) to obtain melt curves and threshold cycle (Ct) values. The housekeeping gene GAPDH standardized the levels of all the target genes and ΔCt was quantified. All experiments were run 3× in triplicate unless otherwise mentioned. The primer information is provided in the Supplementary Table 3.

### Enzyme-linked immunosorbent assay (ELISA)
Plasma and aortic tissue IL1β, IL6, and TNFα levels in mice, plasma FAM3A in mice, and medium FAM3A in cultured VSMCs were detected by commercial ELISA kits (R&D Systems, Inc., IL1β: Cat#MLB00C, IL6: Cat#M6000B, TNFα: Cat#MTA00B; Ls bio, FAM3A (mouse): Cat#LS-F17398, FAM3A (human): Cat#LS-F35367) following the manufacturer's instructions. Briefly, plasma or aortic tissue homogenate of mice was incubated in 96-well ELISA plates precoated with capture antibody for 2 h. After a washing step with PBS-Tween 20, the detection antibody was added, followed by the addition of HRP-conjugated streptavidin and 3,3 0,5,5 0-tetramethylbenzidine. The reaction was stopped, and the absorbance was monitored at 450 nm. For the aortic tissue, the values were normalized to per gram tissue; for the medium of cultured cells, the values were normalized to the cell number.

### Western blot and immunoprecipitation assay
Protein was extracted from aortic tissues or cells in RIPA lysis buffer (Beyotime Biotechnology, Cat#P0013B, Shanghai, China), and a protease and phosphatase inhibitor cocktail (Thermo Scientific, Cat#78447). The lysates were sonicated and cleared at 13,000 × *g* for 20 min. For detection of nuclear localization of a protein, the tissue nuclear extracts were obtained using a commercial nuclear extract kit (Beyotime, Cat#P0027) according to the manufacturer's instructions.

The protein concentration was measured using a Pierce™ BCA Protein Assay Kit (Thermo-Scientific, Cat#23225). Equal amounts of protein extracts (20 - 30 μg per lane) were separated by 4–12% SDS-PAGE (Beijing LABLEAD Biotech Co.,Ltd.; Cat#P41215) and transferred to a PVDF membrane, which was subsequently probed with related antibodies (Supplementary Table 6). For the immunoprecipitation assay, cell lysate was incubated with Flag-Nanoab-Agarose (Lablead,Cat#PFA025) or protein A/G magnetic beads (MCE, Cat#HY-K0202) previously incubated with anti-KLF4 antibody (Abcam, Cat#ab106629, 1:1000) overnight at 4 °C on a rotator. After five washes with RIPA lysis buffer supplemented with a protease inhibitor mixture, complexes were released from the agarose by boiling for 5 min in 2 × SDS-PAGE loading buffer. Immunoblotting of the housekeeping protein GAPDH (1:5000, Abcam, ab181602) was performed to ensure equal protein loading. After three washes with TBST, the membranes were incubated with secondary antibodies (Goat anti-rabbit IgG HRP: Cell Signaling Technology, Cat#7074, 1:3000; Horse anti-mouse IgG HRP: Cell Signaling Technology, Cat#7076, 1:3000) for 1 h at room temperature. Immunoreactive bands were visualized with SuperSignal™ West Pico PLUS Chemiluminescent System (Pierce, Cat#34577). The unprocessed scans of the representative blots are provided within Source Data files.

### Data mining and analysis of cDNA microarray and single-cell RNA sequencing (sc-RNAseq) data in public database
The microarray expression data of AAA patients and control aortic specimens (GSE47472[67], GSE57691[68]) were aquired with the GEOquery package (version 2.66.0) in R (version 4.2.2). Differential gene expression analysis of the microarray data was conducted using the limma package (version 3.54.2) in R, and the resulting box plot was generated using the ggpubr package (version 0.5.0) in R.

To retrieve the sc-RNAseq transcript profiles in AngII-ApoE$^{-/-}$ murine AAA models and control models, the data were downloaded from the GSA database under the code PRJCA006049[14]. To retrieve the sc-RNAseq transcript profiles in CaCl2-induced murine AAA models and control models, the data were downloaded from the GEO database under the code GSE164678[15]. Downstream analysis of the sc-RNAseq data was performed using the Seurat package (version 4.3.0) in R (version 4.2.2). Cells with less than 300 or more than 7500 genes, a minimum of 100,000 mRNA counts, and at least 10% mitochondrial gene count were considered low-quality cells and were filtered out. After normalizing and selecting variable genes, the data were integrated, scaled, and subjected to principle component analysis (PCA) and uniform manifold approximation and projection (UMAP). Cell clusters were identified using a clustering algorithm based on shared nearest neighbor (SNN) modularity optimization. Marker genes for each cluster were identified using the Wilcoxon rank-sum test on normalized data. The cell type of each cluster was determined based

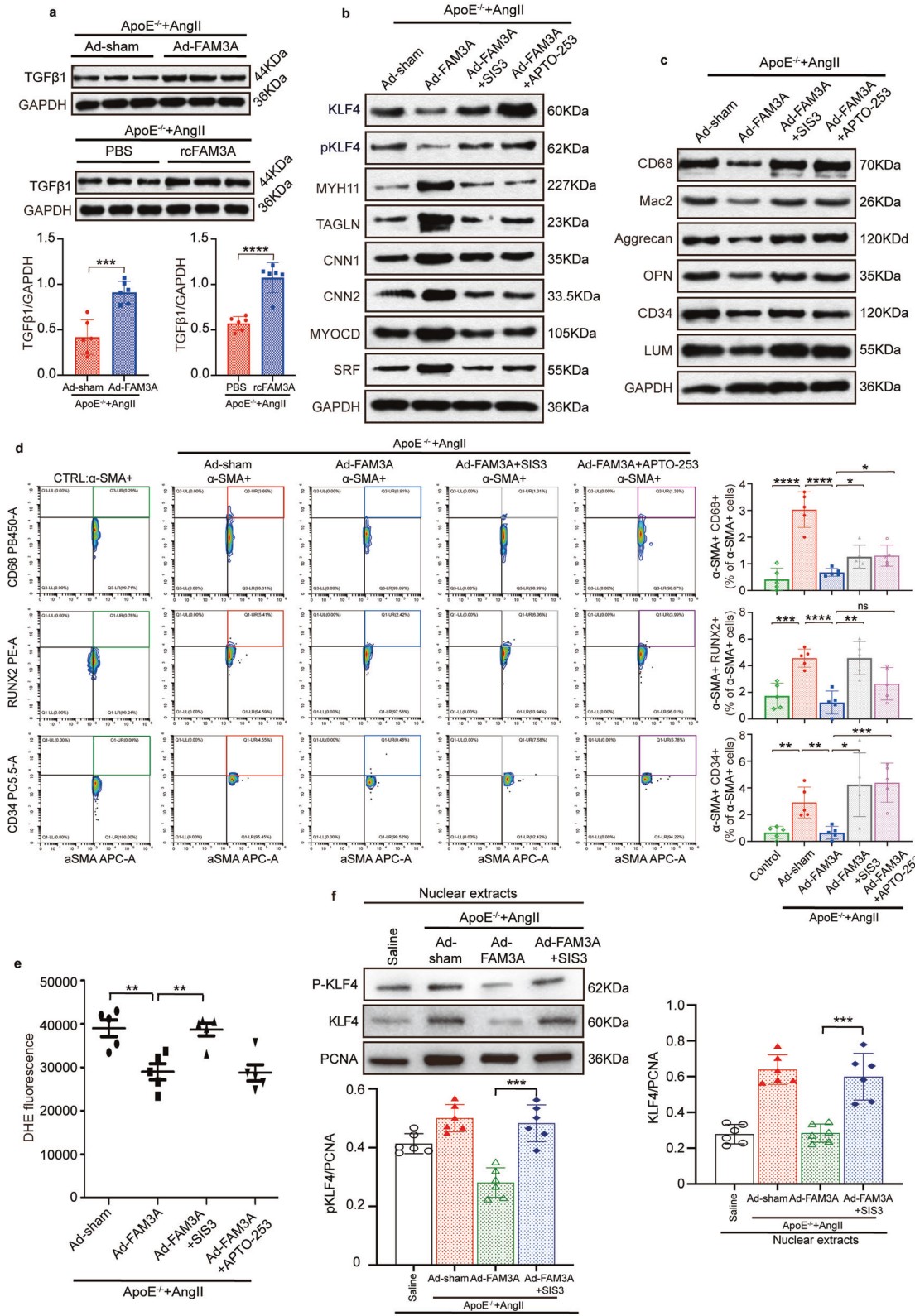

on these marker genes, and violin plots and double dot plots were generated for visualization.

## Quantification and statistical analysis

All data were described as means ± SEM for normally distributed data or median (interquartile range, IQR) for skew distributed data as indicated in every figure legend. Sample size was chosen based on literature and variability observed in previous experience in the laboratory. Survival analysis was conducted with Kaplan–Meier analysis and log-rank test. The normality and equal variance of the data were also tested by Shapiro-Wilk to determine the choice of parametric or nonparametric tests. Independent samples $t$ test was used for normally distributed data and Mann–Whitney $U$ test was performed for skew distributed data. Comparisons of nominal variates between the

**Fig. 8 | TGFβ and KLF4 signaling is involved in the regulatory effect of FAM3A on VSMC differentiation reprogramming in vivo. a** Representative western blot images and quantification of TGFβ1 in aortas from AngII-ApoE⁻/⁻ murine AAA models treated with or without FAM3A overexpression by adenovirus and treated with or without recombinant FAM3A supplementation ($n = 6$ biologically independent animals). **b** Representative western blot images of TGFβ1, KLF4 signaling, and VSMC contractile markers in aortas from AngII-ApoE⁻/⁻ murine AAA models treated with or without FAM3A overexpression by adenovirus in the presence of SIS3 or APTO-253 ($n = 5$ biologically independent animals). **c** Representative western blot images of VSMC transdifferentiation markers in aortas from mice treated as in (**b**) ($n = 5$ biologically independent animals). **d** Flow cytometry plot analysis of transdifferentiated VSMCs in aortas from ApoE⁻/⁻ control mice and AngII-ApoE⁻/⁻ murine AAA models treated with or without FAM3A overexpression by adenovirus in the presence of SIS3 or APTO-253. Graphs show the percentage of VSMCs

transdifferentiated toward the intermediate cell types of macrophages (αSMA⁺CD68⁺), osteogenic cells (αSMA⁺RUNX2⁺) and mesenchymal cells (αSMA⁺CD34⁺) ($n = 5$ biologically independent samples). **e** Quantification of ROS with the DHE method in aortas from mice treated as in (**b**) ($n = 5$ biologically independent animals). **f** Representative western blot images and quantification of KLF4 signaling in aortic nuclear extracts from saline-ApoE⁻/⁻ mice and AngII-ApoE⁻/⁻ murine AAA models treated with or without FAM3A overexpression by adenovirus in the presence of SIS3 ($n = 6$ biologically independent animals). Quantitative comparisons between samples were run on the same gel (**a**–**c**, **f**). Data are presented as mean ± SEM (**a**, **d**, **e**, **f**). Statistical significance was calculated with two-tailed independent $t$ test (**a**) and one-way ANOVA followed by Tukey post hoc test (**d**–**f**) and $P$ values are indicated (*$P < 0.05$, **$P \leq 0.01$, ***$P \leq 0.001$, ****$P \leq 0.0001$). Source data are provided as a Source Data file.

two groups were performed by the Fisher exact test. For the comparison between >2 groups with one factor, we conducted one-way ANOVA followed by Tukey post hoc test or Kruskal–Wallis test followed by Nemenyi post hoc test for normally distributed or skew distributed data, respectively. For the analysis of multiple groups and factors with normal distribution, e.g., comparing the secretion or expression of FAM3A in different cell types, two-way ANOVA with Tukey post-hoc tests were used in this study. Two-tailed $P$ values <0.05 were considered to be statistically significant. The outliers of all data were identified and removed using median absolute deviation algorithm with R software (v. 3.5.2, R-project.org). All statistical analyses were performed with IBM SPSS Statistics (v. 25; IBM Corp, Armonk, NY), and charts were developed with GraphPad Prism (v.9.0; GraphPad Software, La Jolla, CA).

### Reporting summary
Further information on research design is available in the Nature Portfolio Reporting Summary linked to this article.

## Data availability
All data supporting the findings of this study are available in the Source Data file. FASTQ files from the RNA-seq performed on freshly isolated aortas in AngII-ApoE⁻/⁻ mice with AAA described in this paper have been deposited in the Gene Expression Omnibus (GEO) database under the accession code GSE230163. The GRCm38 data in this study is available at the NIH GenBank repository website [https://www.ncbi.nlm.nih.gov/datasets/genome/GCF_000001635.20/]. Public transcript microarray datasets from AAA patients and normal control subjects were downloaded from the GEO database under the accession codes GSE47472 and GSE57691. Public single-cell RNA transcript seq data from AngII-ApoE⁻/⁻ mice with AAA and saline-ApoE⁻/⁻ control mice was downloaded from the Genome Sequence Archive (GSA) under the code PRJCA006049. Public single-cell RNA transcript seq data from Cacl2-C57BL/6 mice with AAA and saline-C57BL/6 control mice was downloaded from the GEO under the code GSE164678. Source data are provided with this paper.

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

## Acknowledgements

We wish to thank Prof. Jing Wang (Institute of Basic Medical Sciences, Chinese Academy of Medical Sciences and Peking Union Medical College) for her kind assistance with this study. Y.Z. was supported by grants from the Natural Science Foundation of China (grant number: 82070492) and the Chinese Academy of Medical Sciences, innovation Fund for Medical Sciences (grant numbers: CIFMS2021-I2M-1-016, CIFMS2021-I2M-C&T-A-006).

## Author contributions

C.L., H.K., X.X., and W.C. performed experiments and generated data. X.X. carried out bioinformatics studies. C.L. and X.X. performed sequencing. X.S. helped with some of the experiments. D.Y., Y.Z., W.X., and J.W. collected and analyzed data. C.L., X.X., and D.Y. wrote the manuscript. D.Y. and Y.Z. supervised the project and provided funding.

## Competing interests

The authors declare no competing interests.
