## [Peer Review File · Nature Communications]

FAM3A reshapes VSMC fate specification in abdominal aortic aneurysm by regulating KLF4 ubiquitinationREVIEWER COMMENTS

Reviewer #1 (Remarks to the Author):

This study by Lei et al investigate the role of FAM3A in smooth muscle cell plasticity to differentiate into other cell types during vascular remodeling in AAAs. Several issues preclude this study in its current form, as described below.

Major Concerns:

1. The overwhelming issue in this study is the lack of novelty as the role of Klf4 in SMC plasticity has been previously demonstrated (PMID: 24030402 and 35224035). Moreover, the mechanistic pathway via SMC-TGF β signaling has also been sufficiently described (PMID:32243809, 33853348, 27739498) in ascending and abdominal AAs.
2. It is unclear what the cellular sources are for FAM3A production. Does its actions involve autocrine or paracrine signaling on SMCs? If it is ubiquitously secreted, then what is the relative contribution of other relevant cell types such as endothelial cells, macrophages and neutrophils in FAM3A -dependent paracrine actions on SMC differentiation?
3. Another major drawback in this study is the lack of FAM3A-mediated ligand-receptor description. What cellular receptors does FAM3A utilize and how do they sequentially alter and what specificity do they have to FAM3A? These issues are pivotal for the mechanistic and translational significance of this study.
4. The lack of transgenic mice i.e., Klf4 $^{-/-}$ (more importantly SMC-specific Klf4 $^{-/-}$ on a cre-flox background) and SMC-Tgf- β 1/ β 2 decrease the rigor of this study. The delineation of SMC-specific molecular pathways is mainly described using in vitro studies, but the relevant crosstalk of immune cells and resident cells in vivo remains to be deciphered.
5. FAM3A has been described in the text to increase ATP synthesis as well as ROS production. Both these processes are counter intuitive to the protective actions of FAM3A as ATP acts as a DAMP and mitochondrial ROS to propagate tissue inflammation. Thus, elucidation of FAM3A in a sequential pattern in the experimental models, in the plasma and aortic tissue, may offer better clues to the relative pro- vs anti-inflammatory signaling initiated by FAM3.
6. Several inhibitors described in this study i.e., MK2206, U0126 and SIS3 were used at 10mM. These are super physiological doses and dose-dependent studies should be described.
7. Please demonstrate if overexpressing Fam3A in mice specifically increases its expression in SMCs or other cells in vivo.
8. The transcriptional expression of Klf4 (Fig. 2H) differs from protein expression (Fig. 5B-C). Apart from attributing this to post-translational modifications, could this be a time-dependent regulated phenomenon? Sequential analyses of day 3, 7 and 14 in the elastase-model and day 14, 21 and 28 in the ApoE $^{-/-}$ models would be prudent to address this relevance.
9. Kindly provide the entire aortic section of representative images in the histology sections for a complete evaluation of spatial expression in aortic layers. Also, quantification of aortic sections for histological images should be provided.
10. Cytokine expression in aortic tissue would be reflective of the local inflammatory milieu and should be provided in addition to plasma expressions.
11. Apart from Klf4, did the authors analyze Klf2 or Zfp148 (PMID: 31025534) that could also regulate SMC-specific genes and chromatin remodeling?

Minor:

Grammatical phrases (i.e. popped, propose a clue, pity etc.) should be reworded in the manuscript at certain places in the text for the scientific audience.

Reviewer #2 (Remarks to the Author):

Chuxiang Lei et al. have investigated the role of FAM3A in vascular smooth muscle cell (SMC) differentiation and reprogramming in the context of abdominal aortic aneurysms (AAAs). They detected decreased levels of FAM3A in human and mouse AAA tissue. Increasing FAM3A levels systemically by adenoviral expression vector or by administration of human recombinant FAM3A (hrFAM3A) significantly reduced AAA formation in two mouse models and decreased (rupture-related) animal death. Gene expression analysis of aortic tissue from treated versus untreated mice revealed various regulated, primarily metabolism-related pathways. In addition to this unbiased approach (which was not further pursued) the authors chose to investigate target genes of SMC differentiation and found a significant impact of FAM3A overexpression/administration in preserving factors of the contractile cell phenotype. Effects were confirmed at the protein level, and further included a decrease in inflammatory and proteolytic regulators of media destruction in the AAA mouse models. In vitro experiments with isolated human aortic smooth muscle cells (HASMCs) documented a dose-dependent, protective effect of hrFAM3A at concentrations ≥ 200 ng/ml in preventing PDGF-BB (or cholesterol) triggered loss of SMC contractile phenotype and differentiation into other cell lineages. Reversal of these effects by the selective SMAD3 inhibitor SIS3 indicated a mechanistic link to TGF-beta signaling, and further downstream effects on phosphorylation and ubiquitination/degradation of KLF-4 were revealed. The authors conclude that "FAM3A maintains a well-differentiation status of VSMCs, at least in part, by a marked activation of TGF β signaling and thereby a suppression of KLF4 action. These findings offer a distinct perspective to develop further therapeutic strategies for AAA aiming at restoring VSMCs homeostasis."

This is a comprehensive study which covers human AAA tissue, intervention approaches in two distinct mouse models of AAA and in vitro analysis of cellular mechanisms. Overall, the manuscript is well composed and describes the experiments in sufficient detail – minor language/grammar corrections may be required. Sample size and experimental replicates seem sufficient, i.e. statistical tests identify significant effects. Comments, points of critique and suggestions for changes are the following:

ORIGINALITY AND SIGNIFICANCE TO THE FIELD

What is already known:

- FAM3A is a ubiquitously expressed cytokine-like protein which promotes mitochondrial ATP synthesis (Yan et al., 2022) and is involved in various pathological conditions including type II diabetes or NAFLD (Zhang et al., 2018).
- FAM3A is required for muscle stem cell commitment and drives muscle stem cell oxidative metabolism and differentiation (Sala et al., 2019).
- KLF-4 is a central regulator of SMC phenotypic modulation (Deaton et al., 2009).
- In aortic aneurysms, reduced TGF-beta signaling via SMAD2/3 increases KLF-4 protein and activity, which results in loss of the contractile SMC phenotype and cell reprogramming/ differentiation into other cell lineages (Chen et al., 2020).

What this study adds:

- In vitro experiments with isolated human aortic smooth muscle cells (HASMCs) document a role of FAM3A in preserving the contractile, differentiated SMC phenotype via TGF-beta signaling in HASMCs and subsequent suppression of KLF-4 action. In vitro doses ≥ 200 ng/ml of human recombinant FAM3A (hrFAM3A) are required.
- Reduced FAM3A expression is detected in human and mouse abdominal aortic aneurysms (AAAs); intervention by systemically increasing FAM3A levels in mouse models interferes with AAA development.

What is not resolved/addressed by this study:

- Is systemic FAM3A administration preventing aneurysm formation via a direct effect on SMC differentiation or via an indirect consequence of FAM3A activity on other cells/pathogenic processes in AAA? Do the in vivo achieved FAM3A doses match the in vitro required concentrations to affect SMC differentiation?

- Which are the main producers of FAM3A in the AAA context? Is FAM3A acting primarily autocrine or paracrine and via which receptors do SMCs receive FAM3A signals?
- How does FAM3A activity in HASMCs regulate TGF-beta signaling in these cells?
- Would systemic FAM3A therapy also work to prevent progression of established disease in the murine AAA models (which corresponds to the clinical situation)?

CONCLUSIONS AND CLAIMS

1. In human healthy aorta (Fig. 1C) FAM3A seems to be predominantly expressed in the adventitia rather than in the media, where FAM3A staining is mainly found closer towards the intima. The authors fail to comment on this circumstance or how this would fit their theory of FAM3A loss in the media/SMCs during AAA development.
2. Systemic administration of hrFAM3A (or FAM3A adenovirus) is likely to affect/infect various cell types: hrFAM3A will more effectively target ECs than SMCs, FAM3A adenovirus will predominantly end up in the liver. The authors did not prove that in vivo effects are mediated directly by SMCs, i.e. the positive impact on SMC differentiation might be an indirect effect based on the responses by other cell types. In line, RNAseq analysis of AAA tissue from Ang-II mice with FAM3A versus control adenovirus revealed a number of regulated pathways which were predominantly not SMC-specific (e.g. cytokine signaling). Since the adenovirally expressed FAM3A carries a FLAG tag, the authors may try to follow protein expression by immunofluorescence staining of aneurysms and other organs. It would also be good to document if additional effects (potential side effects in therapeutic application) were observed upon systemic FAM3A administration - on cells/organs other than SMCs and the aorta? Ultimately, only SMC-specific overexpression of FAM3A would answer the question whether the protective effect against AAA development is indeed mediated by the direct action on SMCs.
3. Concentrations of ≥ 200 ng/ml hrFAM3A were required to rescue the SMC phenotype in vitro. In vivo, weekly administration of 200 ng hrFAM3A per mouse is likely to result in circulating lower levels and even lower levels reaching the SMC compartment. How do the authors reconcile this discrepancy? When administering hrFAM3A i.v. at 1-week intervals it would be of interest to also monitor plasma/serum levels of FAM3A for the achieved blood concentrations and kinetics. Furthermore, while human and murine FAM3A show a high degree of homology, have the authors verified that hrFAM3A is affecting murine SMCs at comparable doses which were effective in in vitro experiments with human aortic SMCs?
4. The claim that "...these results suggested a potential therapeutic strategy by using a supplement of FAM3A in AAA." seems overstated to me due to the lack of
 - a. data on non-SMC effects of systemic FAM3A administration which might have adverse impact/side effects
 - b. mouse studies with FAM3A administration on established AAA disease – assessing inhibition of progression (as would be the case for the clinical situation).
5. Evidence is provided that FAM3A regulates SMC trans-differentiation into other cell types (Fig. 4 and Suppl. Fig. IV-B). While the in vitro data are convincing – again showing regulation by FAM3A at doses ≥ 200 ng/ml, the in vivo data are based on Western blot analysis of cell lineage markers for aorta tissue extracts. This does not provide actual proof for SMC trans-differentiation, since e.g. a lower level of CD68 expression could simply be based on a lower accumulation of macrophages in the aorta. AAA tissue stainings from mouse aortas would be of interest.
6. Supplementary Figure V illustrates that KLF-4 inhibition (or silencing) can rescue the effect of FAM3A silencing with respect to loss of contractile SMC markers. The same experimental setup would be expected for supplementary Figure VI where the authors address SMC trans-differentiation into other cell types. However, the authors switched to SMC stimulation by PDGF-BB and demonstrate that KLF-4 silencing can prevent PDGF-BB induced trans-differentiation (without link to FAM3A activity)?

DATA AND METHODOLOGY

7. Human samples are restricted to 6 AAA patients and 6 healthy controls who appear well matched (except for hyperlipidemia – and age: 66 vs 55 years). It would be of interest to add the information,

whether the AAA or healthy aortas were affected by atherosclerosis.

8. Source/cloning of the adenoviral vector (for FAM3A overexpression) is not specified.

9. Immunostaining of human and mouse aorta cryosections for SMA and FAM3A is shown but not explained in the Methods section (information is restricted to in vitro HASMCs cell cultures).

10. RNA sequencing data are shown but the sample preparation and analysis is not explained in the methods section.

11. The authors present in vitro experiments with isolated HASMCs where they show dose-dependent effects of hrFAM3A on the SMC phenotype (Suppl. Fig. II). All experiments were conducted in the presence of PDGF-BB or cholesterol, but the authors fail to mention or present controls without PDGF-BB or cholesterol addition.

12. The confocal microscopy images provided in Fig. 5F and 7F are not very convincing in demonstrating the effect of FAM3A reducing KLF4 nuclear localization. Similarly, fluorescent images in Suppl. Fig. V-A don't support the proposed ROS increase upon FAM3A silencing. Yet, quantitations with highly significant differences are displayed.

MINOR ISSUES

13. Supplementary Table I: Please specify whether median/mean and SD or SE are given for years of age.

14. Methods/Cell Culture: "100 g/ml of streptomycin" should likely read "100 µg/ml of streptomycin".

15. Methods/qRT-PCR: What is meant by "takeoff values"? The threshold cycle (Ct) values?

16. Legend to Figure 2: Adenoviral gene transfer should be more clearly mentioned.

17. Legend to Figure 3E: While the y-axis refers to serum levels, the legend specifies that plasma was analyzed – please correct.

18. Legend to Figure 5: "subjected to adenoviral FAM3A overexpression (A) and recombinant FAM3A supplement (B)" should read "subjected to adenoviral FAM3A overexpression (B) and recombinant FAM3A supplement (C)". Please also indicate the difference between the two quantification plots (for co-localization) in 5F.

19. Legend to Figure 7: Explanation of Figure part 7F is missing.

REFERENCE TO PREVIOUS LITERATURE

20. While the authors cite essential previous studies, they occasionally fail to address discrepancies or relate their results to these studies, e.g. Jia et al., 2014, Xiang et al., 2020: Xiang R. et al. reported FAM3A effects on angiotensin-II driven hypertension which has not been observed in the present study.

CITATIONS

CHEN, P. Y. et al. 2020. Smooth Muscle Cell Reprogramming in Aortic Aneurysms. *Cell Stem Cell*, 26, 542-557 e11.

DEATON, R. A. et al. 2009. Sp1-dependent activation of KLF4 is required for PDGF-BB-induced phenotypic modulation of smooth muscle. *Am J Physiol Heart Circ Physiol*, 296, H1027-37.

JIA, S. et al. 2014. FAM3A promotes vascular smooth muscle cell proliferation and migration and exacerbates neointima formation in rat artery after balloon injury. *J Mol Cell Cardiol*, 74, 173-82.

SALA, D. et al. 2019. The Stat3-Fam3a axis promotes muscle stem cell myogenic lineage progression by inducing mitochondrial respiration. *Nat Commun*, 10, 1796.

XIANG, R. et al. 2020. VSMC-Specific Deletion of FAM3A Attenuated Ang II-Promoted Hypertension and Cardiovascular Hypertrophy. *Circ Res*, 126, 1746-1759.

YAN, H. et al. 2022. Intracellular ATP Signaling Contributes to FAM3A-Induced PDX1 Upregulation in Pancreatic Beta Cells. *Exp Clin Endocrinol Diabetes*, 130, 498-508.

ZHANG, X. et al. 2018. FAM3 gene family: A promising therapeutic target for NAFLD and type 2

diabetes. *Metabolism*, 81, 71-82.

Reviewer #3 (Remarks to the Author):

Lei C et al. investigated the role of FAM3A as a regulator of differentiation of vascular smooth muscle cells (VSMC) in the pathogenesis of abdominal aortic aneurysm (AAA). In human AAA samples as well as in two experimental models of AAA (Pancreatic elastase-induced AAA and ApoE^{-/-} transgenic mice treated with ANGII), they found a downregulation of FAM3A. Overexpression or supplement of FAM3A prevented in part the dilation of the aorta and improved the survival, but FAM3A did not protect from functional changes (increased blood pressure). Further mechanistic studies demonstrate that FAM3A increases the expression of contractile elements in VSMC and reduces the proliferation of VSMC. Next, they propose that these effects by FAM3A on VSMC function are mediated through an activation of TGFbeta/SMAD3 signaling with subsequent suppression of Klf4 phosphorylation and elevation of Klf4 ubiquitination.

The study includes a comprehensive set of experiments such as human samples, two experimental models of AAA, in vivo and in vitro overexpression of FAM3A, in vivo treatment with FAM3A and transcriptomic analysis. While the data provide new mechanistic insights in the pathogenesis of AAA and a possible therapeutic approach, there are several aspects that should be addressed.

General comments:

1) In both experimental models, the authors demonstrate a reduction of the diameter of the aorta and reduced inflammation (e.g., IL-6), but no differences in function (blood pressure).

a. What is the cause of the discrepancy between structure and function?

b. The authors also propose that "the degree of elastic fiber disintegration was significantly lower" after FAM3A treatment. In line, with this notion MMP2 and MMP9 proteins were decreased in ApoE^{-/-} +ANGII treated with FAM3A when compared to those treated with vehicle. Was this change in proteolytic activity reflected in changes in elastic fibers or collagen? Was the effect of FAM3A on protease expression also detectable in the other models (elastase-induced AAA; adenoviral FAM3A overexpression)?

c. How were elastic fibers studied? H&E staining is not the ideal technique to quantify elastic fibers? What might be the impact of matrix stiffness (elastic fibers, collagens) or increased contractility of VSMC on the increased blood pressure after supplement/overexpression of FAM3A despite reduced diameter of the aorta.

d. Does FAM3A affect fibroblast function?

These aspects should be elaborated further.

2) The authors focused primarily on the differentiation of VSMC. Since SMAD3 and Klf4 are also regulators of cell survival and proposed by the authors to be downstream of FAM3A, it is of interest to assess the impact of FAM3A on survival of VSMC. This is also of importance based on the finding that FAM3A regulates the pro-proliferative AKT signaling.

3) The authors propose that FAM3A reduces Klf4 phosphorylation and increases Klf4 ubiquitination through TGFbeta and SMAD3. However, the link between Smad3 and Klf4 has not been studied. Does blocking of TGFbeta signaling (SMAD2/3) prevent the modulation of Klf4 and the differentiation of VSMC? The FAM3A-SMAD3-Klf4 axis requires further exploration.

4) Klf4 phosphorylation/ubiquitination in animal models of AAA with and without Klf4 overexpression/supplement has not been studied in the present manuscript. Is the cytoplasmic/nuclear localization altered in VSMC by FAM3A in vivo.

5) The authors propose that FAM3A suppresses ROS. Is this evident in the experimental models of AAA after overexpression/supplement of FAM3A? How is ROS production related to TGFbeta/SMAD3-Klf4 axis? Can another technique in addition to immunofluorescence be used to assess ROS?

6) It is unclear why the authors have chosen Klf4 and not another transcription factor from the

transcriptome? Is there a rationale beyond published literature on the functional role of Klf4 in VSMC in AAA?

Specific comments:

7. Figure 1A: typo: it is "Masson" not "Mason"

8. Figure 1E: Please include the proper labeling of the Y-axis. Does it start at 0? It only shows 120.

9. Figure 1G: The images of the aorta show less dilated aorta in elastase when compared to saline-treated aorta. This is not consistent with the quantification.

10. Figure 4B/Suppl. Figure IV:

Are the markers of VSMC differentiation, e.g., CD68, ARG1, Aggrecan, regulated in the experimental models of AAA at baseline (without treatment).

12. Results, Line 332: "It is well accepted that Klf4 is a novel reprogramming factor". The references are missing.

13. Figure 5: Klf4 protein abundance has been measured in the experimental models and in human AAA samples. However, Klf4 nuclear localization and phosphorylation was only assessed in vitro. Is there evidence of Klf4 nuclear-cytoplasmic shift in the in vivo models? For in vivo and/or in vitro, nuclear and cytoplasmic extracts, followed by immunoblot for Klf4 would strengthen the hypothesis that FAM3A regulates nuclear-cytoplasmic of Klf4 in VSMC.

14. In the summary (Figure 8), the term "synthetic VSMC" needs further explanation. I am uncertain what the authors mean.

15. In material and method section:

- Does the animal approval number JS-2629 apply for all animal studies in the manuscript or only to the Pancreatic elastase-induced AAA Model?

- Where are the ApoE transgenic mice obtained from?

16. Were only male mice studied in the project? If so, what is the reason? Are there sex specific differences in the AAA model?

Reviewer #1 (Remarks to the Author):

This study by Lei et al investigate the role of FAM3A in smooth muscle cell plasticity to differentiate into other cell types during vascular remodeling in AAAs. Several issues preclude this study in its current form, as described below.

Response in general: Thanks very much for your comments. Firstly, I wish to express my gratefulness to you for a careful review of our paper. We are very glad to discuss with you concerning our study. Actually, your comments have enlightened us and we make a series of additional experiments to improve our study. Let us first introduce you the research background and workflow in general.

Firstly, we discovered by accident that FAM3A expression was suppressed in AAA patients and mice. We therefore questioned whether replenishment of FAM3A could alleviate AAA formation. Since FAM3A is a cytokine and its receptor has not been identified, we used two ways of administration to obtain a comprehensive knowledge of FAM3A efficacy in AAA. These were administrations of recombinant FAM3A (mainly through endocrine manner maybe) and adenovirus mediated FAM3A overexpression (including autocrine, paracrine, and endocrine maybe). Each of the way of administration has its own characteristics.

However this time we focused on VSMCs to assess the FAM3A function on AAA, due to the important roles of VSMCs in AAA as well as our finding of FAM3A/VSMC plasticity linkage. This does not preclude that other cell types did not involved in the function of FAM3A on AAA disease.

We really feel sorry that some studies concerning the function and mechanism of FAM3A targeting VSMC in vivo is hard to explore currently, due to unsolved FAM3A receptor and deficiency of VSMCs-targeted mediators or vectors.

According to your nice suggestions, we have made extensive corrections to our previous draft. We have added necessary data to supplement our results and edited our article extensively. Please review the detailed responses which are listed point by point below.

Major Concerns:

1. The overwhelming issue in this study is the lack of novelty as the role of Klf4 in SMC plasticity has been previously demonstrated (PMID: 24030402 and 35224035). Moreover, the mechanistic

pathway via SMC-TGF β signaling has also been sufficiently described (PMID:32243809, 33853348, 27739498) in ascending and abdominal AAs.

Response: Thanks very much for your comments. Actually, TGF β and KLF4 signaling have been evidenced to regulate VSMC plasticity in abdominal aneurysm. The publications of PMIDs 24030402 (Reference 16), 35224035 (Reference 17) have been cited and discussed in the revised manuscript in the Results section “*FAM3A influences KLF4 post-translational modification*” paragraph1, line2; and in the Discussion section paragraph5, line5. The publications of PMIDs 32243809 (Reference 23), 33853348 (Reference 24) and 27739498 (Reference 25) have been cited and discussed in the revised manuscript in the Results section “*The signaling pathways involved in regulatory effect of FAM3A on VSMC differentiation reprogramming*” paragraph1, line4; and in the Discussion section paragraph6, line7.

In our study, the novelty is the findings of correlations between FAM3A and AAA. We report 1) a decreased FAM3A expression in aneurysm tissues and plasma from AAA patients and mice; 2) the role of FAM3A in regulated VSMC plasticity (contractile phenotype and trans-differentiation towards other intermediate cell types); 3) the effect of FAM3A on KLF4 phosphorylation, ubiquitination, and nuclear localization.

In the revised paper, we strengthened the functions of FAM3A in AAA as well as SMCs for a more in-depth study. Described as below:

- Further showing plasma FAM3A level in AAA patients and mice beyond AAA tissues. Please review Supplementary Fig. 1a, b.
- Detecting AAA-correlated FAM3A changes in a cell-specific view beyond only gross expression change in AAA tissues in our first submission. Please review Supplementary Fig. 1c-e and Results section “*Cell-specific FAM3A expression changes in AAA microenvironment*”.
- Exploring the potential **producer** (cells which produce FAM3A, Supplementary Fig. 1c,d) and **responder** (cells which have responsiveness to FAM3A, or potential cells which have receptor to bind with FAM3A, Supplementary Fig. 1e and Supplementary Fig. 4c).
- Detecting exogenous FAM3A cell-specific distribution in different cells in AAA tissue and tissue distribution in heart, liver, and kidney after systemic administration of recombinant FAM3A or FAM3A adenovirus. Please review Supplementary Fig. 4a-d.

- Confirming the VSMC transdifferentiation in vivo using FACS by flow cytometry and tissue immunofluorescence co-staining of VSMC and other intermediate cell markers. Please review Figs. 4b,c and 8d.
- Detecting the effect of FAM3A on TGF β and KLF4 total expression, phosphorylation, and nuclear localization in vivo beyond just in vitro in our first submission. Please review Figs. 5c and 8a,b,f.
- Using TGF β /SMAD3 inhibitor SIS3 and KLF4 inducer APO-253 in vivo, we detected the in vivo disturbance of TGF β /SMAD3 and KLF4 signaling in the setting of FAM3A functions on VSMC plasticity, in addition to in vitro experiments. Please review Fig. 8. We hope these new data integrated into the revised paper will display more scientific importance.

2. It is unclear what the cellular sources are for FAM3A production. Does its actions involve autocrine or paracrine signaling on SMCs? If it is ubiquitously secreted, then what's the relative contribution of other relevant cell types such as endothelial cells, macrophages and neutrophils in FAM3A-dependent paracrine actions on SMC differentiation?

Response: Thanks for your review. You mentioned a crucial issue. It concerns FAM3A **producer** (which cell express or produce FAM3A) and the manners of autocrine or paracrine, as well as endocrine to promote SMCs in AAA microenvironment.

1) Concerning cellular sources and autocrine or paracrine signaling on SMCs

In the interstitial space, FAM3A produced and secreted by VSMCs (autocrine) or other cells such as endothelial cells, macrophages, fibroblasts, and T cells (paracrine), could transfer and attach to VSMCs and regulate their plasticity in AAA microenvironment. Therefore, to explore cellular sources about FAM3A production in AAA microenvironment is important. To detecting in vivo FAM3A cellular sources in AAA tissues:

a) Immunofluorescence in histology section is unfeasible: Since FAM3A is a cytokine, it is hard to determine a FAM3A which co-localized with a cell (such as VSMC) is produced by VSMC per se or secreted by other cells and transferred to VSMC membrane by immunofluorescence co-staining of FAM3A and a marker specific to one kind of cell using histology section.

b) Fluorescence-activated cell sorting (FACS) technique by flow cytometry or magnetic-activated cell sorting (MACS) is theoretically feasible and practically unfeasible: In the AAA tissue, there are too few living cells (about $4-5 \times 10^4$ cells in an AAA tissue) to

obtain enough amount of protein or mRNA to do western blot or realtime PCR (about 1×10^6 necessarily). So, it is hard to sort several main cell types by FACS, including VSMCs, macrophages, fibroblasts, endothelial cells, and T cells in AAA microenvironment to determine FAM3A cell-specific source (as a high throughput method, single-cell RNA transcript sequencing is not discussed here).

Exploring whether or not a cell could produce FAM3A mRNA is a direct way to determine FAM3A cellular source in AAA microenvironment. Therefore, we used single-cell RNA transcript sequencing data from murine AAA models to make an analysis.

Solutions:

- Please review Supplementary Fig. 1c: Using single-cell mRNA sequencing data from murine AAA tissue or control (AngII-ApoE^{-/-} and Saline-ApoE^{-/-}, PMID:34901214, Genome Sequence Archive (GSA) under the code PRJCA006049, this publication was cited as Reference 14 in our paper), we analyzed cell-specific FAM3A mRNA expressions in AAA and normal aorta.
- Please review Supplementary Fig. 1d: We detected cell-specific FAM3A protein secretion level using cultured primary cells in situations of physiology and pathology (choosing common inflammatory cytokines in AAA microenvironment).

These results suggest that FAM3A may be expressed and secreted abundantly in VSMCs and endothelial cells, and with a few amount in fibroblasts and macrophages. It seemed that endothelial cells, fibroblasts and macrophages could secrete FAM3A to influence VSMC plasticity in a paracrine manner. And VSMCs can secrete FAM3A.

Additionally, the endocrine may be another manner of FAM3A regulating VSMC plasticity, due to the presence of FAM3A in blood circulation (Supplementary Fig. 1a,b).

2) Concerning the relative contribution of other relevant cell types such as endothelial cells, macrophages and neutrophils in FAM3A-dependent paracrine actions/cell-cell crosstalks on SMC differentiation?

We thought that FAM3A regulated VSMCs in four manners: ① endocrine FAM3A from circulating sources (direct function of FAM3A); ② paracrine FAM3A from other cells in AAA microenvironment (direct function of FAM3A); ③ autocrine FAM3A by VSMC itself (direct function of FAM3A); ④ other cells which were phenotype-resaped by FAM3A, interacted and influenced VSMCs by cell-cell crosstalk in AAA microenvironment (indirect function of FAM3A). We illustrate our meaning as below:

In our study, we observed a decreased level of endogenous FAM3A in AAA disease. We speculated that the weakened actions of endogenous FAM3A were involved in AAA pathology.

In our study, the in vivo data evidenced that systematic administration of exogenous FAM3A could inhibit AAA formation. We think that these functions of FAM3A in vivo may be a combination of direct and indirect functions of FAM3A. The FAM3A-mediated cell-cell crosstalk also contributed in some degree to reshaping of VSMCs.

Solutions:

- We make a discussion about these key points concerning FAM3A cellular production and function manners in VSMCs. Please review Discussion section, paragraph3, paragraph7.

3. Another major drawback in this study is the lack of FAM3A-mediated ligand-receptor description. What cellular receptors does FAM3A utilize and how do they sequentially alter and what specificity do they have to FAM3A? These issues are pivotal for the mechanistic and translational significance of this study.

Response: Thanks very much for your comments. You mentioned an important issue. However it is a limitation that the receptor of FAM3A was not given in the present study. Actually, most receptors of the FAM3 family members have not been identified currently, including FAM3A. Recently, one of the FAM3 family member, FAM3B, its receptor was identified (that is FGFR) (*FAM3B receptor FGFR, Proc Natl Acad Sci U S A. 2021 May 18;118(20):e2100342118. doi: 10.1073/pnas.2100342118. PMID: 33975953*).

We are working to screening FAM3A receptor by using Mass-Spectra and Yeast-Two-Hybrid methods, and we also inhibit some common receptors (such as FGFR, insulin receptor, the ligand of which show a slight clue of similar actions with FAM3A) to detect whether functions of FAM3A could be inhibited (Please see picture below). However no significant findings were obtained at present.

Solutions:

- Please review Supplementary Fig. 1e: We explored the affinities between different cells and FAM3A to provide some indirect clues concerning FAM3A receptor. These indirect results showed that different cell types in AAA microenvironment have differential affinity for FAM3A in our additional study by using recombinant FAM3A which has a c-myc label facilitating the detecting.
- Please review Supplementary Fig. 4c: We detected the co-localization between c-myc-FAM3A and VSMCs, endothelial cells, fibroblasts, or macrophages respectively, using immunofluorescence co-staining. It seemed that the VSMCs as well as endothelial cells had stronger affinities to FAM3A.
- Please review Discussion section for line19 in paragraph2, line8 in paragraph4, and line6 in paragraph7: We stated and discussed the lack of FAM3A receptor in the revised paper.

4. The lack of transgenic mice i.e., *Klf4*^{-/-} (more importantly SMC-specific *Klf4*^{-/-} on a cre-flox background) and SMC-Tgf-β1/β2 decrease the rigor of this study. The delineation of

SMC-specific molecular pathways is mainly described using in vitro studies, but the relevant crosstalk of immune cells and resident cells in vivo remains to be deciphered.

Response: Thanks very much for your comments.

1) Concerning the lack of transgenic mice i.e., Klf4^{-/-} or Tgfβ1/β2^{-/-}

It is limited really that we have no gene-deficiency mice of Klf4 and Tgfβ1/β2 currently. As you can see, the in vivo promotion or inhibition of KLF4 and TGFβ is important to determine FAM3A-mediated mechanistic pathway. Therefore, we used commercial TGFβ inhibitor SIS3 (PMID: 28262747, cited as Reference 61) and KLF4 inducer APTO-253 (PMIDs: 33139312, 34568499, 33918002, cited as Reference 62-64, respectively) to detect their roles in FAM3A-mediated function in vivo. Briefly, the mice were injected with TGFβ/smad3 inhibitor (SIS3) or KLF4 inducer (APTO-253) before FAM3A adenovirus injection, and then whether FAM3A maintained regulatory functions on VSMC plasticity was explored.

Solutions:

- Please review Fig. 8b, c, d, f: The VSMC contractile and transdifferentiation markers, and KLF4 nuclear localization were determined in vivo in context of FAM3A actions with or without the TGFβ inhibitor (SIS3) or KLF4 inducer (APTO-253).

2) Concerning that SMC-specific molecular pathways is mainly described using in vitro studies

Really, SMC-specific Klf4 and SMC-specific Tgf-β1/β2 transgene or deficiency mice are useful to analyze the mechanism of FAM3A actions on VSMC in vivo. We feel sorry that we have no VSMC-specific gene-deficiency or transgenic mice of Klf4 and Tgfβ1/β2 currently.

Solutions:

- In the revised paper, we made a discussion about this limitation about cell-specific research. Please review Discussion section, the second paragraph to last.
- Please review Fig. 8d: We administrated AAA mice with the TGFβ/SMAD3 inhibitor (SIS3) or KLF4 inducer (APTO-253), and then targeted VSMC in vivo (using VSMC specific marker αSMA) to observe VSMC transdifferentiation status by FACS with flow cytometry using AAA tissues, providing a slight clue about SMC-specific mechanisms in vivo in some degree.

5. FAM3A has been described in the text to increase ATP synthesis as well as ROS production.

Both these processes are counter intuitive to the protective actions of FAM3A as ATP acts as a DAMP and mitochondrial ROS to propagate tissue inflammation. Thus, elucidation of FAM3A in a sequential pattern in the experimental models, in the plasma and aortic tissue, may offer better clues to the relative pro- vs anti-inflammatory signaling initiated by FAM3.

Response: Thanks very much for your reviews. In our findings, ROS production and inflammation was suppressed by FAM3A in the context of AAA.

Solutions:

- Please review Supplementary Fig. 5c for ROS change in vitro and Fig. 8e in vivo: We detected ROS production change induced by FAM3A. As shown, FAM3A inhibited ROS production in vitro and in vivo, offering some clue to the anti-inflammatory function of FAM3A under pathological conditions.
- Please review supplementary Figs. 2c and 3e, and supplementary Fig. 2c for inflammatory mediator detection in plasma and supplementary Fig. 2e for that in AAA tissues: In the revised paper, we provided data about plasma inflammatory cytokine level in murine AAA models (AngII-ApoE^{-/-} AAA mice model with FAM3A overexpression by Adenoviral injection or recombinant FAM3A; C57BL/6-Elastase mice model with FAM3A overexpression by Adenoviral injection), and did additional experiment to analyze inflammatory cytokine level in local AAA microenvironment using AngII-ApoE^{-/-} and Elastase-C57 AAA mice models and Ad-FAM3A or recombinant FAM3A administration method.

6. Several inhibitors described in this study i.e., MK2206, U0126 and SIS3 were used at 10mM. These are super physiological doses and dose-dependent studies should be described.

Response: Thank you very much for your reviews. The 10mM is a storage concentration, and the storage solution was diluted at 1:1000 as a working solution for the experimental conduction.

Solutions:

- Please review: In the revised paper, the pathway inhibitor concentration was described more clearly in the Methods section “*Cell culture and stimulation*”, line10-12.

7. Please demonstrate if overexpressing Fam3A in mice specifically increases its expression in SMCs or other cells in vivo.

Response: Thank you very much for your suggestion. A flag tag protein is fused in the adenovirus-FAM3A we used. In the revised paper, we have detected the flag tag to monitoring the FAM3A overexpression mediated by adenovirus injection.

Solutions:

- Please review Supplementary Fig. 4a, b: FAM3A-flag was located in VSMC, macrophage, endothelial cell, and fibroblast in Ad-FAM3A group in AAA tissues. Furthermore, we also detected adenovirus-mediated FAM3A distribution in other organs including heart, liver, and kidney by monitoring flag tag protein.
- Please review Results section: the results of distribution of FAM3A-flag were stated in Results section “*FAM3A overexpression attenuates in vivo pathological outcomes in murine AAA models and maintains the contractile phenotype of VSMCs*”, paragraph4.

8. The transcriptional expression of Klf4 (Fig. 2H) differs from protein expression (Fig. 5B-C). Apart from attributing this to post-translational modifications, could this be a time-dependent regulated phenomenon? Sequential analyses of day 3, 7 and 14 in the elastase-model and day 14, 21 and 28 in the ApoE^{-/-} models would be prudent to address this relevance.

Response: Thank you very much for your good suggestion.

Solutions:

- Please review supplementary Fig. 8b: The Sequential analyses of KLF4 mRNA and protein levels at day 3, 7 and 14 in the C57BL/6-elastase model and day 14, 21 and 28 in the AngII-ApoE^{-/-} model were performed additionally during the revision. It seemed that KLF4 mRNA level down-regulated by FAM3A overexpression at day 3 in the elastase-model. However, no significant changes of KLF4 mRNA levels induced by FAM3A occurred at other time points.
- The results of Fig. 8b were also demonstrated at Results section of “*FAM3A influences KLF4 post-translational modification*”, paragraph3, line1-3.

9. Kindly provide the entire aortic section of representative images in the histology sections for a complete evaluation of spatial expression in aortic layers.

Also, quantification of aortic sections for histological images should be provided.

Response: Thank you very much for your reminder of these details. In the revised paper, the entire aortic sections and relevant quantifications were provided.

Solutions:

- Please review Fig. 1a, c for AAA patient sections and quantifications: We showed the entire section of patients and made quantifications of Masson staining and FAM3A immunofluorescence. It should be noted that the human section was not an entire vessel ring. Generally, AAA patient specimens obtained from surgical operations are fragments, because the diseased vessel rather than entire vascular ring are surgically removed. In addition, an entire human aorta tissue is too big to be embedded in one paraffin-box.
- Please review Figs. 1f, g, and 2a, b, and supplementary Fig. 2a, b for AAA murine sections and quantifications: We showed the entire aorta tissues of AAA mice and made relevant quantifications.
- Please review Fig. 4c, and Supplementary Fig. 4a, c: Moreover, the other results of murine histological images which were tested additionally during revision, were provided with the entire aortic section, such as Fig. 4c, Supplementary Fig. 4a, c.

10. Cytokine expression in aortic tissue would be reflective of the local inflammatory milieu and should be provided in addition to plasma expressions.

Response: Thank you very much for your suggestion. We have detected cytokine expressions in the local inflammatory milieu in addition to plasma expressions.

Solutions:

- Please review Supplementary Fig. 2e for inflammatory cytokines in aortic tissue: Additional experiment to analyze the effects of FAM3A on the local inflammatory milieu using AngII-ApoE^{-/-} and Elastase-C57BL/6 AAA murine models and Ad-FAM3A or recombinant FAM3A administration method.
- Please review the description of these results: Results section of “*FAM3A overexpression attenuates in vivo pathological outcomes in murine AAA models and maintains the contractile phenotype of VSMCs*”, paragraph1, line6 and line11; Results section of “*Supplement of recombinant FAM3A attenuates in vivo pathological outcomes in murine AAA models and maintains the contractile phenotype of VSMCs*”, paragraph1, line11.

11. Apart from Klf4, did the authors analyze Klf2 or Zfp148 (PMID: 31025534) that could also regulate SMC-specific genes and chromatin remodeling?

Response: Thank you very much for your suggestions. We have detected the effects of FAM3A on Klf2 and Zfp148 in vivo.

Solutions:

- Please review Supplementary Fig. 8a: We did experiment to analyze whether FAM3A also regulated Klf2 or Zfp148 protein expression. Surprisingly and interestingly, both Klf2 and Zfp148 protein amount was suppressed by FAM3A. We described these results (please review Results section of "*FAM3A influences KLF4 post-translational modification*", paragraph1, line9) and cited the relevant reference (PMID:31025534 as Reference 18) in the revised paper.

Minor:

Grammatical phrases (i.e. popped, propose a clue, pity etc.) should be reworded in the manuscript at certain places in the text for the scientific audience.

Response: Thank you very much for your nice suggestions. All these inappropriate phrases have been reworded in the revised manuscript considering the scientific audience.

Thanks again for your careful review.

Reviewer #2 (Remarks to the Author):

Chuxiang Lei et al. have investigated the role of FAM3A in vascular smooth muscle cell (SMC) differentiation and reprogramming in the context of abdominal aortic aneurysms (AAAs). They detected decreased levels of FAM3A in human and mouse AAA tissue. Increasing FAM3A levels systemically by adenoviral expression vector or by administration of human recombinant FAM3A (hrFAM3A) significantly reduced AAA formation in two mouse models and decreased (rupture-related) animal death. Gene expression analysis of aortic tissue from treated versus untreated mice revealed various regulated, primarily metabolism-related pathways. In addition to this unbiased approach (which was not further pursued) the authors chose to investigate target genes of SMC differentiation and found a significant impact of FAM3A overexpression/administration in preserving factors of the contractile cell phenotype. Effects were confirmed at the protein level, and further included a decrease in inflammatory and proteolytic regulators of media destruction in the AAA mouse models. In vitro experiments with isolated human aortic smooth muscle cells (HASMCs) documented a dose-dependent, protective effect of hrFAM3A at concentrations ≥ 200 ng/ml in preventing PDGF-BB (or cholesterol) triggered loss of SMC contractile phenotype and differentiation into other cell lineages. Reversal of these effects by the selective SMAD3 inhibitor SIS3 indicated a mechanistic link to TGF-beta signaling, and further downstream effects on phosphorylation and ubiquitination/degradation of KLF-4 were revealed. The authors conclude that "FAM3A maintains a well-differentiation status of VSMCs, at least in part, by a marked activation of TGF β signaling and thereby a suppression of KLF4 action. These findings offer a distinct perspective to develop further therapeutic strategies for AAA aiming at restoring VSMCs homeostasis."

This is a comprehensive study which covers human AAA tissue, intervention approaches in two distinct mouse models of AAA and in vitro analysis of cellular mechanisms. Overall, the manuscript is well composed and describes the experiments in sufficient detail – minor language/grammar corrections may be required. Sample size and experimental replicates seem sufficient, i.e. statistical tests identify significant effects. Comments, points of critique and suggestions for changes are the following:

ORIGINALITY AND SIGNIFICANCE TO THE FIELD

What is already known:

- FAM3A is a ubiquitously expressed cytokine-like protein which promotes mitochondrial ATP synthesis (Yan et al., 2022) and is involved in various pathological conditions including type II

diabetes or NAFLD (Zhang et al., 2018).

- FAM3A is required for muscle stem cell commitment and drives muscle stem cell oxidative metabolism and differentiation (Sala et al., 2019).
- KLF-4 is a central regulator of SMC phenotypic modulation (Deaton et al., 2009).
- In aortic aneurysms, reduced TGF-beta signaling via SMAD2/3 increases KLF-4 protein and activity, which results in loss of the contractile SMC phenotype and cell reprogramming/differentiation into other cell lineages (Chen et al., 2020).

What this study adds:

- In vitro experiments with isolated human aortic smooth muscle cells (HASMCs) document a role of FAM3A in preserving the contractile, differentiated SMC phenotype via TGF-beta signaling in HASMCs and subsequent suppression of KLF-4 action. In vitro doses ≥ 200 ng/ml of human recombinant FAM3A (hrFAM3A) are required.
- Reduced FAM3A expression is detected in human and mouse abdominal aortic aneurysms (AAAs); intervention by systemically increasing FAM3A levels in mouse models interferes with AAA development.

What is not resolved/addressed by this study:

- Is systemic FAM3A administration preventing aneurysm formation via a direct effect on SMC differentiation or via an indirect consequence of FAM3A activity on other cells/pathogenic processes in AAA? Do the in vivo achieved FAM3A doses match the in vitro required concentrations to affect SMC differentiation?
- Which are the main producers of FAM3A in the AAA context? Is FAM3A acting primarily autocrine or paracrine and via which receptors do SMCs receive FAM3A signals?
- How does FAM3A activity in HASMCs regulate TGF-beta signaling in these cells?
- Would systemic FAM3A therapy also work to prevent progression of established disease in the murine AAA models (which corresponds to the clinical situation)?

Response in general: We feel great thanks for your professional review work on our article. As you are concerned, there are several problems that need to be addressed. According to your nice suggestions, we have made extensive corrections to our previous draft. We have added necessary data to supplement our results and edited our article extensively. Please review the detailed responses which are listed point by point below.

CONCLUSIONS AND CLAIMS

1. In human healthy aorta (Fig. 1C) FAM3A seems to be predominantly expressed in the

adventitia rather than in the media, where FAM3A staining is mainly found closer towards the intima. The authors fail to comment on this circumstance or how this would fit their theory of FAM3A loss in the media/SMCs during AAA development.

Response: Thanks for your careful review of our paper. We feel sorry that we did not provide enough information about this image previously.

1) Concerning the FAM3A locations in aorta tissues

In the revised paper, we provided clearer pictures of entire sections of aorta and indicate the side of adventitia or intima. Endogenous FAM3A seems to be localized (being expressed or secreted and transferred here) abundantly in the media as well as intima in normal aorta. However in aneurysm aorta, the intima is much damaged and even lost.

Solutions:

- Please review Fig. 1c, f: We provided clearer pictures of entire sections about the FAM3A locations in human or murine aorta tissues as well as its co-localization with SMC (α SMA). And the side of intima was indicated.

2) Concerning the gross change and VSMC-specific change of FAM3A in AAA microenvironment (fit theory of FAM3A loss in the media/SMCs)

In our study, we found that the gross endogenous FAM3A amount in AAA tissue was decreased in contrast to normal aorta. Thus, we further explored cell-specific changes of endogenous FAM3A in AAA tissue: 1) Using single-cell mRNA sequencing data from murine AAA models, we found that endogenous FAM3A is expressed in VSMCs, endothelial cells, fibroblasts, and macrophages, and its expression decreased in VSMCs, endothelial cells, fibroblasts, and macrophages, with an overall reduction in AAA tissues (supplementary Fig. 1c); 2) Using co-staining of immunofluorescence between α SMA and FAM3A, we found that endogenous FAM3A was located abundantly in VSMCs (being expressed by VSMCs or being secreted by other cells and moving to VSMC site) and decreased in AAA tissue (Fig. 1c, f).

Despite alteration of endogenous FAM3A amount in other cells, we thought that the reduction of endogenous FAM3A expression and location in VSMC was significant. Also considering that VSMCs are the main functional cells in aorta, we then studied the effect of FAM3A on VSMCs in context of AAA at the first stage of our research.

Solutions:

- Please review Supplementary Figs. 1c, d, e and 4a, c and Results section of “*Cell-specific FAM3A expression changes in AAA microenvironment*”: We showed the

results of cell-specific FAM3A status or changes (endogenous or exogenous FAM3A: expression, binding, or location) in the revised paper.

- Please review the Discussion section (paragraph2, line9 and paragraph3): In the revised paper, we made a more detailed discussion about FAM3A expression (cellular sources) and functional manners in VSMCs (autocrine, paracrine, and even endocrine) in AAA pathology.

2. Systemic administration of hrFAM3A (or FAM3A adenovirus) is likely to affect/infect various cell types: hrFAM3A will more effectively target ECs than SMCs, FAM3A adenovirus will predominantly end up in the liver. The authors did not prove that in vivo effects are mediated directly by SMCs, i.e. the positive impact on SMC differentiation might be an indirect effect based on the responses by other cell types. In line, RNAseq analysis of AAA tissue from Ang-II mice with FAM3A versus control adenovirus revealed a number of regulated pathways which were predominantly not SMC-specific (e.g. cytokine signaling). Since the adenovirally expressed FAM3A carries a FLAG tag, the authors may try to follow protein expression by immunofluorescence staining of aneurysms and other organs. It would also be good to document if additional effects (potential side effects in therapeutic application) were observed upon systemic FAM3A administration - on cells/organs other than SMCs and the aorta? Ultimately, only SMC-specific overexpression of FAM3A would answer the question whether the protective effect against AAA development is indeed mediated by the direct action on SMCs.

Response: Thanks very much for your careful review and nice suggestions. What you mentioned are important issues.

Unlike intracellularly positioned proteins, FAM3A is a cytokine which play function not restrictedly in cytoplasm or organelle. To explore the direct FAM3A function on one specific cell type, intervention of its receptor in this cell type is the most effective way.

1) Concerning the direct or indirect effects of FAM3A on VSMC differentiation reprogramming and SMC-specific overexpression of FAM3A

We think that the FAM3A regulates VSMCs in four manners: ① endocrine FAM3A from circulating sources (direct function of FAM3A); ② paracrine FAM3A from other cells in AAA microenvironment (direct function of FAM3A); ③ autocrine FAM3A by VSMC itself (direct function of FAM3A); ④ other cells which were phenotype-resaped by FAM3A, interact and influence VSMC by cell-cell crosstalk in AAA microenvironment (indirect function of FAM3A). In a simple and visual way, we illustrate our knowledge as below:

In our study, the in vivo data just evidenced that systematic administration of exogenous FAM3A could inhibit AAA formation through regulating VSMC plasticity. And these functions of FAM3A on VSMCs in vivo may be a combination of direct and indirect functions of FAM3A on VSMCs. Further, it is likely that FAM3A protected AAA also through its effect on other cell types such as endothelial cells, which could not be precluded from our current study.

We think that FAM3A therapeutic applications, targeting SMCs and systematic administration, have special and different significances as well as values in context of alleviation of AAA formation.

Maybe it is hard to elucidate the “**direct function**” of FAM3A in vivo on VSMCs alone excluding other cell types, because FAM3A receptor is not identified currently. We thought that even if we over-express or silence FAM3A specifically in VSMCs in vivo, the FAM3A from other cellular sources could influence VSMCs, and the over-produced FAM3A by VSMCs could protect AAA through targeting other cell types. Importantly, SMC-specific overexpression of FAM3A is a good way to study the FAM3A autocrine effects on VSMCs. So it is a limitation here. We feel sorry that FAM3A receptors have not been identified and it is difficult to silence/block the receptor specifically in VSMCs to observe the direct function of FAM3A on SMCs. Actually, we've been screening FAM3A receptor for a lot of time.

Solutions:

- Please review the Discussion section the second paragraph to last (paragraph7): In the revised paper, we made a discussion about the limitations and the functional features of systematic administration of FAM3A, and gave a prospect to the future study concerning cell-specific FAM3A administration/intervention.

- Please review the Discussion section, paragraph4, line8: We made an explanation why we carry out research currently in a manner of systematic administration of FAM3A: “*Since the absent knowledge of FAM3A receptor, we first explored FAM3A function systematically by treating murine AAA models in a manner of systematical administration*”.
- Please review supplementary Figs. 1e, 5a, 7b and Figs 5d-f and 6 (in vitro) and Figs. 4b,c and 8d (in vivo): Really, the direct function of FAM3A on VSMCs is one of the important issues. We explored this issue as far as possible. ①We provided in vitro evidences about FAM3A “**direct function**” on VSMCs, observing that recombinant FAM3A could attach to VSMCs (Supplementary Fig. 1e), and recombinant FAM3A altered VSMC contractile (Supplementary Fig. 5a) and trans-differentiation markers expression (Supplementary Fig. 7b), influenced KLF4 total protein, phosphorylation, ubiquitination, and nuclear localization (Figs. 5d-f and 6). ②Also, we did additional experiments, and some in vivo evidences also have been provided about “**direct function**” of FAM3A on VSMCs to some degree, such as detection of co-localization between VSMC marker and intermediate cell markers using FACS by flow cytometry (Figs. 4b and 8d) and immunofluorescence co-staining (Fig. 4c).

2) Concerning the in vivo distribution of FAM3A when administrating FAM3A systematically with Ad-FAM3A or recombinant FAM3A

In the revised paper, the in vivo distribution of FAM3A by systematic administration with Ad-FAM3A or recombinant FAM3A were determined by flag or c-myc tag proteins respectively. The exogenous FAM3A distribution in AAA tissues and other organs such as heart, liver, and kidney were detected. The results showed that recombinant FAM3A and Ad-FAM3A distributed in aorta, heart, liver, and kidney. In AAA tissue, the Ad-FAM3A co-located in VSMCs, endothelial cells, macrophages, and fibroblasts, and the recombinant FAM3A co-located in VSMCs and endothelial cells.

Solutions:

- Please review the supplementary Fig. 4a-d: In the revised paper, in vivo distribution of exogenous FAM3A was detected in aorta (specific to cell types), heart, liver, and kidney. Also, these results were stated in the Results section of “*FAM3A overexpression attenuates in vivo pathological outcomes in murine AAA models and maintains the contractile phenotype of VSMCs*” paragraph4, and “*Supplement of recombinant FAM3A attenuates in vivo pathological outcomes in murine AAA models and maintains the contractile phenotype of VSMCs*” paragraph2.

3) Concerning the additional effects or potential side effects in therapeutic application

In the revised paper, the in vivo potential side effects of FAM3A therapeutic application was detected in heart, liver, kidney, and plasma, and in vitro cytotoxicity was examined in VSMCs, macrophages, fibroblasts, and endothelial cells. The supplement of recombinant FAM3A showed no significant cytotoxicity and organ toxicities of heart, liver, and kidney in mice. Also, the plasma showed no significant inflammation status.

Solutions:

- Please review the Supplementary Fig. 4e, f: The potential side effects in therapeutic application of recombinant FAM3A was provided. These results were stated in the Results section of “*Supplement of recombinant FAM3A attenuates in vivo pathological outcomes in murine AAA models and maintains the contractile phenotype of VSMCs*”, paragraph2, line9; paragraph4.

3. Concentrations of ≥ 200 ng/ml hrFAM3A were required to rescue the SMC phenotype in vitro. In vivo, weekly administration of 200 ng hrFAM3A per mouse is likely to result in circulating lower levels and even lower levels reaching the SMC compartment. How do the authors reconcile this discrepancy? When administering hrFAM3A i.v. at 1-week intervals it would be of interest to also monitor plasma/serum levels of FAM3A for the achieved blood concentrations and kinetics. Furthermore, while human and murine FAM3A show a high degree of homology, have the authors verified that hrFAM3A is affecting murine SMCs at comparable doses which were effective in in vitro experiments with human aortic SMCs?

Response: Thanks for your careful review and suggestion. What you mentioned are important issues.

1) Concerning the “discrepancy of in vitro and in vivo FAM3A concentrations”

Generally, the total blood amount of a mouse is about 2 ml. We injected a mouse 200 ng FAM3A each time (1 μ g/mL, 200 μ l). Theoretically, the exogenous FAM3A contributed a 100 ng/ml at the plasma total concentration. Furthermore, the endogenous FAM3A secreted by different types of cells, such as endothelial cells, is another source contributing plasma total concentration of FAM3A. Additionally, the degradation rate of FAM3A may be different between in vitro and in vivo. In the in vitro experiments, FAM3A stimulation (200 ng/ml) was just once without additional supplement during the experimental course of 48 hours, and we detected that the hrFAM3A concentration in the medium decreased less than 10 ng/ml at the 48-hour time point (supplementary Fig. 4g).

The in vivo hrFAM3A administration was 200 ng each time with a later additional supplement of 200 ng per week for four times (a total of five times), and we detected that the hrFAM3A concentration maintained no less than 10 ng/ml in plasma during the mice modeling process (supplementary Fig. 4g). We thought that these events together reduced the discrepancy between in vivo and in vitro FAM3A concentrations. Moreover, in our primary experiments, we also tried the dosage of 1ug FAM3A each time (1 μ g/mL, 1 mL) which also had an inhibitory efficacy for mice AAA, and we chose the effective and medium dosage of FAM3A (200 ng each time) in the following experiment.

2) Concerning the “plasma/serum concentrations and kinetics of FAM3A”

The monitoring of FAM3A plasma concentration and kinetics is necessary to study therapeutic function of FAM3A by systematic administration and determine optimum administration frequency. We have detected plasma concentrations and kinetics of hrFAM3A in the revising processes.

Solutions:

- Please review Supplementary Fig. 4g: We showed the plasma hrFAM3A concentrations in vivo and medium hrFAM3A concentrations in vitro at different time points in the revised paper.

3) Concerning the “high degree of homology between human and mice FAM3A” and “verified that hrFAM3A is affecting murine SMCs at comparable doses ”

Actually, there is a high degree of homology between human and mice FAM3A. And that hrFAM3A (Origene, cat# TP303495) being used in mice (both in vivo and in vitro) was reported by Sala et al. previously (*PMID: 30996264: in their Method section of “Animal procedures” paragraph4, line1 and Method section of “Cell culture procedures” paragraph1, line10*) (the *PMID: 30996264* was cited as Reference 13 in our paper), confirming its high homology in human and mice.

Also, we selected several markers to verify the effects of recombinant human FAM3A on murine SMC differentiation at comparable doses (200 ng/mL) in vitro, such as the below for TAGLN. Please make a review.

4. The claim that "...these results suggested a potential therapeutic strategy by using a supplement of FAM3A in AAA." seems overstated to me due to the lack of

a. data on non-SMC effects of systemic FAM3A administration which might have adverse impact/side effects

b. mouse studies with FAM3A administration on established AAA disease – assessing inhibition of progression (as would be the case for the clinical situation).

Response: Thank you for your pertinent comments. Concerning "therapeutic strategy", the description and relevant results should be strict. The word "therapeutic strategy" for AAA is not suitable from our study which explored the FAM3A function in AAA gradually forming process rather than the FAM3A efficacy in already established AAA.

Solutions:

- We have made a modification as "...these results suggested that supplement of recombinant FAM3A had a potential efficacy in alleviation of AAA formation". Actually, the states of "therapeutic" in AAA "progression" in our FAM3A study are not pertinent and strict, and we changed these words in context of our experimental results throughout the manuscript.

5. Evidence is provided that FAM3A regulates SMC trans-differentiation into other cell types (Fig. 4 and Suppl. Fig. IV-B). While the in vitro data are convincing – again showing regulation by FAM3A at doses ≥ 200 ng/ml, the in vivo data are based on Western blot analysis of cell lineage markers for aorta tissue extracts. This does not provide actual proof for SMC trans-differentiation, since e.g. a lower level of CD68 expression could simply be based on a lower accumulation of macrophages in the aorta. AAA tissue stainings from mouse aortas would be of interest.

Response: Thank you very much for your great suggestion. Your comments are instructive and valuable for us to promote the study.

To acquire the further evidences for VSMC trans-differentiation regulated by FAM3A in AAA microenvironment, the VSMC-targeted analysis is important. There are two ways for us targeting VSMC in AAA tissues: ① By histology staining with VSMC-specific marker such as α SMA; ② By flow cytometry technique sorting VSMCs in AAA tissues.

In the revised paper, we have detected VSMC-targeted trans-differentiation in vivo by both histology immunofluorescence co-staining and fluorescence-activated cell sorting

(FACS) with flow cytometry. These results confirmed the regulatory roles of FAM3A in VSMC trans-differentiations in vivo.

Solutions:

- Please review Fig. 4b, c: We also described these new data in Results section of “*FAM3A influences VSMC trans-differentiation towards other intermediate cell types*”, paragraph1, line8, line13.

Thanks again for your valuable suggestion.

6. Supplementary Figure V illustrates that KLF-4 inhibition (or silencing) can rescue the effect of FAM3A silencing with respect to loss of contractile SMC markers. The same experimental setup would be expected for supplementary Figure VI where the authors address SMC trans-differentiation into other cell types. However, the authors switched to SMC stimulation by PDGF-BB and demonstrate that KLF-4 silencing can prevent PDGF-BB induced trans-differentiation (without link to FAM3A activity)?

Response: Thank you for your careful review. The same experimental setup as original Supplementary Fig. V (KLF-4 inhibition can rescue the effect of FAM3A silencing on loss of contractile SMC markers), was also performed on SMC trans-differentiation detection.

Solutions:

- Please review Supplementary Fig. 9: The chemical inhibition or gene silencing of KLF-4 can rescue the effect of FAM3A silencing on regulation of SMC trans-differentiation markers. We also described these new data in Results section of “*FAM3A influences KLF4 post-translational modification*”, paragraph5.

DATA AND METHODOLOGY

7. Human samples are restricted to 6 AAA patients and 6 healthy controls who appear well matched (except for hyperlipidemia – and age: 66 vs 55 years). It would be of interest to add the information, whether the AAA or healthy aortas were affected by atherosclerosis.

Response: Thank you for your nice suggestion. Please review Supplementary Table 1: In the revised paper, the atherosclerosis information has been added in AAA patients and controls.

8. Source/cloning of the adenoviral vector (for FAM3A overexpression) is not specified.

Response: Thank you for pointing out this problem. The FAM3A overexpression adenovirus (pAV[Exp]-EF1A>mFam3A [NM_001379181.1]/FLAG:IRES:mCherry) was constructed and packaged by pAd/CMV/V5-DEST Gateway Vector and subsequently titered in 293A cell line using a commercial ViraPower Adenoviral Gateway Expression Kit (Invitrogen, Cat#K4930-00).

In the revised paper, we have described these information for source/cloning of the adenoviral vector. Please review Methods section of "*Overexpression of FAM3A in mice by adenoviruses and aorta RNA sequencing analysis*", line3.

9. Immunostaining of human and mouse aorta cryosections for SMA and FAM3A is shown but not explained in the Methods section (information is restricted to in vitro HASMCs cell cultures).

Response: Thank you for pointing out this problem. We are really sorry about this carelessness. In the revised paper, we have described the process of immunostaining of human and mouse aorta for α -SMA and FAM3A. Also, immunofluorescence co-staining about VSMC transdifferentiation (added in this revision) was stated. Please review Methods section of "*Cytological staining*" and "*Histological staining*".

10. RNA sequencing data are shown but the sample preparation and analysis is not explained in the methods section.

Response: Thank you for your careful review. In the revised paper, we have described the detailed process of RNA sample preparation and data analysis. Please review Methods section of "*Overexpression of FAM3A in mice by adenoviruses and aorta RNA sequencing analysis*" paragraph2.

11. The authors present in vitro experiments with isolated HASMCs where they show dose-dependent effects of hrFAM3A on the SMC phenotype (Suppl. Fig. II). All experiments were conducted in the presence of PDGF-BB or cholesterol, but the authors fail to mention or present controls without PDGF-BB or cholesterol addition.

Response: Thank you for your careful review. In the revised paper, we have presented the relevant controls without PDGF-BB in all the in vitro experiments of VSMCs. Please review Figs. 5d, e, 6a, c, and 7a and Supplementary Figs. 5a, 6a, b, and 7b.

12. The confocal microscopy images provided in Fig. 5F and 7F are not very convincing in

demonstrating the effect of FAM3A reducing KLF4 nuclear localization. Similarly, fluorescent images in Suppl. Fig. V-A don't support the proposed ROS increase upon FAM3A silencing. Yet, quantitations with highly significant differences are displayed.

Response: Thank you very much for pointing out these problems. In the revised paper, more convincing images were presented. Please review Figs. 5f and 7f, and Supplementary Fig. 8c.

MINOR ISSUES

13. Supplementary Table I: Please specify whether median/mean and SD or SE are given for years of age.

Response: Thank you very much for your careful review. Please review Supplementary Table 1: The median/mean and SD or SE were specified in term of Age.

14. Methods/Cell Culture: “100 g/ml of streptomycin” should likely read “100 µg/ml of streptomycin”.

Response: Thank you very much for your careful review. We feel sorry for this mistake and have made a correction as “100 µg/ml” in the Methods section.

15. Methods/qRT-PCR: What is meant by “takeoff values”? The threshold cycle (Ct) values?

Response: Thank you very much for your careful review. Yes, the “takeoff values” we mentioned is the “threshold cycle (Ct) values” generated from PCR analysis software. We have changed the description as “threshold cycle (Ct) values”. Please review Methods section of “*Quantitative real-time PCR (qRT-PCR)*”, line6.

16. Legend to Figure 2: Adenoviral gene transfer should be more clearly mentioned.

Response: Thank you very much for your review. We feel really sorry for our carelessness. In the revised paper, we detailed experiment processes in Methods section, and legend to all Figures.

17. Legend to Figure 3E: While the y-axis refers to serum levels, the legend specifies that plasma was analyzed – please correct.

Response: Thank you very much for your careful review. It was really our carelessness. We have corrected as “plasma” in the figure. Please review Fig. 3e in the revised version.

18. Legend to Figure 5: “subjected to adenoviral FAM3A overexpression (A) and recombinant FAM3A supplement (B)” should read “subjected to adenoviral FAM3A overexpression (B) and recombinant FAM3A supplement (C)”. Please also indicate the difference between the two quantification plots (for co-localization) in 5F.

Response: Thank you very much for your careful review. We have corrected these mistakes.

In Fig. 5f, we used two different methods of computation by Image J software to quantify the KLF4 nucleus location: Pearson’s correlation coefficient (PCC, left) and Manders' Colocalization Coefficients (MCC, right). Briefly, ① Pearson’s coefficient, namely PCC computation method, results in a value describing the **relative KLF4 localized in nucleus**; ② Manders' Colocalization Coefficients (MCC), which focus on overlapped area of two different color fluorescence, results in a value describing the proportion of overlapped area (both KLF4⁺ and DAPI⁺, namely the red fluorescence co-located with blue fluorescence) to the total red fluorescence (KLF4⁺) area (roughly, **overlap coefficient=KLF4 in nucleus/KLF4 total area**).

In the revised paper, we specified the quantification methods in legends of Figs. 5f and 7f. There are some literatures to describe these fluorescence quantification methods such as the below. Please make a review:

[1]. Dunn K W , Kamocka M M , Mcdonald J H . A practical guide to evaluating colocalization in biological microscopy[J]. *AJP: Cell Physiology*, 2011, 300(4):C723-C742.

[2]. Measurement of co-localization of objects in dual-colour confocal images[J]. *Journal of Microscopy*, 1993, 169(3):375-382.

19. Legend to Figure 7: Explanation of Figure part 7F is missing.

Response: Thank you very much for your careful review. Legend to Fig. 7f has been described in the revised paper.

REFERENCE TO PREVIOUS LITERATURE

20. While the authors cite essential previous studies, they occasionally fail to address discrepancies or relate their results to these studies, e.g. Jia et al., 2014, Xiang et al., 2020: Xiang R. et al. reported FAM3A effects on angiotensin-II driven hypertension which has not been observed in the present study.

Response: Thank you very much for your careful review. These two references are from the same laboratory and they proposed FAM3A as a mitochondrial protein. However David Sala et al. cited their publications (as references 25 and 26 in his paper PMID:30996264) and denied their viewpoint of mitochondrial location of FAM3A by the experimental evidences in 2019 at *Nature communications* (PMID:30996264). We favor the viewpoint of David Sala that FAM3A is a cytokine which is secreted by cells and not anchored intracellularly. However, it should be noted that *fam3a* gene have a large number of splice variants. Maybe it is an explanation of discrepancy between their reports and ours. Was the FAM3A in their publications the same one in our study? We have look up their paper. However we failed to find the FAM3A gene ID or Accession Number they used in constructing murine models. We surmise that the complexity is very likely due to FAM3A alternative splicing. We presented the FAM3A gene information in our paper at the Adenovirus over-expression (mFam3A [NM_001379181.1]) and human FAM3A recombinant protein we used (hFAM3A [NM_021806]). Moreover, the manufacturer and product number of human FAM3A recombinant protein were provided in the manuscript, through which the amino acid sequences of FAM3A could be traced online. Also, in Supplementary Table 3 (PCR primer), the FAM3A primer sequence was provided. The information provided could trace which peptide or protein specified to the FAM3A we studied in our paper.

Further, we have cited these references and made a discussion about these controversial issues in our paper. Please review Discussion section, paragraph 2, line5, line14.

CITATIONS

CHEN, P. Y. et al. 2020. Smooth Muscle Cell Reprogramming in Aortic Aneurysms. *Cell Stem Cell*, 26, 542-557 e11.

DEATON, R. A. et al. 2009. Sp1-dependent activation of KLF4 is required for PDGF-BB-induced phenotypic modulation of smooth muscle. *Am J Physiol Heart CircPhysiol*, 296, H1027-37.

JIA, S. et al. 2014. FAM3A promotes vascular smooth muscle cell proliferation and migration and exacerbates neointima formation in rat artery after balloon injury. *J Mol Cell Cardiol*, 74, 173-82.

SALA, D. et al. 2019. The Stat3-Fam3a axis promotes muscle stem cell myogenic lineage progression by inducing mitochondrial respiration. *Nat Commun*, 10, 1796.

XIANG, R. et al. 2020. VSMC-Specific Deletion of FAM3A Attenuated Ang II-Promoted

Hypertension and Cardiovascular Hypertrophy. *Circ Res*, 126, 1746-1759.

YAN, H. et al. 2022. Intracellular ATP Signaling Contributes to FAM3A-Induced PDX1 Upregulation in Pancreatic Beta Cells. *ExpClin Endocrinol Diabetes*, 130, 498-508.

ZHANG, X. et al. 2018. FAM3 gene family: A promising therapeutical target for NAFLD and type 2 diabetes. *Metabolism*, 81, 71-82.

Response: Thanks very much for your careful review. We have cited all these references as below:

- CHEN, P. Y. et al. 2020. Smooth Muscle Cell Reprogramming in Aortic Aneurysms. *Cell Stem Cell*, 26, 542-557 e11. (**Reference 23**)
- DEATON, R. A. et al. 2009. Sp1-dependent activation of KLF4 is required for PDGF-BB-induced phenotypic modulation of smooth muscle. *Am J Physiol Heart CircPhysiol*, 296, H1027-37. (**Reference 42**)
- JIA, S. et al. 2014. FAM3A promotes vascular smooth muscle cell proliferation and migration and exacerbates neointima formation in rat artery after balloon injury. *J Mol Cell Cardiol*, 74, 173-82. (**Reference 7**)
- SALA, D. et al. 2019. The Stat3-Fam3a axis promotes muscle stem cell myogenic lineage progression by inducing mitochondrial respiration. *Nat Commun*, 10, 1796. (**Reference 13**)
- XIANG, R. et al. 2020. VSMC-Specific Deletion of FAM3A Attenuated Ang II-Promoted Hypertension and Cardiovascular Hypertrophy. *Circ Res*, 126, 1746-1759. (**Reference 8**)
- YAN, H. et al. 2022. Intracellular ATP Signaling Contributes to FAM3A-Induced PDX1 Upregulation in Pancreatic Beta Cells. *ExpClin Endocrinol Diabetes*, 130, 498-508. (**Reference 37**)
- ZHANG, X. et al. 2018. FAM3 gene family: A promising therapeutical target for NAFLD and type 2 diabetes. *Metabolism*, 81, 71-82. (**Reference 36**)

Thanks again for your careful review.

Reviewer #3 (Remarks to the Author):

Lei C et al. investigated the role of FAM3A as a regulator of differentiation of vascular smooth muscle cells (VSMC) in the pathogenesis of abdominal aortic aneurysm (AAA). In human AAA samples as well as in two experimental models of AAA (Pancreatic elastase-induced AAA and ApoE^{-/-} transgenic mice treated with ANGII), they found a downregulation of FAM3A. Overexpression or supplement of FAM3A prevented in part the dilation of the aorta and improved the survival, but FAM3A did not protect from functional changes (increased blood pressure). Further mechanistic studies demonstrate that FAM3A increases the expression of contractile elements in VSMC and reduces the proliferation of VSMC. Next, they propose that these effects by FAM3A on VSMC function are mediated through an activation of TGFbeta/SMAD3 signaling with subsequent suppression of Klf4 phosphorylation and elevation of Klf4 ubiquitination.

The study includes a comprehensive set of experiments such as human samples, two experimental models of AAA, in vivo and in vitro overexpression of FAM3A, in vivo treatment with FAM3A and transcriptomic analysis. While the data provide new mechanistic insights in the pathogenesis of AAA and a possible therapeutic approach, there are several aspects that should be addressed.

Response in general: Firstly, we sincerely thank you for your valuable feedback on our article. According to your nice suggestions, we have made extensive corrections to our previous draft. We have added necessary data to supplement our results and edited our article extensively. Please review the detailed responses which are listed point by point below.

General comments:

1) In both experimental models, the authors demonstrate a reduction of the diameter of the aorta and reduced inflammation (e.g., IL-6), but no differences in function (blood pressure).

a. What is the cause of the discrepancy between structure and function?

Response: Thank you for your instructive comments. We thought there are some matters below:

Firstly, Blood vessel is a tissue with elasticity and compliance. Within a certain pressure range, through relaxing and contracting, the vessel buffer blood pressure and maintain a homeostasis of blood pressure. Once the damage is up to some extent that leads the vessel structure to collapse, its ability is too weak to maintain homeostasis of blood pressure, leading to significant changes in blood pressure.

Secondly, the blood pressure we measured was at the tail vein which reflected much more the systemic vascular state than local aneurysm aorta.

Thirdly, the protection of FAM3A on the local aneurysm aorta (maintenance of integrity of vascular structure, anti-inflammation) may indeed alleviate the blood pressure disorder at the aneurysm tissue. However, at the tail vein site (given a probability that there is no significant change in the vascular tissue structure of the tail vein), there will be an adjustment/buffer leading to a maintenance of blood pressure homeostasis, and FAM3A function will be covered up or relieved even if it has an effect on blood pressure. Under this circumstance, FAM3A function on blood pressure is undetectable.

Concerning FAM3A function, the discrepancy between maintenance of vessel structure and failure in regulation of blood pressure, we thought it is likely due to the method of detecting blood pressure, especially the monitored location of vessel as well as the extent of the vessel damage. Theoretically, FAM3A could regulate vessel structure as well as vessel's function (blood pressure). The vessel histological structure or pathological change (a more microscopic detection) could not be reflected completely by monitoring blood pressure (it is a more macroscopic detection).

b. The authors also propose that “the degree of elastic fiber disintegration was significantly lower” after FAM3A treatment. In line, with this notion MMP2 and MMP9 proteins were decreased in ApoE-/-+ANGII treated with FAM3A when compared to those treated with vehicle. Was this change in proteolytic activity reflected in changes in elastic fibers or collagen? Was the effect of FAM3A on protease expression also detectable in the other models (elastase-induced AAA; adenoviral FAM3A overexpression)?

Response: Thank you very much for your comments.

In AAA microenvironment, disintegration and structure disorder of elastic fibers or collagen in extracellular matrix (ECM) is a major contributor to aorta weakness and AAA formation. The disintegration of elastic fibers or collagen in ECM is mainly induced by an elevated proteolytic activity of MMPs which is caused by abnormal VSMCs. The elevated proteolytic activity of MMPs reflects disintegration in elastic fibers or collagen in AAA tissues.

Thus, the detection of MMP expressions which reflect the degradation of elastic fibers or collagen in ECM, is significant to assess inhibitory function of FAM3A in AAA formation.

Solutions:

- Please review Figs. 2i and 3f, and Supplementary Fig. 2d: We have presented the inhibitory actions of FAM3A in activity of proteolytic enzymes of MMP2 and MMP9 (ApoE^{-/-}+AngII-induced AAA, elastase-induced AAA; recombinant FAM3A, adenoviral FAM3A overexpression, respectively).

c. How were elastic fibers studied? H&E staining is not the ideal technique to quantify elastic fibers? What might be the impact of matrix stiffness (elastic fibers, collagens) or increased contractility of VSMC on the increased blood pressure after supplement/overexpression of FAM3A despite reduced diameter of the aorta.

Response: Thank you very much for your comments.

Generally, in AAA pathological study, it is important for investigators to detect MMP expression/activity through which the elastic fiber integrity or dynamic were reflected. Really, H&E staining is not the ideal technique to quantify elastic fibers.

Concerning the blood pressure

If its maintenance of vessel structure so great leading to a nearly normal vessel structure, we thought that the blood pressure at local aorta would be decreased by FAM3A; If its ability of maintaining vessel integrity is not great enough, the aneurysm aorta is also in a pathological status (even if an alleviation of injury) and do not recover to its normal structure. The blood pressure that we observed may not be influenced by FAM3A significantly.

The observation that FAM3A had an ability to protect vessel structure, was not equivalent to the issue that we could observe FAM3A changing the blood pressure. The blood pressure is a value/data which depend on vessel status (physiological or pathological) and the location we tested.

Monitoring blood pressure can reflect the status of blood vessels, within a certain range. The histology/structure vessel changes, such as subtle alteration in the early stage of the lesion and the severe damage in the late stage of vessel disease, are difficult to be reflected accurately in blood pressure monitoring. The vessel is an interesting tissue which has compliance and great buffering function.

Thank you very much for raising this interesting and thought-provoking question.

d. Does FAM3A affect fibroblast function? These aspects should be elaborated further.

Response: Thank you very much for your comments.

In our exploration, we also assessed the correlation between fibroblast and FAM3A, including ① fibroblast's ability to produce FAM3A (please review Supplementary Fig. 1c,d), ② fibroblast's affinity to FAM3A (please review Supplementary Figs. 1e, 4c), and ③ the cytotoxicity in the situation of fibroblast exposed to recombinant FAM3A (please review Supplementary Fig. 4f).

We thought, if fibroblast could be influenced by FAM3A, its ability to bind to FAM3A is of importance. From our preliminary results, it seemed that there was not a strong or obvious affinity between FAM3A and fibroblast, due to c-myc-FAM3A located few at fibroblasts in vitro (Supplementary Fig. 1e) and in vivo (Supplementary Fig. 4c), despite a few amount FAM3A produced by fibroblast (Supplementary Fig. 1d). However its expression in and affinity to VSMCs were more significant. Maybe, FAM3A could influence fibroblasts not as strong as VSMCs.

Solutions:

- Please review Supplementary Fig. 1 c-e: We also described these results in Results section of "*Cell-specific FAM3A expression changes in AAA microenvironment*" line6, line9, line15.
- Please review Supplementary Fig. 4c: We also described these results in Results section of "*Supplement of recombinant FAM3A attenuates in vivo pathological outcomes in murine AAA models and maintains the contractile phenotype of VSMCs*", paragraph2, line2.
- Please review Supplementary Fig. 4f: We also described these results in Results section of "*Supplement of recombinant FAM3A attenuates in vivo pathological outcomes in murine AAA models and maintains the contractile phenotype of VSMCs*", the last paragraph.

2) The authors focused primarily on the differentiation of VSCM. Since SMAD3 and Klf4 are also regulators of cell survival and proposed by the authors to be downstream of FAM3A, it is of interest to assess the impact of FAM3A on survival of VSMC. This is also of importance based on the finding that FAM3A regulates the pro-proliferative AKT signaling.

Response: Thank you very much for your comments. In the revised paper, we further assessed survival fraction of VSMCs exposed to recombinant FAM3A by detecting colony formation ratio. A little unexpectedly, it seemed that FAM3A inhibited VSMC proliferation. We present the result in Supplementary Fig. 4h. Please make a review. We thought it is likely that FAM3A promote cell maturation and differentiation (promote the survival of

terminally differentiated/functional cells), and inhibit cell immaturity/dedifferentiation (suppress the survival of dedifferentiated or undifferentiated cells, such as stem cells).

3) The authors propose that FAM3A reduces Klf4 phosphorylation and increases Klf4 ubiquitination through TGFbeta and SMAD3. However, the link between Smad3 and Klf4 has not been studied. Does blocking of TGFbeta signaling (SMAD2/3) prevent the modulation of Klf4 and the differentiation of VSMC? The FAM3A-SMAD3-Klf4 axis requires further exploration.

Response: Thank you very much for your comments. Really, blocking of TGFβ/SMAD3 with SIS3 prevent the function of FAM3A in modulation of Klf4 and the differentiation of VSMC. As a TGFβ downstream signal, SMAD3 can be inhibited specifically by SIS3. In our study, SIS3 could hamper the effects of FAM3A on KLF4 modification and VSMC differentiation, suggesting a link between SMAD3 and KLF4 in the context of FAM3A administration.

Solutions:

- Please review Fig. 7d-f for the **in vitro** results of correlation between **TGFβ/SMAD3** and the **modulation of Klf4**: inhibition of SMAD3 by SIS3 hampered FAM3A functions on KLF4 total amount, phosphorylation level, nuclear localization, and ubiquitination.
- Please review Fig. 7b,c for the **in vitro** results of correlation between **TGFβ/SMAD3** and **differentiation of VSMC**: inhibition of SMAD3 by SIS3 hampered FAM3A functions on VSMC differentiation.
- Please review Fig. 8b,f for the **in vivo** results of correlation between **TGFβ/SMAD3** and the **modulation of Klf4**: inhibition of SMAD3 by SIS3 hampered FAM3A functions on KLF4 total amount, phosphorylation level, and nuclear localization.
- Please review Fig. 8b-d for the **in vivo** results of correlation between **TGFβ/SMAD3** and **differentiation of VSMC**: inhibition of SMAD3 by SIS3 hampered FAM3A functions on VSMC differentiation.

4) Klf4 phosphorylation/ubiquitination in animal models of AAA with and without Klf4 overexpression/supplement has not been studied in the present manuscript. Is the cytoplasmic/nuclear localization altered in VSMC by FAM3A in vivo.

Response: Thank you very much for your comments.

(1) Concerning the “Klf4 phosphorylation/ubiquitination in animal models of AAA with and without Klf4 overexpression/supplement”

In the revised paper, we induced KLF4 by APTO-253 in the AAA mice model in the context of FAM3A administration. The total and phosphorylation levels of KLF4 were significantly elevated by KLF4 inducer (Fig. 8b), leading to a suppressed FAM3A functions in maintaining VSMC contractile phenotype and inhibiting VSMC trans-differentiation (Fig. 8c).

Solutions:

- Please review Fig. 8b, c: In FAM3A-overexpressed AAA murine models, KLF4 signal was induced by APTO-253, and the KLF4 phosphorylation, VSMC plasticity were determined. The KLF4 induction led to a suppressed FAM3A functions on maintaining VSMC contractile phenotype and inhibiting VSMC trans-differentiation.

(2) Concerning the “Klf4 ubiquitination in vivo”

Concerning the detection of KLF4 ubiquitination in animal models of AAA, there is a technique limitation of in vivo ubiquitination test, mainly being attributed to MG132, the proteasome inhibitor, which could not be used safely in living animals. So few studies investigated ubiquitination in animal models currently.

Actually, we tried to monitor the KLF4 ubiquitination in AAA tissues. However the result was unsatisfied: after immunoprecipitation with anti-KLF4, few Ub-smear in immunoprecipitation extracts was detectable in PVDF membrane. We think there are two reasons: a) the amount of living cells is not enough in local aorta tissues due to the characteristic of AAA per se (originally, aorta is not a tissue with abundant amount of protein like heart or liver, and aneurysm aorta is a necrotic tissue with much less protein amount); b) the ubiquitinated KLF4 in vivo is adequate to be detectable only if the MG132 inhibits its degradation and maintained the KLF4-ub smear status. We are really sorry about the failure of in vivo KLF4 ubiquitination detection, and in fact we very much want to acquire the result of KLF4 ubiquitination in vivo. We hope there will be technique breakthroughs in ubiquitination detection in aneurysm tissues in the future.

(3) Concerning the “the KLF4 cytoplasmic/nuclear localization altered in VSMC by FAM3A in vivo”

Consulting your suggestion in Comment 13, we collected nuclear extracts in AAA tissues, and then performed immunoblot to detect total- and phosphorylated-KLF4.

This result we presented is from gross AAA tissues, not VSMC-specific. Actually, to acquire VSMCs specifically in AAA tissues, there were methods: using fluorescence-activated cell sorting (FACS) technique by flow cytometry or magnetic-activated cell sorting (MACS) to sort and collected VSMCs in AAA tissues and then obtaining nuclear or cytoplasmic extracts, followed by immunoblot for KLF4. However based on AAA pathological features, there are not so much living cells in AAA tissue. For VSMCs part, it is too few to obtain adequate protein amount from VSMCs to perform immunoblot. Moreover, it is interesting that the cytoplasm extract obtained from AAA tissues was much less than nucleus extract and therefore the results of KLF4 in cytoplasm extract have poor reproducibility and is unsatisfied. So we provided the result of nuclear localization of KLF4 in vivo.

Solutions:

- Please review Figs. 5c and 8f: We presented the KLF4 nuclear localization altered by FAM3A as well as TGF β /SMAD3 inhibitor SIS3 using immunoblot and nuclear extracts from AAA tissues.

5) The authors propose that FAM3A suppresses ROS. Is this evident in the experimental models of AAA after overexpression/supplement of FAM3A? How is ROS production related to TGF β /SMAD3-Klf4 axis? Can be another technique in addition to immunofluorescence be used to assess ROS?

Response: Thank you very much for your comments.

1) Concerning “Is this evident in the experimental models of AAA after overexpression/supplement of FAM3A?”

Certainly, the ROS level was suppressed in the experimental models of AAA after FAM3A_{overexpression}. In the revised paper, we presented these data.

Solutions:

- Please review Fig. 8e: The ROS level was detected with Dihydroethidium and shown in the experimental models of AAA after FAM3A overexpression with or without TGF β /SMAD3 inhibitor SIS3 or KLF4 inducer APTO-253.

2) Concerning “How is ROS production related to TGF β /SMAD3-Klf4 axis?”

TGF β /SMAD3-KLF4 axis plays prominent roles in cell growth and differentiation in many tissues as well as inflammatory processes. Abnormal ROS production is implicated in many pathological situations and diseases. In our revised paper, we further detected

the impact of TGF β /SMAD3-KLF4 axis in ROS production level in vivo in the context of FAM3A overexpression. It seemed that suppression of TGF β /SMAD3 abolished anti-oxidative stress of FAM3A function (please review Fig. 8e).

3) Concerning “Can be another technique in addition to immunofluorescence be used to assess ROS?”

Certainly, ROS can be detected by some different techniques. In the revised paper, Dihydroethidium (DHE) was used to assess ROS level in the experimental models of AAA after overexpression of FAM3A. Please review Method section of “*Reactive oxygen species (ROS) assay*”.

6) It is unclear why the authors have chosen Klf4 and not another transcription factor from the transcriptome? Is there a rationale beyond published literature on the functional role of Klf4 in VSMC in AAA?

Response: Thanks for your comments.

At first we found that FAM3A decreased in AAA patients and murine model and supplement of FAM3A showed a protective effect on AAA mice. We tried to discover the crucial mediators, and thus tested an array of factors which have been well-established in regulating phenotype switching in VSMCs, such as HDAC9, AHR, DKK3, TCF21, TGF β , and KLF4 (PMIDs: 30385745, 34470477, 29284609, 32441123, etc.). Among them, TGF β and KLF4 were significant due to their stable results. Accordingly, we have chosen KLF4 as a marker in the context of protective function of FAM3A in AAA.

Specific comments:

7. Figure 1A: typo: it is “Masson” not “Mason”

Response: Thank you very much for your careful review. We feel sorry for mistake like this. We have corrected as “Masson” in the Fig. 1a.

8. Figure 1E: Please include the proper labeling of the Y-axis. Does it start at 0? It only shows 120.

Response: Thank you very much for your careful review. Maybe, there was an error of image input. We have added the full labeling of the Y-axis. Please review Fig. 1e.

9. Figure 1G: The images of the aorta show less dilated aorta in elastase when compared to saline-treated aorta. This is not consistent with the quantification.

Response: Thank you very much for your careful review. We have presented proper images and indicated the dilation position. Please review Fig. 1g. Additionally, it should be mentioned that the researchers choose the infrarenal aorta rather than the upper part of abdominal aorta in elastase AAA model to avoid the liver being damaged by elastase.

10. Figure 4B/Suppl. Figure IV:

Are the markers of VSMC differentiation, e.g., CD68, ARG1, Aggrecan, regulated in the experimental models of AAA at baseline (without treatment).

Response: Thanks you for your review. Actually, the markers of VSMC differentiation which were detected in our study such as CD68, Aggrecan, OPN etc. was generally acknowledged with a close relationship with AAA. Their expressions were altered in murine AAA models in compared with normal control. Lots of literatures have documented their associations with AAA (such as PMIDs: 32243809, 34470477, 36552822, 34698377).

11? **Response:** Maybe there is a problem about the serial number.

12. Results, Line 332: “It is well accepted that Klf4 is a novel reprogramming factor”. The references are missing.

Response: Thank you for your careful review. We have added the Reference 15 (Title: *Six Shades of Vascular Smooth Muscle Cells Illuminated by KLF4 (Kruppel-Like Factor 4)*) Reference 16 (Title: *KLF4 regulates abdominal aortic aneurysm morphology and deletion attenuates aneurysm formation*), and Reference 17 (Title: *EGR1 and KLF4 as Diagnostic Markers for Abdominal Aortic Aneurysm and Associated With Immune Infiltration*) in this place. Please review Results section of “*FAM3A influences KLF4 post-translational modification*”, paragraph1, line2.

13. Figure 5: Klf4 protein abundance has been measured in the experimental models and in human AAA samples. However, Klf4 nuclear localization and phosphorylation was only assessed in vitro. Is there evidence of Klf4 nuclear-cytoplasmatic shift in the in vivo models? For in vivo and/or in

vitro, nuclear and cytoplasmic extracts, followed by immunoblot for Klf4 would strengthen the hypothesis that FAM3A regulates nuclear-cytoplasmic of Klf4 in VSMC.

Response: Thank you very much for your nice suggestion. The method you suggested of obtaining nuclear and cytoplasmic extracts, followed by immunoblot for Klf4, is very useful.

We acquired nuclear extracts from entire murine AAA tissues to do immunoblot.

Solutions:

- Please review Figs. 5c and 8f: In the revised paper, the in vivo nuclear KLF4 localization was presented in Figs. 5c and 8f. And the results were described in the Results section of “FAM3A influences KLF4 post-translational modification” paragraph2, line7 (Fig. 5c), and “The signaling pathways involved in regulatory effect of FAM3A on VSMC differentiation reprogramming” paragraph5, line4 (Fig. 8f).

14. In the summary (Figure8), the term “synthetic VSMC” needs further explanation. I am uncertain what the authors mean.

Response: Thanks very much for your careful review. In terms of synthetic VSMC, what we meant was that the VSMC was characterized by downregulation of contractile proteins and increase in proliferation and transdifferentiation proteins.

Actually, this expression is confusing. We have made a change as “*De-differentiated VSMCs*” in the revised version. Please review Supplementary Fig. 11.

15. In material and method section:

- Does the animal approval number JS-2629 apply for all animal studies in the manuscript or only to the Pancreatic elastase-induced AAA Model?
- Where are the ApoE transgenic mice obtained from?

Response: Thank you for your careful review.

- The animal approval number JS-2629 applies for all murine AAA model in our study.
- The ApoE^{-/-} mice were commercially obtained from Beijing Huafukang Bioscience Co., Ltd, China. We presented this information in Methods section of “*Mouse abdominal aortic aneurysm model/AngII (Angiotensin II) infusion in ApoE^{-/-} (apolipoprotein E) mice*” line2.

16. Were only male mice studied in the project? If so, what is the reason? Are there sex specific differences in the AAA model?

Response: Thank you very much for your careful review.

We also detected the FAM3A function on female AAA mice. It seemed that there were no significant sex specific differences in the murine AAA model. In the revised paper, we presented the protective functions of FAM3A in AAA model established by female mice. Please review Supplementary Fig. 3.

Thanks again for your careful review.

REVIEWER COMMENTS

Reviewer #1 (Remarks to the Author):

This study by Lei et al investigates the role of FAM3A in smooth muscle cell plasticity to differentiate into other cell types during vascular remodeling in AAAs. Some of the concerns were addressed but several other prominent issues still remain to be deciphered, as summarized below:

1. The primary question is whether FAM3A prevents aneurysm formation via a direct effect on SMCs *in vivo* or is it more systemic effect due to paracrine effects on other cells. This will be conclusively achieved by SMC-specific deletion of FAM3A, Klf4 and TGF- β (as described below) rather than recombinant protein administration (although essential for subsequent confirmation and therapeutic application).

2. Previous studies have reported SMC plasticity using SMC-specific conditional Klf4 knockout mice (PMID:25985364 and 24030402). To specifically determine the SMC plasticity via FAM3A and Klf4 pathway, using a cell-specific deletion approach is required.

3. Previous studies have elucidated the role of Tgfb1/b2 signaling in modulating SMC plasticity as, loss of the TGF- β signaling pathway impairs the contractile apparatus of vascular smooth muscle, damages elastin, increases the number of macrophage markers, and increases cell proliferation and matrix accumulation in tamoxifen-induced TGFBR2 SMC knockout mice (PMID: 24401272, 35517795 and 26494233). The role of recombinant FAM3A in modulating SMC-Tgfb pathways should be shown by using cell specific deletion of Tgfb1/b2 to specifically delineate this signaling mechanism. The use of Tgf- β inhibitors or KLF4 inducers does not specify the role of FAM3A in SMC plasticity and requires a more specific and rigorous approach.

4. A recent study showed that vascular smooth muscle cell-specific deletion of FAM3A attenuated Ang II-promoted vascular remodeling (PMID: 32279581). Using VSMC-specific FAM3A^{-/-} mice, these results show that FAM3A regulates blood pressure and that FAM3A expression is increased in hypertensive rodents. These results are counter intuitive to the data presented in the current study, which relies heavily on recombinant proteins (non-cell specific) to derive the conclusions. Therefore due to this dichotomy, a cell specific deletion of FAM3A in SMCs for the *in vivo* studies is imperative for a comparative analysis in this aortopathy.

5. A major drawback of this study is the lack of FAM3A-dependent receptor identification. The authors eluded to FGFR as one of the targets which should be explored further in this model. Moreover, PPAR γ (peroxisome proliferator-activated receptor γ) has been demonstrated to activate FAM3A transcription and antagonism of PPAR γ has been shown to inhibit Ang II-induced change in FAM3A expression (PMID: 32279581 and 23562554). Therefore, this study must identify FAM3A-dependent ligand-receptor interactions for a meaningful interpretation of the proposed hypothesis.

6. Clinical application of this study will be relevant only if preformed AAAs can be attenuated by recombinant FAM3A which should be strongly considered.

7. The flow cytometry data shown in the results requires more details regarding the complete gating strategies used for the experiment. Please show the schematic flow panel from beginning to subsequent gates till the final analysis as shown in Figs. 4b and 8d. Please show the baseline saline controls for ApoE^{-/-} for the percentage of SMCs that co-express CD68⁺, RUNX2⁺ and CD34⁺ (Fig. 8d), as well as LUM⁺ and ADIPQ⁺ in Fig. 4b. Since the percentages of SMCs undergoing differentiation appear to be fractional, the appropriate controls (as above) and results

8. What was the significance characteristics used for p value determination which is missing in statistical analysis methodology? Was a power calculation used for the human and murine tissue analysis? How many times were the experiments repeated for ensuring rigor and reproducibility (cumulative raw data should be shown in supplemental information)?

9. Were the blood pressure differences significant in Figs. 3b and 1e (not shown)?

10. Please show the densitometric quantification of western blot analysis in Fig. 8b.

11. Why is the Klf4 mRNA expression different than protein expression on sequential analysis (Suppl. Fig. 8)? The AngII model describes Klf2 in the ApoE^{-/-}/AngII model; what were the results for Klf4 in this model?

Reviewer #2 (Remarks to the Author):

By an impressive amount of additional data provided in the revised manuscript, the authors have comprehensively addressed the list of major and minor concerns specified in my initial review, in particular regarding

- the cell-type specific expression of FAM3A and the potential contributions by cell types other than VSMCs to FAM3A effects on AAA
- off-target expression and effects of FAM3A in the mouse models
- concentration and stability of hrFAM3A achieved in the in vitro versus in vivo experiments
- SMC transdifferentiation as documented at the cellular level.

Considering the fact that the FAM3A receptor is currently unknown and an SMC-specific FAM3A receptor knock-out mouse is not available, the authors have provided feasible evidence to sustain their conclusion that FAM3A effects on VSMCs via KLF-4 regulation are likely to explain (at least in part) the protective impact of FAM3a against AAA formation.

While the novel data have provided essential evidence to support the claims and conclusions of the authors, they have also resulted in new text passages and manuscript rearrangements which would require professional language editing for a consistently high level of scientific writing/grammar.

Furthermore, the newly introduced passages have raised the following (minor) concerns:

1. On page 4, line 108 the authors state that „Using scRNA transcript sequencing data in murine models and secreted protein level in the medium of cultured primary cells, it seemed that FAM3A was produced abundantly in VSMCs and endothelial cells, and a few in fibroblasts and macrophages (Supplementary Fig. 1c, d).”

I do not entirely agree with this conclusion based on the presented data. While ECs are certainly less abundant than the other vascular cell types in the in vivo samples, their normalized FAM3A expression frequency/levels (Supplementary Fig. 1c, right panel) seems minor and unaltered in AAA compared to fibroblasts and SMCs. Yet, in vitro ECs exhibited more secretion and binding of FAM3A than fibroblasts. Due to the design of the binding assay for hrFAM3A to cell cultures, I think it is difficult to deduce differences in binding affinity/capacity between cell types. While I assume that a comparable number of cells were seeded, the two hours of incubation (at 37°C or 40°C? – would be helpful to add this information to the Suppl. Methods) with hrFAM3A might not only lead to surface binding but also to ligand internalization in the process of receptor binding/signaling resulting in less surface-bound signal. Thus, I would not entirely discard the contribution by fibroblast-derived FAM3A in aortas/AAA based on the in vitro results, as the scRNA data argue in favor of FAM3A expression by fibroblasts.

2. The authors state on page 6, line 156: “However, KLF4, a crucial regulator of VSMC phenotypic switching, was not significantly altered at the transcriptional level after FAM3A overexpression at 4-week timepoint in AngII-infused ApoE^{-/-} AAA mice (Fig. 2h).” Yet, the respective plot in Fig. 2h specifies a highly significant p-value ($p=0.0002$) which is not supported by the graph. Please correct.

3. The paragraph on female ApoE KO mice treated with hrFAM3A (page 6, line 159) was inserted in the chapter on adenoviral overexpression of hrFAM3A and would better fit to the subsequent chapter that is devoted to the administration of hrFAM3A.

4. Regarding the cytotoxicity tests (page 8, line 220) with hrFAM3A, I am not entirely sure what is meant by “The results showed that there was no significant toxicity of FAM3A in VSMCs, endothelial cells, fibroblasts, and macrophages in the baseline (Supplementary Fig. 4f).” Does “in the baseline” refer to the absence of other stimuli in this in vitro experiment?

5. Suppl. Fig. 4e: Please specify whether toxicity tests were performed on mouse organs harvested after single hrFAM3A treatment (as for immunofluorescence stainings) or on organs harvested after a complete treatment cycle (as for measurements of aortic diameter).

6. Suppl. Fig. 4f: Please specify in the legend for how long cell cultures were treated with hrFAM3A and correct the labeling of cell types within the graph (erroneously all labeled “VSCM”).

Reviewer #3 (Remarks to the Author):

The authors have performed a considerable amount of additional experiments and addressed the concerns properly. I have no further comments.

REVIEWER COMMENTS

Reviewer #1 (Remarks to the Author):

This study by Lei et al investigates the role of FAM3A in smooth muscle cell plasticity to differentiate into other cell types during vascular remodeling in AAAs. Some of the concerns were addressed but several other prominent issues still remain to be deciphered, as summarized below:

Response in general: Thank you very much. I wish to express my gratefulness for your careful review of our paper once again. The biggest issue is about the FAM3A transgene/deletion specifically in SMCs to make a SMC-specific exploration. Indeed, the SMC-specific investigation is valuable. However, FAM3A is a secreted protein and its receptor is unclear currently. It is hard to study FAM3A function specifically in SMCs excluding other cell types by using SMC-specific FAM3A knockout mice (the FAM3A produced and secreted by other cell sources could still influence SMCs). The detailed evidence has been presented in responses to Question1 and Question4. Furthermore, please review the responses to your concerns which are listed point by point below.

1. The primary question is whether FAM3A prevents aneurysm formation via a direct effect on SMCs in vivo or is it more systemic effect due to paracrine effects on other cells. This will be conclusively achieved by SMC-specific deletion of FAM3A, Klf4 and TGF- β (as described below) rather than recombinant protein administration (although essential for subsequent confirmation and therapeutic application).

Response: Thank you very much for your careful review and kind advice.

We appreciate that these issues are key points.

- To determine the effect of FAM3A on SMC phenotype (rather than on other cells) in vivo is crucial. Except for in vitro results, our findings from flow cytometry (detecting SMC-specifically, Figs. 4b and 8d) and immunofluorescence co-staining (detecting co-staining of SMC marker α SMA and other intermediate cell markers, Fig. 4c) suggested that SMC differentiation in vivo was altered by systematical administration of FAM3A.

- To determine the direct effect of FAM3A (targeting directly on SMCs, not mediated by other cells) on SMC phenotype in vivo is also crucial. To complete the cell-specific exploration, cell-specific deletion of the FAM3A receptor is needed. However, the receptor of FAM3A is unclear currently.

1) Concerning “whether FAM3A prevents aneurysm formation via a direct effect on SMCs in vivo or is it more systemic effect due to paracrine effects on other cells.”

FAM3A is a secreted cytokine. Because of the lack of knowledge of the FAM3A receptor, we first systematically explored FAM3A function by treating murine AAA models via systematic administration. In addition, considering the autocrine, paracrine and endocrine manners, there are two manners of the systematic administration of exogenous FAM3A:

① recombinant FAM3A: by tail vein injection, recombinant FAM3A was transferred following circulation to target cells (including SMCs) (like in a manner of endocrine). In this case, FAM3A can directly influence SMCs.

② adenovirus mediated FAM3A (Ad-FAM3A) overexpression: by tail vein injection, Ad-FAM3A infected cells. The infected cells (including SMCs) overexpressed FAM3A and secreted FAM3A which influenced SMCs in manners of endocrine, paracrine, as well as autocrine. In this case, FAM3A can influence SMCs both directly and systemically.

2) Concerning “SMC-specific deletion of FAM3A” (SMC-specific deletion of Klf4 and TGF β are replied at Question2 and 3)

There is some important information: ① FAM3A is a secreted cytokine (relevant evidence has been presented in response 2) ② to Question4); ② FAM3A is not produced and secreted by SMCs uniquely, and it is also expressed by other cell types. Under these cases, SMC-specific FAM3A deletion could not block the effects of FAM3A on SMCs in vivo.

If we use SMC-specific FAM3A^{-/-} mice, the FAM3A produced and secreted by other cell sources could still influence SMCs.

Certainly, SMC-specific deletion of FAM3A is a good way to study the FAM3A **autocrine** effects on SMCs.

Solutions:

- We make a discussion about these key points concerning FAM3A cellular production and function manners in SMCs. Please review Discussion section, paragraph3 (line4), paragraph4 (line8), paragraph7.

2. Previous studies have reported SMC plasticity using SMC-specific conditional Klf4 knockout mice (PMID:25985364 and 24030402). To specifically determine the SMC plasticity via FAM3A and Klf4 pathway, using a cell-specific deletion approach is required.

Response: Thank you very much for your kind advice.

In our findings, FAM3A suppressed KLF4 protein level. To determine the FAM3A and Klf4 pathway in SMC plasticity in vivo, theoretically SMC-specific KLF4 transgene (overexpression) mice are preferred. However, we really fell sorry that it is hard to complete these animal experiments due to the lack of SMC-specific KLF4 genetic mice in our laboratory currently.

In the Discussion section, these crucial publications have been cited to make a more in-depth correlation with our results: deletion of KLF4 maintained SMA and suppressed numbers of SMC-derived MSC-like, and macrophage-like cells (PMID:25985364, 24030402 as **reference 43 and 17** in our manuscript).

Solutions:

- The **in vitro** experiments showed the relevant results

Please review Figs. 5d-f and 6, Supplementary Fig 9: these results showed that FAM3A suppressed KLF4 expression as well as phosphorylation and nuclear localization, and induced KLF4 ubiquitination. KLF4 could abolish FAM3A's suppressive effect on SMC transdifferentiation using cultured primary aortic SMCs. These in vitro results partly indicated a FAM3A/KLF4 axis.

3. Previous studies have elucidated the role of Tgfb1/b2 signaling in modulating SMC plasticity as, loss of the TGF- β signaling pathway impairs the contractile apparatus of vascular smooth muscle, damages elastin, increases the number of macrophage markers, and increases cell proliferation and matrix accumulation in tamoxifen-induced TGFBR2 SMC knockout mice (PMID: 24401272, 35517795 and 26494233). The role of recombinant FAM3A in modulating SMC-Tgfb pathways should be shown by using cell specific deletion of Tgfb1/b2 to specifically delineate this signaling mechanism. The use of Tgf-b inhibitors or KLF4 inducers does not specify the role of FAM3A in SMC plasticity and requires a more specific and rigorous approach.

Response: Thanks very much for your kind advice. The TGF- β and KLF4 has been evidenced in regulating AAA in previous reports. From another perspective, in our study, we propose FAM3A as a novel regulator for TGF- β /KLF4.

However, we feel sorry again for not being able to complete these animal experiments due to the lack of SMC-specific TGF β receptor-deficient mice under our laboratory conditions currently. Like FAM3A, TGF β is a cytokine. To observe SMC-specific TGF β 's effects, the researchers choose SMC-specific TGF β receptor knockout mice (PMID: 24401272, 35517795 and 26494233). And we discuss the limitations of non cell specificity and lack of receptor in Discussion section in paragraph7.

Solutions:

- The in vitro experiments showed the relevant results

Please review Fig. 7: these results showed that FAM3A upregulated TGF β expression in SMCs. Inhibition of TGF β /smad3 pathway by SIS3 abolished FAM3A's effect on regulating SMC plasticity and KLF4 ubiquitination through cultured primary aortic SMCs.

Not the preferred approach, these in vitro results partly support the FAM3A/TGF β /KLF4 axis.

- Make a discussion: the limitations of non cell specificity and lack of receptor were discussed in Discussion section in paragraph 7.

4. A recent study showed that vascular smooth muscle cell-specific deletion of FAM3A attenuated Ang II-promoted vascular remodeling (PMID: 32279581). Using VSMC-specific FAM3A^{-/-} mice, these results show that FAM3A regulates blood pressure and that FAM3A expression is increased in hypertensive rodents. These results are counter intuitive to the data presented in the current study, which relies heavily on recombinant proteins (non-cell specific) to derive the conclusions. Therefore due to this dichotomy, a cell specific deletion of FAM3A in SMCs for the in vivo studies is imperative for a comparative analysis in this aortopathy.

Response: Thank you very much for your careful review. This reference *PMID: 32279581* and the reference you mentioned in Question 5 (*PMID: 23562554*) are from the same laboratory and they proposed FAM3A as a mitochondrial protein. We recognized FAM3A as a secreted peptide.

Whether or not FAM3A is secreted is crucial for explaining the discrepant results and for our experimental design (whether or not using of VSMC-specific FAM3A deletion animal).

1) Concerning “cell specific deletion of FAM3A in SMCs”: SMC-specific FAM3A deletion could not block the effects of FAM3A on SMCs in vivo.

Again as described in Question 1, there is some important information: ① FAM3A is a secreted cytokine; ② FAM3A was produced and secreted by SMCs as well as other cell types. Under these cases, SMC-specific FAM3A deletion could not block the effects of FAM3A on SMCs in vivo.

If we use SMC-specific FAM3A^{-/-} mice, the FAM3A from other cell sources could still influence SMCs, such as endothelial cells which could also produce abundant FAM3A (PMID: 31000420, 12160727), and reshape SMCs (in manners of paracrine as well as endocrine).

Certainly, SMC-specific deletion of FAM3A is a good way to study the FAM3A **autocrine** effects on SMCs.

2) Concerning the discrepancy from “PMID: 32279581”

① Mitochondria-located vs. secreted: Their team (*PMID: 32279581*) proposed that FAM3A was located in mitochondria (*PMID: 24857820*). It was important for the

subsequent study as well as Experimental Design. Based on mitochondria-localization of FAM3A, it was logical that they used VSMC-specific FAM3A^{-/-} mice. No FAM3A could influence VSMCs any more, since FAM3A in other cells was spatially restricted intracellularly and could not interfere in VSMCs.

② The evidence: FAM3A is secreted

a) Nature communications. 2019;10(1):1796 (PMID:30996264)

In this publication, David Sala et al. provided experimental evidence and denied the viewpoint of mitochondria-localization of FAM3A.

Fam3a is secreted by myogenic cells. Fam3a protein sequence analysis using LocTree3⁴³ predicted that Fam3a is a secreted protein (Supplementary Fig. 6a). However, previous reports indicated a localization of Fam3a in the mitochondria²⁵⁻²⁷. To strengthen our prediction, we used two additional softwares that predict protein localization: TargetP⁴⁴ and MitoFates⁴⁵. TargetP is a software that predicts protein localization by analyzing the presence of N-terminal presequences containing mitochondrial targeting peptides or secretory SPs⁴⁴. TargetP analysis of the Fam3a protein sequence also predicted that Fam3a is a secreted protein, while it correctly predicted that the protein Citrate Synthase was mitochondrial (Supplementary Fig. 6a). Finally, we used MitoFates as software that analyzes the presence of N-terminal mitochondrial targeting signals and their cleavage sites⁴⁵. Consistent with the previous predictions, Fam3a did not contain any mitochondrial localization presequence while it was identified in Citrate Synthase (Supplementary Fig. 6b). Overall, the analysis of the Fam3a protein sequence using three different softwares indicated that Fam3a is a secreted protein that lacks a mitochondrial localization signal.

To directly investigate the subcellular localization of Fam3a in muscle cells, we performed colocalization studies in transiently transfected C2C12 myoblasts with a construct expressing Fam3a-Myc-Flag. In order to label Golgi, endoplasmic reticulum (ER), and mitochondria, we used the markers GM-130, KDEL, and Tomm20, respectively. In accordance with previously published studies, these markers were specific for each compartment as they presented no or minimal colocalization among them (Supplementary Fig. 6c, d)⁴⁶⁻⁴⁹. Consistent with our Fam3a localization predictions, we did not detect colocalization between exogenous Fam3a and mitochondria 48 h after transfection (Supplementary Fig. 6d, e). Instead, exogenous Fam3a colocalized with GM-130, marker of cis-Golgi (Fig. 5a and Supplementary Fig. 6d). Upon treatment with the secretion inhibitor monensin, Fam3a-Myc-Flag strongly colocalized with the ER, as shown by colocalization with KDEL (Fig. 5b and Supplementary Fig. 6d). Western blot analysis detected accumulation of Fam3a-Myc-Flag within C2C12 myoblasts upon monensin treatment (Fig. 5c, d). Finally, we observed the presence of Fam3a-Myc-Flag in the media of transfected cells (Fig. 5e). Together, these findings demonstrate that Fam3a is secreted by myogenic cells.

25. Jia, S. et al. FAM3A promotes vascular smooth muscle cell proliferation and migration and exacerbates neointima formation in rat artery after balloon injury. *J. Mol. Cell. Cardiol.* 74, 173–182 (2014).
26. Wang, C. et al. FAM3A activates PI3K p110alpha/Akt signaling to ameliorate hepatic gluconeogenesis and lipogenesis. *Hepatology* 59, 1779–1790 (2014).
27. Song, Q., Gou, W. L. & Zhang, R. FAM3A protects HT22 cells against hydrogen peroxide-induced oxidative stress through activation of PI3K/Akt but not MEK/ERK pathway. *Cell. Physiol. Biochem.* 37, 1431–1441 (2015).

They presented in silico data and immunofluorescence images indicating FAM3A was localized at vesicular transport system (endoplasmic reticulum, golgi apparatus, and cell membrane) instead of mitochondria.

b) Our evidence about FAM3A being secreted

We believed that FAM3A was a cytokine which was secreted by cells and not restricted intracellularly.

- FAM3A has **signal peptide** which is a marker for secreted protein. Please review the chart below which indicates that human and mouse fam3a does carry signal peptide sequence:

HUMAN: hFAM3A [NM_021806] [P98173]
 MOUSE: mFam3A [NM_001379181.1] [Q9D8T0]

- FAM3A was detectable in the cell culture medium by ELISAs. Please review Supplementary Fig 1g. FAM3A was found in culture medium of VSMCs, endothelial cells, and fibroblasts.

③ Concerning the PMID: 32279581 “FAM3A expression is increased in hypertensive rodents”

In our study, FAM3A expression was decreased in aneurysm tissues. However, another result from their team in 2014 (*PMID:24857820*) also showed that FAM3A expression was reduced in injured rodent arteries (Highlights section). There is some accordance between our findings.

Moreover, using recently published microarray datasets with large samples (sample size of AAA patients >10) from GEO public database (GSE47472, GSE57691), we also found that FAM3A mRNA was decreased in aortic tissues from aneurysm patients (please review Supplementary Fig. 1a).

④ Alternative splicing

However, it should be noted that *fam3a* gene have a large number of splice variants. In Ensembl database, human FAM3A has more than twenty transcripts. Maybe it is an explanation of discrepancy between their reports and our findings. Is the FAM3A in their publications the same one in our study?

3) Concerning the use of “recombinant proteins of FAM3A (non-cell specific)”

In general, peptide drug is valued for its safety in therapeutic application. Considering these significance of peptide drug as well as healthcare products, we used recombinant human FAM3A as well as the adenovirus FAM3A overexpression (the product of which was secreted peptide) to treat AAA animal in our study for some clue for potential utilization in future.

Solutions:

1) Make a discussion

Further, we have cited these references and made a discussion about these controversial issues in our paper. Please review Discussion section, paragraph 2, line3-15.

2) The specific information for the FAM3A we used was fully provided in the paper

We presented the FAM3A gene information with the Adenovirus over-expression (**mFam3A [NM_001379181.1]**) and human FAM3A recombinant protein (**hFAM3A [NM_021806]**) we used in the Method section. Moreover, the manufacturer and product number of human FAM3A recombinant protein were provided in the major resources in Supplementary information (Supplementary Table 7), through which the amino acid sequences of FAM3A could be traced accurately.

Secretory proteins as well as cytokines typically have a wide range of effects, affecting many cells, unlike the way structural proteins function locally. Therefore, in our study we first make a comprehensive and systematic (non-cell specific) evaluation to assess the role of FAM3A in AAA.

Certainly, to elucidate VSMC-specific mechanism is indeed necessary (including receptor). It must be our subsequent exploring work.

5. A major drawback of this study is the lack of FAM3A-dependent receptor identification. The authors eluded to FGFR as one of the targets which should be explored further in this model. Moreover, PPAR γ (peroxisome proliferator-activated receptor γ) has been demonstrated to activate FAM3A transcription and antagonism of PPAR γ has been shown to inhibit Ang II-induced change in FAM3A expression (PMID: 32279581 and 23562554). Therefore, this study must identify FAM3A-dependent ligand-receptor interactions for a meaningful interpretation of the proposed hypothesis.

Response: Thank you very much for your careful review.

1) Concerning “the lack of FAM3A-dependent receptor identification” and “FGFR”

The FAM3A receptor is an important issue. So far, most receptors of the FAM3 family members have not been identified, including FAM3A. Recently, the receptor of FAM3B was identified as FGFR (*FAM3B receptor FGFR, Proc Natl Acad Sci U S A. 2021 May 18;118(20):e2100342118. doi: 10.1073/pnas.2100342118. PMID: 33975953*).

We detected whether or not FGFR was also FAM3A receptor because FAM3A have homology to FAM3B. However, we found that FGFR was not a receptor of FAM3A. So we did not further explore correlation between FGFR and FAM3A.

Identification of a cytokine's receptor is a kind of complex work: obtaining membrane protein complex containing FAM3A, separating the compounds, analyzing the proteins, and identifying which protein is the potential receptor by using mass spectrometer; subsequently, detecting which potential candidates could interact directly with FAM3A by yeast two hybrid system or co-IP technologies. And finally, the candidate receptor is confirmed by inhibiting the candidate receptor to detect whether or not FAM3A still work. As you know, these experiments are technically difficult and time-consuming. Certainly, screening FAM3A receptor must be an important work in our subsequent exploration.

2) Concerning “PPAR γ (peroxisome proliferator-activated receptor γ) has been demonstrated to activate FAM3A transcription (PMID: 32279581 and 23562554)”

It is suggested that PPAR γ upstream regulated FAM3A. Accordingly, the FAM3A interacting downstream receptor is not involved in or relevant to this finding.

In our laboratory, we also found that PPAR γ agonist (pioglitazone, 20uM, 12h) or inhibitor (SR1664, 20uM, 12h) induced or suppressed FAM3A protein level. Please see chart below:

Really, numerous research reported a protective role of PPAR γ agonist in cardiovascular disease (such as PMID: 26213347: PPAR- γ agonist attenuates inflammation in aortic aneurysm patients; PMID: 27855608: Peroxisome Proliferator-Activated Receptor (PPAR) Gamma Agonists as Therapeutic Agents for Cardiovascular Disorders: Focus on Atherosclerosis). Indirectly, our findings are in accordance with these publications, since supplementation of FAM3A attenuated aneurysm formation, like PPAR γ agonist. Maybe PPAR γ agonist protect blood vessel partly via induction of FAM3A.

In our article, we did not report and discuss PPAR γ , and we emphatically elucidated FAM3A **downstream** TGF β and KLF4 ubiquitination, in the current paper. However, PPAR γ as the **upstream** of FAM3A, is worth study next.

3) Concerning “a meaningful interpretation of proposed hypothesis”

Is “proposed hypothesis” you mentioned that FAM3A is secreted protein and work through receptor, please? If so, we are truly sorry about the absence of FAM3A receptor or mediator this time. We made an effort to improve the elucidation of protective function of FAM3A in aneurysm in this paper.

We appreciate that identifying FAM3A receptor or mediator in mechanism is an important work.

Solutions:

We stated and discussed the lack of FAM3A receptor in the revised paper. Please review Discussion section for line17 in paragraph2, line8 in paragraph4, and line3, 7 in paragraph7.

6. Clinical application of this study will be relevant only if preformed AAAs can be attenuated by recombinant FAM3A which should be strongly considered.

Response: Thank you very much for your pertinent advice.

We thought there were two key findings relevant to clinical potential application in this basic medical research: 1) plasma FAM3A level was decreased; 2) FAM3A inhibited AAA formation process (not preformed AAA).

- The clinical significance of “1) plasma FAM3A level was decreased”

By monitoring plasma FAM3A levels, a potential risk of AAA would be indicated for certain populations with low plasma FAM3A levels.

- The clinical significance of “2) FAM3A inhibited AAA formation process”

In populations at high-risk of AAA, supplementation with FAM3A would prevent or protect against AAA formation.

- FAM3A would be a potential AAA-related gene.

Screening its familial hereditary features as well as gene polymorphism would be meaningful.

Monitoring and prevention are important, since most disease is an irreversible process. However, the present study is more basic research than clinical. It provides potential clues for clinical applications.

On the other hand, another study to explore the therapeutic effects of FAM3A on preformed AAA using preformed AAA mice would be very interesting.

Solutions:

To make a more objective view of FAM3A decreasing in AAA disease, we further provided analytic results of public microarray datasets from AAA patients with large sample size (as also stated in response 2)/③ to Question4). The results indicated that FAM3A mRNA was decreased in aortic tissues from aneurysm patients (Supplementary Fig. 1a).

7. The flow cytometry data shown in the results requires more details regarding the complete gating strategies used for the experiment. Please show the schematic flow panel from beginning to subsequent gates till the final analysis as shown in Figs. 4b and 8d. Please show the baseline saline controls for ApoE^{-/-} for the percentage of SMCs that co-express CD68⁺, RUNX2⁺ and CD34⁺ (Fig. 8d), as well as LUM⁺ and ADIPQ⁺ in Fig. 4b. Since the percentages of SMCs undergoing differentiation appear to be fractional, the appropriate controls (as above) and results

Response: Thanks very much for your careful review and pertinent advice. The complete gating strategy for the flow cytometry (from beginning to subsequent gates till the final analysis) was shown in the Supplementary Fig. 11b,c.

Furthermore, please review that the baseline saline controls for ApoE^{-/-} for the percentage of SMCs that co-express CD68⁺, RUNX2⁺, CD34⁺, LUM⁺, and ADIPQ⁺ have been shown appropriately in Figs. 4b and 8d.

8. What was the significance characteristics used for p value determination which is missing in statistical analysis methodology? Was a power calculation used for the human and murine tissue analysis? How many times were the experiments repeated for ensuring rigor and reproducibility (**cumulative** raw data should be shown in supplemental information)?

Response: Thanks for your critical and meaningful question. We are very sorry for the omission of the definition of statistical significance in the statistical methodology. In this study, two-sided *P* value less than 0.05 was defined as statistically significant, and this has been detailed in the Methods section of "*Quantification and statistical analysis*" in the revised manuscript.

We did use the power calculation to determine statistical power and calculate sample size before conducting the experiments on human and mouse tissues. However, via the power calculation we concluded that the required sample size was much larger than that the experimental conditions could accommodate, such as >80 samples per group for detection of a protein expression. Therefore, the sample size in our study was chosen based on literature and variability observed in the previous experience in our laboratory. We expanded the sample size as much as possible to obtain more rigorous conclusions, and the experimental results of different batches in this study were standardized and normalized to minimize the batch deviations. Moreover, the raw data have been provided in a single Excel file with each figure/table in a separate sheet (file name 'Source Data').

9. Were the blood pressure differences significant in Figs. 3b and 1e (not shown)?

Response: Thank you very much for your careful review. There were significant differences between ApoE^{-/-} mice treated with saline and AngII. We have labeled the differences with asterisk (*) in Figs. 1e and 3d, and indicated these differences in the legends of Figs. 1e and 3d. Please make a review.

10. Please show the densitometric quantification of western blot analysis in Fig. 8b.

Response: Thank you for your kind advice. The densitometric quantification of western blot analysis in Fig. 8b, c have been shown in Supplementary Fig. 10a, b. Please make a review.

11. Why is the Klf4 mRNA expression different than protein expression on sequential analysis (Suppl. Fig. 8)? The AngII model describes Klf2 in the ApoE^{-/-}/AngII model; what were the results for Klf4 in this model?

Response: Thank you very much for your review. In our study, we found that KLF4 mRNA level was not in accordance with its protein level. Therefore, we detected KLF4 **posttranslational modification**, and found that FAM3A induced KLF4 ubiquitination and thus suppressed KLF4 level. The KLF4 protein level in the ApoE^{-/-}/AngII model (*like KLF2 in Supplementary Fig. 8a*) was shown in Fig. 5b (left). Please make a review.

Thanks again for your careful review and nice suggestions.

Reviewer #2 (Remarks to the Author):

By an impressive amount of additional data provided in the revised manuscript, the authors have comprehensively addressed the list of major and minor concerns specified in my initial review, in particular regarding

- the cell-type specific expression of FAM3A and the potential contributions by cell types other than VSMCs to FAM3A effects on AAA
- off-target expression and effects of FAM3A in the mouse models
- concentration and stability of hrFAM3A achieved in the in vitro versus in vivo experiments
- SMC transdifferentiation as documented at the cellular level.

Considering the fact that the FAM3A receptor is currently unknown and an SMC-specific FAM3A receptor knock-out mouse is not available, the authors have provided feasible evidence to sustain their conclusion that FAM3A effects on VSMCs via KLF-4 regulation are likely to explain (at least in part) the protective impact of FAM3a against AAA formation.

While the novel data have provided essential evidence to support the claims and conclusions of the authors, they have also resulted in new text passages and manuscript rearrangements which would require professional language editing for a consistently high level of scientific writing/grammar.

Response: Thank you very much. I wish to express my gratefulness for your careful review of our paper once again. In this revision, some figures were rearranged (such as Supplementary Figs. 8-10) for a more appropriate illustration, and the manuscript has been polished by professional language editing team (Springer Nature Author Services).

Furthermore, the newly introduced passages have raised the following (minor) concerns:

1. On page 4, line 108 the authors state that “Using scRNA transcript sequencing data in murine models and secreted protein level in the medium of cultured primary cells, it seemed that FAM3A was produced abundantly in VSMCs and endothelial cells, and a few in fibroblasts and macrophages (Supplementary Fig. 1c, d).”

I do not entirely agree with this conclusion based on the presented data. While ECs are certainly less abundant than the other vascular cell types in the in vivo samples, their normalized FAM3A expression frequency/levels (Supplementary Fig. 1c, right panel) seems minor and unaltered in AAA compared to fibroblasts and SMCs. Yet, in vitro ECs exhibited more secretion and

binding of FAM3A than fibroblasts. Due the design of the binding assay for hrFAM3A to cell cultures, I think it is difficult to deduce differences in binding affinity/capacity between cell types. While I assume that a comparable number of cells were seeded, the two hours of incubation (at 37°C or 40°C? – would be helpful to add this information to the Suppl. Methods) with hrFAM3A might not only lead to surface binding but also to ligand internalization in the process of receptor binding/signaling resulting in less surface-bound signal. Thus, I would not entirely discard the contribution by fibroblast-derived FAM3A in aortas/AAA based on the *in vitro* results, as the scRNA data argue in favor of FAM3A expression by fibroblasts.

Response: Thank you very much for your careful review. We are really enlightened by your comments.

1) Concerning cell-specific FAM3A expression in aortas

The cell-specific FAM3A elucidation in aortas is a significant issue. There were several technique matters: ① scRNA sequencing is a high throughput data which is more variable in contrast with experiment of individual detection; ② the number of one kind of cells resulted from scRNA sequencing has deviation from real conditions (the cell survival rate is not identical in different kinds of cells during the digestion of aneurysm aorta, such as VSMCs with a lower survival than fibroblasts (based on our experimental experience and other team's); labeling efficiency is also different between all kinds of cells); ③ western blot is a better way to quantify the FAM3A amount in different kinds of cells. However, the aortic aneurysm is a degenerate disease with little cell amount in tissues. It is difficult to acquire adequate amount of one kind of cells *in vivo* to carry western blot (in this case, we use primary cells as our second preference).

To be more objective, we further improved these experiments in this revision. Described as below:

- Using another public scRNA sequencing dataset (GEO, under the code GSE164678) to analyze the FAM3A cell-specific mRNA expression (Supplementary Fig. 1d, e).

These two scRNA sequencing datasets we used were from different laboratory with different AAA animal models and different quality control methods leading to different cell numbers. Although the murine AAA models and acquired total cell numbers are different, the features of FAM3A mRNA levels in VSMCs and fibroblasts are similar with abundant amount. Secondary to VSMCs and fibroblasts, the FAM3A

is expressed a lot in endothelial cells. T cells expressed few FAM3A. However, it seemed that FAM3A in these two scRNA datasets did not matched up completely. In GSE164678 datasets, macrophages expressed few FAM3A. As the predominant sources of FAM3A in aortas, VSMCs and fibroblasts expressed fewer FAM3A under AAA conditions in contrast with control.

- We re-detected the FAM3A level in the culture medium and this time normalized to its cell number (by cell counting chamber) for a more objective reflection of individual cell's ability of secreting FAM3A (Supplementary Fig. 1g).

It seemed that the secreted FAM3A levels were comparable between VSMCs, endothelial cells, and fibroblasts, and macrophages secreted less FAM3A than these three kinds of cells. Moreover, pathological stimuli (PDGF and TNF α) inhibited FAM3A level in the medium in VSMCs, fibroblasts, and endothelial cells.

- We additionally carry western blot to determine the FAM3A protein level using cultured cells (Supplementary Fig. 1f).

As shown, relative to macrophages, VSMCs, endothelial cells, and fibroblasts produced abundant FAM3A protein, especially VSMCs. Pathological stimuli inhibited FAM3A production in VSMCs and fibroblasts significantly.

Taken together these results, FAM3A was produced abundantly in VSMCs, endothelial cells, as well as fibroblasts, and few in macrophages.

However, there were indeed limitations in the detection of cell-specific FAM3A protein levels, such as the absence of the AAA microenvironment.

2) Concerning the binding assay for hrFAM3A to cell cultures

Indeed, the ligand internalization was not included in the previous binding assay for hrFAM3A to cell cultures.

The previous method is *"the cells were treated for 2 hours with recombinant FAM3A which was fused with c-myc (100 ng/mL) at 37 C°, 5% CO2 incubator. Then the HRP conjugated anti-c-myc antibody was added and incubated for 1 hour. The medium was removed and cells were washed gently with warm PBS. Then cells were further soaked in PBS, DAB substrate was added, and supernate was collected. The optical density (OD) value was measured and the relative cellular affinity to FAM3A was quantified by normalizing the OD value to cell numbers and expressed as a fold change vs. VSMC-PBS group."*

In this method, only the c-myc-FAM3A in the cell surface could be detectable, and the internalized c-myc-FAM3A was precluded. We also detected the binding efficacy of 1 hours of incubation rcFAM3A with cells. However, the results showed no significant difference between different time-points in same kind of cells.

We also tried other methods to detect the cellular capture of FAM3A (intracellular and cell surface), such as cytochemistry staining. After incubation cells with c-myc-FAM3A, we fixed cells with paraformaldehyde and permeated cell membrane with permeability buffer (containing detergent). Then we detected c-myc-FAM3A using the antibody against c-myc label. However, the results were unsatisfied data with few signal detected. We speculated that ligand/receptor binding would not be a solid and stable form and the structure of ligand/receptor complex would be sensitive to paraformaldehyde fixation or other chemical stimuli.

We also tried to improve this experiment by incubating cells with c-myc-FAM3A (200 ng/mL) for 1 hour or 2 hours, and then used whole cell lysate to do western blot using antibody against c-myc label. In this case, the exogenous c-myc-conjugated FAM3A could be detected in both extracellular and intracellular locations. The results showed that FAM3A could be captured by VSMCs and fibroblasts. However, concerning endothelial cells, the c-myc-FAM3A showed at another places. We doubted that c-myc-FAM3A was dimerized with other protein or it was a non-specific band irrelevant with FAM3A. We could not draw an objective conclusion from these findings. We present the results here to provide reference to readers.

Considering complicated factors involved in the quantification of cellular binding/capturing FAM3A and that the results were not reasonable and objective adequately to reflect the real conditions, we did not show these data again in this

revision to avoid misunderstanding. In the final analysis, it's a matter of screening the receptors of FAM3A as well as analyzing the molecular structure of ligand/receptor complex. We stated and discussed the limitations of lack of FAM3A receptor in the Discussion section (please review line17 in paragraph2, line8 in paragraph4, and line3, 7 in paragraph7).

Solutions:

- 1) Please review Supplementary Fig. 1d-1g for these results.
- 2) The revision of Results and Discussion sections: all these results were provided in the Result section "*Cell-specific FAM3A expression changes in AAA microenvironment*", and discussed in Discussion section in paragraph3, line4-10.

Thanks again for your guidance. It is valuable for our next research.

2. The authors state on page 6, line 156: "However, KLF4, a crucial regulator of VSMC phenotypic switching, was not significantly altered at the transcriptional level after FAM3A overexpression at 4-week timepoint in AngII-infused ApoE^{-/-} AAA mice (Fig. 2h)." Yet, the respective plot in Fig. 2h specifies a highly significant p-value ($p=0.0002$) which is not supported by the graph. Please correct.

Response: Thank you very much for your careful review. This p-value is between PBS control and Ad-sham groups. We feel sorry for this wrong label and have made a correction in the Figure. 2h.

3. The paragraph on female ApoE KO mice treated with hrFAM3A (page 6, line 159) was inserted in the chapter on adenoviral overexpression of hrFAM3A and would better fit to the subsequent chapter that is devoted to the administration of hrFAM3A.

Response: Thank you very much for your careful review. We have shifted the results concerning "female ApoE KO mice treated with hrFAM3A" to the proper place in Results section of "*Supplementation with recombinant FAM3A attenuates in vivo pathological outcomes in murine AAA models and maintains the contractile phenotype of VSMCs*", paragraph2.

4. Regarding the cytotoxicity tests (page 8, line 220) with hrFAM3A, I am not entirely sure what is meant by "The results showed that there was no significant toxicity of FAM3A in

VSMCs, endothelial cells, fibroblasts, and macrophages in the baseline (Supplementary Fig. 4f).” Does “in the baseline” refer to the absence of other stimuli in this in vitro experiment?

Response: Thank you very much for your careful review. Really, “in the baseline” refers to the absence of other stimuli in this in vitro experiment. To describe more clearly, we have change the sentence as “*The results showed that recombinant FAM3A had no significant toxicity in the normal cultured VSMCs, endothelial cells, fibroblasts, and macrophages (Supplementary Fig. 4f).*”.

5. Suppl. Fig. 4e: Please specify whether toxicity tests were performed on mouse organs harvested after single hrFAM3A treatment (as for immunofluorescence stainings) or on organs harvested after a complete treatment cycle (as for measurements of aortic diameter).

Response: Thank you very much for your careful review. The toxicity tests performed on mouse organs were harvested after single hrFAM3A treatment. In Supplementary Fig. 4, ApoE^{-/-} mice with “single hrFAM3A treatment” is labeled as “ApoE^{-/-}”, and ApoE^{-/-} mice with “a complete AAA treatment cycle” is labeled as “ApoE^{-/-}+AngII”. We also added the word “normal” prior to “ApoE^{-/-} mice” to make a more clear demonstration of “single hrFAM3A treatment” in the legend of Supplementary Fig. 4e.

6. Suppl. Fig. 4f: Please specify in the legend for how long cell cultures were treated with hrFAM3A and correct the labeling of cell types within the graph (erroneously all labeled “VSCM”).

Response: Thank you very much for your careful review. We feel sorry for these wrong labels with cells and have made corrections in the Supplementary Fig. 4f. Here, the cellular hrFAM3A-exposure time is 48 hours. We have specified this time in the legend of Supplementary Fig. 4f.

Thanks again for your careful review and kind instruction.

Reviewer #3 (Remarks to the Author):

The authors have performed a considerable amount of additional experiments and addressed the concerns properly. I have no further comments.

REVIEWERS' COMMENTS

Reviewer #1 (Remarks to the Author):

The authors have described the limitations of non-cell specificity and lack of FAM3A receptor identification as the major drawbacks of the present study, that were my predominant concerns. To circumvent these issues, in vitro experiment results were discussed (although not the perfect approach) for partly supporting the FAM3A/TGF β /KLF4 axis in SMCs. Although an impressive amount of additional data has been generated, the core questions still remain to be rigorously elucidated.

In my opinion, treatment of recombinant FAM3A on preformed AAAs to demonstrate a therapeutic relevance still remains a relevant issue for a preclinical strategy that has not been addressed. This experiment could be easily performed without the limitations of relevant cell-specific transgenic mouse strains, as above. I am not sure if authors response that 'monitoring and prevention are important, since most disease is an irreversible process' is relevant for most patients with AAAs which is an incidental finding in the clinics. Also, the authors point out that 'screening its familial hereditary features as well as gene polymorphism would be meaningful'. Does that imply that FAM3A mediated signaling is more relevant in genetic disorders like Marfans syndrome and associated aortopathies rather than non-genetic causes of AAAs?

Reviewer #2 (Remarks to the Author):

In their second round of revisions, the authors have addressed all of my remaining concerns. Hence, I do not have any further comments or suggestions.

REVIEWERS' COMMENTS

Reviewer #1 (Remarks to the Author):

The authors have described the limitations of non-cell specificity and lack of FAM3A receptor identification as the major drawbacks of the present study, that were my predominant concerns. To circumvent these issues, in vitro experiment results were discussed (although not the perfect approach) for partly supporting the FAM3A/TGF β /KLF4 axis in SMCs. Although an impressive amount of additional data has been generated, the core questions still remain to be rigorously elucidated.

In my opinion, treatment of recombinant FAM3A on preformed AAAs to demonstrate a therapeutic relevance still remains a relevant issue for a preclinical strategy that has not been addressed. This experiment could be easily performed without the limitations of relevant cell-specific transgenic mouse strains, as above. I am not sure if authors response that 'monitoring and prevention are important, since most disease is an irreversible process' is relevant for most patients with AAAs which is an incidental finding in the clinics. Also, the authors point out that 'screening its familial hereditary features as well as gene polymorphism would be meaningful'. Does that imply that FAM3A mediated signaling is more relevant in genetic disorders like Marfans syndrome and associated aortopathies rather than non-genetic causes of AAAs?

Response: Thanks very much. We wish to express our gratefulness for your careful review of our paper again.

We appreciate that treatment of recombinant FAM3A on preformed AAAs to demonstrate a therapeutic relevance is significant. Indeed, this study suggests that FAM3A is protective during AAA formation (or modeling, progression) process (not FAM3A treatment of the preformed murine AAA models). The FAM3A intervening murine AAA model during the course of 0-28 days (ApoE^{-/-}+AngII) or 0-14 days (C57BL/6+Elastase)^a and FAM3A treatment at preformed models (after 28 days of ApoE^{-/-}+AngII or 14 days of C57BL/6+Elastase)^b are two kinds of experiments. Both a and b are all of significance. As you know, operationally, to perform FAM3A treatment at preformed models (after 28 days of ApoE^{-/-}+AngII or 14 days of C57BL/6+Elastase)^b, it need 14 days for C57BL/6 modeling and 28 days for ApoE^{-/-} modeling, and then identifying of preformed AAA mice by ultrasonic testing, FAM3A treatment for several weeks (such as 1 or 2 weeks), and completing aneurysm pathological-associated measurements (there may be new problems, such as different mechanism in preformed AAA treated by FAM3A, etc.), the entire process need several weeks. It exceeded the revision time. We schemed carefully these experiments. Furthermore, the amount of data along with the article currently is bigger than usual. Considering these things together, we therefore did not supplement these experiments in this article. We hope you will understand. Actually, to explore FAM3A treatment of preformed AAA will be the next study emphases in our laboratory. Concerning this issue, we have further discussed the significance of therapeutic effects of FAM3A on preformed AAA which also deserves a further study (please review the second to last paragraph in Discussion section).

About the perspective of clinical potential applications, we can image that monitoring of FAM3A in general population may have potential value to assess whether or not an individual is at a potential risk of developing AAA (in the general population rather than already formed AAA patients). After all, FAM3A is reduced in AAA patients and we therefore suspect that shortage of FAM3A expression is involved in or aid AAA formation process.

What we mentioned about “screening its familial hereditary features as well as gene polymorphism would be meaningful” is due to the consideration that genetic mutation or change of *fam3a* gene would result in a likely outcome of shortage or change of its expression level. We think that aneurysm cannot be simply induced by genetic disorder following a gene mutation like Marfans syndrome etc. AAA is a complex disease and FAM3A may be one of the numerous complex factors involved in its pathogenesis. From our study, it could not be deduced that AAA is genetic disorder induced by *fam3a* (mutation or other family genetic changes), like that Marfans syndrome is genetic disorder as a consequence of *fbn1* mutation. All these potential perspectives are not overstated in the manuscript and just for academic communication here. Actually, our study is focused on the relationship between FAM3A protein level and AAA formation process.

Thanks again for your careful review and kind instruction.

Reviewer #2 (Remarks to the Author):

In their second round of revisions, the authors have addressed all of my remaining concerns. Hence, I do not have any further comments or suggestions.